# PLASH: Provably Linear-Time Attention with Selective Higher-Order Feature Sketching

**Yuwen Huang** [1]   **Xiang Pan** [2]

## Abstract

Standard softmax attention scales quadratically with sequence length, which makes long-context training and inference expensive. We introduce PLASH, an attention block whose cost grows linearly in the number of keys. PLASH compresses the original keys and values into $M$ learned prototypes, where $M \in \mathbb{Z}_{>0}$ is much smaller than the number of keys. The compressed prototypes are then enriched with randomized polynomial features that recover inter-token information lost to compression. The output is computed by exact scaled dot-product softmax attention from each query to the enriched prototypes, so PLASH preserves the standard attention interface. The construction applies to self- and cross-attention. We prove sketch-error bounds for the enrichment step, a per-input certificate that upper-bounds the deviation from standard softmax attention on each forward pass, and a runtime bound linear in the number of queries and keys. Experiments on long-context language modeling (Qwen3-4B on PG-19) and time-series forecasting (ETT, ECL, Weather) show competitive accuracy and favorable scaling against efficient-attention baselines.

## 1. Introduction

Self-attention couples every token in a sequence to every other token, but its standard form scales quadratically with sequence length (Vaswani et al., 2017). Let $N_q, N_k, d_k, d_v \in \mathbb{Z}_{>0}$ denote the number of queries, number of keys, key dimension, and value dimension respectively. Scaled dot-product attention takes queries $\mathbf{Q} \in$

[1]Thrust of Data Science and Analytics, The Hong Kong University of Science and Technology (Guangzhou), Guangzhou, China. [2]Division of Industrial Data Science, Lingnan University, Hong Kong, China. Correspondence to: Xiang Pan <xxiang-pan@ln.edu.hk>.

*Proceedings of the 43rd International Conference on Machine Learning*, Seoul, South Korea. PMLR 306, 2026. Copyright 2026 by the author(s).

$\mathbb{R}^{N_q \times d_k}$, keys $\mathbf{K} \in \mathbb{R}^{N_k \times d_k}$, and values $\mathbf{V} \in \mathbb{R}^{N_k \times d_v}$, and returns

$$\mathrm{Atten}(\mathbf{Q}; \mathbf{K}, \mathbf{V}) = \mathrm{softmax}\left(\frac{1}{\sqrt{d_k}} \cdot \mathbf{Q} \cdot \mathbf{K}^{\top}\right) \cdot \mathbf{V}.$$

The dominant cost is the $N_q \times N_k$ logit matrix $\mathbf{Q} \cdot \mathbf{K}^{\top}$. When $N_q \approx N_k \approx N$, computing it takes $\mathcal{O}(N^2 d_k)$ arithmetic and stores $\mathcal{O}(N^2)$ attention weights. This quadratic scaling is the main obstacle to long-context training and inference.

**Why hardware optimizations alone are insufficient.** Input / Output-aware (IO-aware) implementations such as *FlashAttention* reduce memory overhead while preserving exact standard-attention expressivity (Dao et al., 2022), extending feasible context lengths to the 200K–1M range (Anthropic, 2024). Standard attention still requires $\mathcal{O}(N^2)$ pairwise scores: roughly $4 \times 10^9$ at $N$=64K and $10^{12}$ at $N$=1M. Long-context scaling therefore demands an *algorithmic* route to sub-quadratic interaction counts, not only faster implementations of the quadratic primitive.

**Two families of prior linear-time methods.** Existing methods reduce the quadratic cost in one of two ways: restrict which token pairs interact, or replace softmax mixing with a structured operator. The first family includes sparse and block-pattern attention (Child et al., 2019; Beltagy et al., 2020; Zaheer et al., 2020), low-rank and Nyström approximations (Wang et al., 2020; Xiong et al., 2021), and hashing-based routing (Kitaev et al., 2020). The second includes kernel and feature linearization (Katharopoulos et al., 2020; Choromanski et al., 2021), Fourier mixers (Lee-Thorp et al., 2022), and state-space or gated-recurrence mixers (Gu et al., 2022a; Gu & Dao, 2024; Poli et al., 2023; De et al., 2024; Yang et al., 2024). Both families scale to long contexts, but their accuracy–efficiency trade-offs depend on architectural choices and hyperparameters. Few of them provide *layer-level* approximation guarantees that remain faithful to the scaled dot-product softmax interface (Tay et al., 2022; 2021).

**Our approach.** We introduce PLASH (**P**rovably **L**inear-Time **A**ttention with **S**elective **H**igher-order feature **S**ketching). Two design choices distinguish it. First, PLASH compresses keys and values to length $M \in \mathbb{Z}_{>0}$ with $M \ll N_k$. Second, it keeps an *exact* scaled dot-product softmax readout from queries to the compressed keys and

values. All approximation sits in a single enrichment operator applied *after* compression:

$$(\mathbf{K}, \mathbf{V}) \xrightarrow{\text{compressing}} (\widetilde{\mathbf{K}}, \widetilde{\mathbf{V}}) \xrightarrow[\text{enrichment}]{\text{feature}} (\mathbf{K}_g, \mathbf{V}_g)$$
$$\xrightarrow{\text{readout}} \mathrm{Atten}(\mathbf{Q}; \mathbf{K}_g, \mathbf{V}_g).$$

This design preserves the softmax interface at the block boundary while localizing approximation to the intermediate enrichment step.

**Comparison with related work.** Efficient attention methods place approximation in two locations. *Random-feature kernel methods* (Choromanski et al., 2021; Peng et al., 2021; Zheng et al., 2023; Kacham et al., 2024) replace softmax with a randomized kernel estimator and mix randomness into every one of the $N_q \times N_k$ query–key interactions; they admit population-level variance bounds. *Deterministic compression methods* (Wang et al., 2020; Xiong et al., 2021; Peng et al., 2022) project $(\mathbf{K}, \mathbf{V})$ onto a smaller subspace with no randomness; they admit spectral bounds. PLASH occupies a third position: deterministic compression (Stage I), randomized higher-order enrichment confined to a single stage (Stage II), and *exact* softmax readout from $\mathbf{Q}$ to the enriched $(\mathbf{K}_g, \mathbf{V}_g)$ (Stage III). This localization yields a forward-pass, *per-input* deviation certificate (Theorem 3.2 in Section 3.2) that complements, rather than replaces, the population-level guarantees of prior work.

**Why higher-order interactions, with explicit budgets.** Stage I averages many tokens into each prototype, so the compressed keys and values retain only per-group means. A linear readout from these means cannot distinguish two groups that share the same mean but differ in higher-order statistics such as variance or covariance. Adding polynomial features of the input coordinates (squared products, triple products, and so on) restores this distinguishing power (Schölkopf & Smola, 2002; Gao et al., 2016). Explicitly enumerating all degree-$k$ products of an input vector in $\mathbb{R}^d$ (for $d \in \mathbb{Z}_{>0}$) costs $d^k$ entries, which becomes impractical even for small $k$. TensorSketch (Pham & Pagh, 2013; Avron et al., 2014; Ahle et al., 2020) approximates the inner product of two degree-$k$ polynomial feature maps using only $D_k \in \mathbb{Z}_{>0}$ random coordinates instead of $d^k$. Stage II applies TensorSketch with two control knobs: a finite set of interaction degrees $\mathcal{K} \subset \mathbb{Z}_{\geq 1}$ (for example, $\mathcal{K} = \{1, 2\}$ retains linear and quadratic terms), and per-degree sketch sizes $\{D_k\}_{k \in \mathcal{K}} \subset \mathbb{Z}_{>0}$ that trade approximation accuracy against compute (Gao et al., 2016; Fukui et al., 2016).

**Per-input certificates from forward-pass quantities.** For each input $(\mathbf{Q}, \mathbf{K}, \mathbf{V})$, PLASH produces an upper bound on the Frobenius deviation between its own output and standard softmax attention on the same input. The bound decomposes into a Stage II concentration term and a deterministic propagation through Stages I and III; both are computed from

the forward pass without forming the $N_q \times N_k$ logit matrix. Given a task-specific tolerance $\epsilon > 0$, the input is *certified* when the bound is at most $\epsilon$, and the corresponding PLASH output is then guaranteed to differ from standard softmax attention by at most $\epsilon$; otherwise the practitioner falls back to standard attention on that input. Theorem 3.2 in Section 3.2 states the bound formally.

### 1.1. Contributions

Our contributions are threefold:

1. **Key-value (KV)-side compression with localized randomness and explicit budgets.** PLASH compresses only the key-value pair $(\mathbf{K}, \mathbf{V})$ to length $M \ll N_k$ and keeps an *exact* softmax readout from $\mathbf{Q}$ to $(\mathbf{K}_g, \mathbf{V}_g)$. All approximation localizes to one TensorSketch (Pham & Pagh, 2013; Avron et al., 2014; Ahle et al., 2020) enrichment step with two explicit knobs: the degree set $\mathcal{K}$ and sketch sizes $\{D_k\}_{k \in \mathcal{K}}$. The compression module is reusable for any subset of the $(\mathbf{Q}, \mathbf{K}, \mathbf{V})$ triple, supporting self- and cross-attention uniformly.

2. **Forward-pass, per-input deviation certificates.** We prove a high-probability $(\epsilon, \delta)$ bound on how far PLASH's output deviates from standard softmax attention on the *same* input. The bound is computable from the forward pass: we first bound the sketching error in Stage II, then propagate it through the deterministic mixer, projections, and exact readout, never forming the $N_q \times N_k$ attention matrix. At inference time, the certificate identifies, without access to ground-truth labels, every input on which PLASH's output may deviate from standard softmax attention by more than the chosen tolerance; population-level variance bounds do not yield this per-input identification.

3. **Evaluation on long-context language modeling and long-sequence forecasting.** We evaluate PLASH on long-context language modeling with Qwen3-4B on PG-19 at $N \in \{16{,}384, 32{,}768\}$ tokens (Section 4.1) and on the Electricity Transformer Temperature (ETT), Electricity Consuming Load (ECL), and Weather (WTH) forecasting benchmarks under the Informer protocol (Zhou et al., 2021; Zhang et al., 2023) (Section 4.2). PLASH attains the lowest EMA training loss at $N$=16K on PG-19 and runs $2.4\times$ faster than Standard FlashAttention-2 at $N$=32K; on forecasting it achieves the best score on WTH and ETTm1 with markedly milder latency and memory growth. The certificate of Theorem 3.2 (Section 3.2) holds on $100\%$ of 35,058 real test inputs across the four forecasting datasets, providing a per-input reliability monitor at inference time without ground-truth labels. A block-wise causal adaptation (Block-Causal PLASH, Ap-

pendix J) extends the construction to autoregressive language modeling at $O\big((B + M)Nd\big)$ cost (block size $B \in \mathbb{Z}_{>0}$, $N_q = N_k = N$) and preserves the certificate, with strict causality verified across 40 leakage tests.

**Conflict of Interest Disclosure.** The authors declare no financial conflicts of interest related to this work. The models and benchmarks evaluated (Qwen3-4B, GPT-2, Informer, and the listed efficient-attention baselines) are open-source artifacts developed by parties unaffiliated with the authors.

## 2. PLASH: A Provably Linear-Time Attention Mechanism

This section formalizes the three stages of PLASH: prototype-based KV compression (Stage I), randomized polynomial-feature enrichment via TensorSketch (Pham & Pagh, 2013; Avron et al., 2014; Ahle et al., 2020) (Stage II), and exact softmax readout from $\mathbf{Q}$ to the enriched KV (Stage III). We begin by recalling standard self-attention and quantifying its quadratic cost (Section 2.1), then formalize the architecture (Section 2.2).

### 2.1. Background: Scaled Dot-Product Attention

Scaled dot-product attention computes a weighted sum of values, where weights come from query–key similarities passed through a row-wise softmax. We let $\mathbf{Q} \in \mathbb{R}^{N_q \times d_k}$, $\mathbf{K} \in \mathbb{R}^{N_k \times d_k}$, and $\mathbf{V} \in \mathbb{R}^{N_k \times d_v}$, where $N_q, N_k, d_k, d_v \in \mathbb{Z}_{>0}$. For $\mathbf{z} \in \mathbb{R}^m$ (with $m \in \mathbb{Z}_{>0}$) and any $j \in [m]$, we define $\mathrm{softmax}(\mathbf{z})_j \triangleq \exp(\mathbf{z}_j)/\big(\sum_{\ell=1}^{m} \exp(\mathbf{z}_\ell)\big)$ and apply it row-wise to a matrix. The scaled dot-product attention operator is (Vaswani et al., 2017):

$$\mathrm{Atten}(\mathbf{Q}; \mathbf{K}, \mathbf{V}) \triangleq \mathrm{softmax}\left(\frac{\mathbf{Q} \cdot \mathbf{K}^\top}{\sqrt{d_k}}\right) \cdot \mathbf{V}. \quad (1)$$

In the Transformer architecture, this single operator serves both self-attention and cross-attention. In self-attention, $\mathbf{Q}$, $\mathbf{K}$, and $\mathbf{V}$ are all derived from the same input sequence (with $N_q = N_k$). In cross-attention, $\mathbf{Q}$ typically comes from a target sequence (e.g., in a decoder), while $\mathbf{K}$ and $\mathbf{V}$ come from a source sequence.

The main cost in (1) is forming the logit matrix $\mathbf{Q} \cdot \mathbf{K}^\top \in \mathbb{R}^{N_q \times N_k}$, which takes $\mathcal{O}(N_q N_k d_k)$ time and typically stores $\mathcal{O}(N_q N_k)$ weights. In self-attention, $N_q \approx N_k \approx N$, so this becomes $\mathcal{O}(N^2 d_k)$ time and $\mathcal{O}(N^2)$ storage, which limits long contexts. Prior work reduces this cost by replacing dense interactions with structured mixing, such as $\mathcal{O}(N \log N)$ Fourier token mixing (Lee-Thorp et al., 2022; Cooley & Tukey, 1965) or $\mathcal{O}(N)$ kernel / random-feature attention (Katharopoulos et al., 2020; Choromanski et al., 2021). These methods are effective, but they often introduce

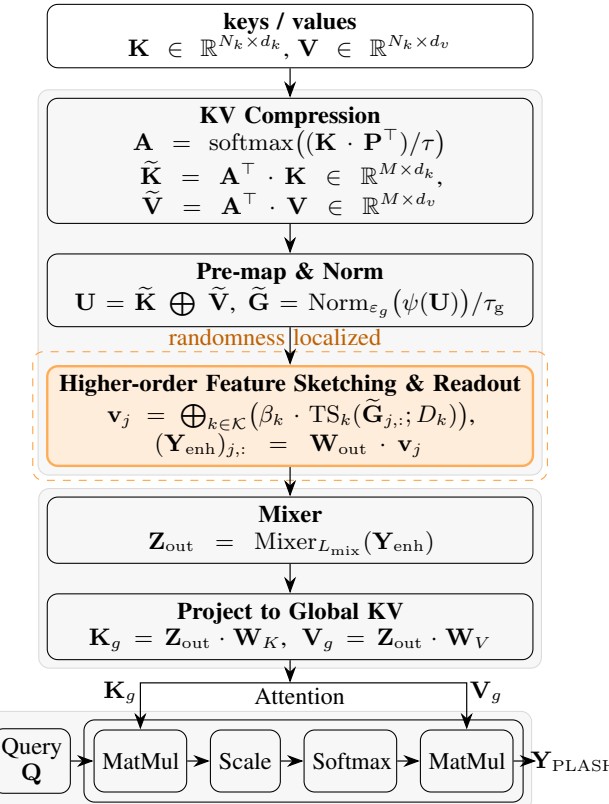

*Figure 1.* PLASH block in the $(\mathbf{Q}, \mathbf{K}, \mathbf{V})$ interface.

extra structure (kernel choices, fixed mixing, or specific approximations), making layer-level comparisons to the softmax attention inaccessible, which motivates the error analysis in Section 3.

### 2.2. The PLASH Block: Architecture and Linear-Time Cost

PLASH achieves linear time by compressing only the key / value side. We keep $\mathbf{Q}$ at full resolution and map $(\mathbf{K}, \mathbf{V})$ to a shorter pair $(\mathbf{K}_g, \mathbf{V}_g)$ of length $M \ll N_k$, then apply standard attention unchanged: $\mathbf{Y}_{\mathrm{PLASH}} \triangleq \mathrm{Atten}(\mathbf{Q}; \mathbf{K}_g, \mathbf{V}_g)$. Thus, PLASH is a drop-in replacement for the usual $(\mathbf{Q}, \mathbf{K}, \mathbf{V})$ block. See Figure 1 for an overview of the three-stage PLASH block.

**Knob 1: compressed length $M$.** Replacing $\mathbf{K}$ by $\mathbf{K}_g$ changes the computation from $\mathbf{Q} \cdot \mathbf{K}^\top \in \mathbb{R}^{N_q \times N_k}$ to $\mathbf{Q} \cdot \mathbf{K}_g^\top \in \mathbb{R}^{N_q \times M}$, reducing the cost from $N_q N_k$ to $N_q M$.

**Knob 2: sketch sizes $\{D_k\}_{k \in \mathcal{K}}$.** After compression, we enrich each compressed item with selective higher-order feature sketches. The finite degree set $\mathcal{K} \subset \mathbb{Z}_{\geq 1}$ chooses which interaction orders are included, and the per-degree sketch sizes $\{D_k\}_{k \in \mathcal{K}} \subset \mathbb{Z}_{>0}$ control the approximation fidelity at each degree $k \in \mathcal{K}$: larger $D_k$ yields a more accurate higher-order representation. Since randomness appears only in this enrichment step, we can certify its error

and propagate it through the deterministic subsequent maps.

### 2.2.1. STAGE I: KV COMPRESSION

This stage compresses the original keys $\mathbf{K} \in \mathbb{R}^{N_k \times d_k}$ and values $\mathbf{V} \in \mathbb{R}^{N_k \times d_v}$ (both with $N_k$ rows) into a compact pair $(\widetilde{\mathbf{K}}, \widetilde{\mathbf{V}})$ with $M \in \mathbb{Z}_{>0}$ rows ($M \ll N_k$) via prototype-based routing and weighted pooling. The goal is to form a compact representation while preserving the query–key–value relationship: $\widetilde{\mathbf{K}}$ stores representative *keys*, and $\widetilde{\mathbf{V}}$ stores the corresponding aggregated *values* from the cluster.

The compression uses a learnable prototype matrix $\mathbf{P} \in \mathbb{R}^{M \times d_k}$, updated via standard training gradients. A soft assignment (routing) is computed as

$$\mathbf{S} \triangleq \mathbf{K}\mathbf{P}^\top \in \mathbb{R}^{N_k \times M}, \tag{2}$$

$$\mathbf{A} \triangleq \mathrm{softmax}(\mathbf{S}/\tau) \in \mathbb{R}^{N_k \times M}, \tag{3}$$

where $\tau \in \mathbb{R}_{>0}$ is a fixed temperature parameter that controls the sharpness of the assignment. Stage I introduces *no algorithmic randomness*: once $(\mathbf{K}, \mathbf{P}, \tau)$ are fixed, $\mathbf{A}$ is fully deterministic. We then use $\mathbf{A}$ to *compress* both $\mathbf{K}$ and $\mathbf{V}$ in one pass:

$$\widetilde{\mathbf{K}} \triangleq \mathbf{A}^\top \cdot \mathbf{K} \in \mathbb{R}^{M \times d_k}, \tag{4}$$

$$\widetilde{\mathbf{V}} \triangleq \mathbf{A}^\top \cdot \mathbf{V} \in \mathbb{R}^{M \times d_v}. \tag{5}$$

Applying $\mathbf{A}$ identically to both $\mathbf{K}$ and $\mathbf{V}$ is the core of the method. It ensures that, for each $j \in [M]$, the row pair $(\widetilde{\mathbf{K}}_{j,:}, \widetilde{\mathbf{V}}_{j,:})$ is a semantically consistent, reduced-dimension representation of a cluster of original key–value pairs. The resulting compression $(\widetilde{\mathbf{K}}, \widetilde{\mathbf{V}})$ is a drop-in replacement for the originals, enabling subsequent attention to operate on a linearly-sized representation. We remark that the mechanism can be adapted to compress any two of the $\mathbf{Q}, \mathbf{K}, \mathbf{V}$ triple.

### 2.2.2. STAGE II: HIGHER-ORDER FEATURE INTERACTION

While Stage I yields a compressed sketch $(\widetilde{\mathbf{K}}, \widetilde{\mathbf{V}})$ of length $M$, this representation may lack the expressivity needed to model complex relationships. Stage II addresses this limitation through two complementary enhancements: (i) enriching each item in the sketch with higher-order interactions via a randomized feature map, and (ii) enabling communication between the $M$ sketch items using a short-sequence mixer. Only the enrichment step employs randomization; the mixer and the final readout remain deterministic components.

**Enrichment via randomized feature maps.** In order to jointly process the keys ($\widetilde{\mathbf{K}}$) and values ($\widetilde{\mathbf{V}}$), we first concatenate them row-wise into a combined representation $\mathbf{U} \in \mathbb{R}^{M \times d_m}$ with $d_m \triangleq d_k + d_v \in \mathbb{Z}_{>0}$. This matrix

undergoes a preprocessing and normalization step:

$$\mathbf{G} \triangleq \psi(\mathbf{U}) \in \mathbb{R}^{M \times d'}, \tag{6}$$

$$\widetilde{\mathbf{G}} \triangleq \mathrm{Norm}_{\varepsilon_g}(\mathbf{G})/\tau_{\mathrm{g}} \in \mathbb{R}^{M \times d'}, \tag{7}$$

where $\psi : \mathbb{R}^{d_m} \to \mathbb{R}^{d'}$ is a row-wise feature map (with $d' \in \mathbb{Z}_{>0}$), $\mathrm{Norm}_{\varepsilon_g}$ (defined in Definition A.4) is a stabilized normalization (similar to LayerNorm in Definition A.5) with stabilization $\varepsilon_g \in \mathbb{R}_{>0}$, and $\tau_{\mathrm{g}} \in \mathbb{R}_{>0}$ is a temperature parameter. This step ensures numerical stability by bounding the row norms of $\widetilde{\mathbf{G}}$, which is critical for controlling the variance of subsequent randomized sketches.

The core enrichment applies a *randomized, higher-order feature map*. For a finite set of polynomial degrees $\mathcal{K} \subset \mathbb{Z}_{\geq 1}$ with learned mixture weights $\{\beta_k\}_{k \in \mathcal{K}} \subset \mathbb{R}_{\geq 0}$ and sketch dimensions $\{D_k\}_{k \in \mathcal{K}} \subset \mathbb{Z}_{>0}$, the enriched row $j \in [M]$ is

$$\mathbf{v}_j \triangleq \bigoplus_{k \in \mathcal{K}} \big(\beta_k \cdot \mathrm{TS}_k(\widetilde{\mathbf{G}}_{j,:}; D_k)\big) \in \mathbb{R}^{D_{\mathrm{tot}}}, \tag{8}$$

$$D_{\mathrm{tot}} \triangleq \sum_{k \in \mathcal{K}} D_k \in \mathbb{Z}_{>0}, \tag{9}$$

$$(\mathbf{Y}_{\mathrm{enh}})_{j,:} \triangleq \mathbf{W}_{\mathrm{out}} \mathbf{v}_j \in \mathbb{R}^d, \tag{10}$$

where $\bigoplus_{k \in \mathcal{K}}$ denotes the concatenation operator (vertical stacking of vectors, formally defined in Definition A.1 in Appendix A), $\mathrm{TS}_k$ is a TensorSketch (Pham & Pagh, 2013; Avron et al., 2014; Ahle et al., 2020) transformation (defined in Appendix C) that approximates a degree-$k$ polynomial kernel, and $\mathbf{W}_{\mathrm{out}} \in \mathbb{R}^{d \times D_{\mathrm{tot}}}$ is a learnable linear readout into the embedding dimension $d \in \mathbb{Z}_{>0}$. The dimensions $\{D_k\}_{k \in \mathcal{K}}$ serve as per-degree *accuracy knobs*: larger $D_k$ yields a more faithful approximation of $k$-th order interactions at a modest increase in computational cost.

**Interaction via a short-sequence mixer.** In order to let the $M$ enriched items in $\mathbf{Y}_{\mathrm{enh}} \in \mathbb{R}^{M \times d}$ exchange information, we apply a lightweight $L_{\mathrm{mix}}$-layer mixer (with $L_{\mathrm{mix}} \in \mathbb{Z}_{>0}$) that operates *across* the length-$M$ axis:

$$\mathbf{Z}_{\mathrm{out}} \triangleq \mathrm{Mixer}_{L_{\mathrm{mix}}}(\mathbf{Y}_{\mathrm{enh}}) \in \mathbb{R}^{M \times d}. \tag{11}$$

Here $\mathrm{Mixer}_{L_{\mathrm{mix}}} : \mathbb{R}^{M \times d} \to \mathbb{R}^{M \times d}$ is the standard $L_{\mathrm{mix}}$-layer post-LayerNorm Transformer encoder applied along the length-$M$ axis (multi-head self-attention followed by a position-wise feed-forward network with residual connections and LayerNorm); the full operator is defined formally in Definition A.7 (Appendix A). Because $M \ll N_k$, this global mixing adds only a small overhead while propagating information across the $M$ compressed items before the final attention readout.

### 2.2.3. STAGE III: FINAL OUTPUT

In the final stage, the compressed representation from Stage II is projected back into the key and value spaces and

then passed through the standard attention operator without further approximation: $\mathbf{K}_g \triangleq \mathbf{Z}_{\text{out}} \cdot \mathbf{W}_K \in \mathbb{R}^{M \times d_k}$ and $\mathbf{V}_g \triangleq \mathbf{Z}_{\text{out}} \cdot \mathbf{W}_V \in \mathbb{R}^{M \times d_v}$, where $\mathbf{W}_K \in \mathbb{R}^{d \times d_k}$ and $\mathbf{W}_V \in \mathbb{R}^{d \times d_v}$ are trainable projection matrices. The output is computed as:

$$\mathbf{Y}_{\text{PLASH}} = \text{softmax}\left( \frac{1}{\sqrt{d_k}} \cdot \mathbf{Q} \cdot \mathbf{K}_g^\top \right) \cdot \mathbf{V}_g. \qquad (12)$$

Overall, PLASH reduces the dominant cost from $N_q \times N_k$ to $N_q \times M$, achieving linear-time complexity while confining all randomness to Stage II; this enables the rigorous error analysis in Section 3. The pseudocode for the entire PLASH block appears as Algorithm 1 in Appendix B.

## 3. Error and Complexity Analysis on PLASH

PLASH concentrates all algorithmic randomness in one Stage II operation. We bound the error from this single source and propagate it through the remaining deterministic computation, without forming the full $N_q \times N_k$ attention matrix.

**Reference.** Given $(\mathbf{Q}, \mathbf{K}, \mathbf{V})$, the reference is standard scaled dot-product attention $\mathbf{Y}_{\text{soft}} \triangleq \text{Atten}(\mathbf{Q}; \mathbf{K}, \mathbf{V})$.

**Where the end-to-end deviation comes from.** The error $\|\mathbf{Y}_{\text{soft}} - \mathbf{Y}_{\text{PLASH}}\|_F$ has three parts: (i) a deterministic Stage I compression error (Appendix F), (ii) a deterministic Stage II reference mismatch (bias), and (iii) the randomized Stage II sketching error after deterministic post-processing. The main text certifies (iii) and states the combined bound; the appendix provides the Stage I bound and full proofs.

### 3.1. Stage II: Certifying the Randomized Feature Map

Stage II is the only source of randomness in PLASH. We therefore certify Stage II at the level of Stage II features and Stage II embeddings, and then propagate the bound through the deterministic post-processing pipeline.

**Implemented sketch features and a deterministic reference.** For each input row $j \in [M]$ and each degree $k \in \mathcal{K}$, we define two degree-$k$ feature vectors in $\mathbb{R}^{D_k}$: an *implemented* (random) feature $\mathbf{z}_{j,k}^{\text{ts}} \triangleq \text{TS}_k(\widetilde{\mathbf{G}}_{j,:}; D_k)$ produced by TensorSketch, and a *deterministic reference* $\mathbf{z}_{j,k}^{\text{det}} \triangleq \widetilde{\mathbf{G}}_{j,:} * \cdots * \widetilde{\mathbf{G}}_{j,:}$ ($k$ copies) given by exact $k$-fold circular convolution. Here $\widetilde{\mathbf{G}} \in \mathbb{R}^{M \times d'}$ is from (7), each row is zero-padded to length $D_k$ (no randomness), and $*$ is the circular convolution on $\mathbb{R}^{D_k}$ (formal definition: Appendix G). Since both vectors lie in $\mathbb{R}^{D_k}$, their $\ell_2$ distance $\|\mathbf{z}_{j,k}^{\text{ts}} - \mathbf{z}_{j,k}^{\text{det}}\|_2$ is the Stage II approximation error: Theorem 3.1 bounds it with high probability, and the appendix propagates this bound through the deterministic post-processing (mixer, projections, exact softmax readout) to obtain the end-to-end certificate. Randomness in $\mathbf{z}_{j,k}^{\text{ts}}$ enters only through the hash / sign seeds inside $\text{TS}_k$, so $\mathbf{z}_{j,k}^{\text{det}}$ is the

natural deterministic target that $\mathbf{z}_{j,k}^{\text{ts}}$ approximates (Pham & Pagh, 2013). Downstream, both $\mathbf{z}$-families are weighted by $\{\beta_k\}_{k \in \mathcal{K}} \subset \mathbb{R}_{\geq 0}$, concatenated via $\bigoplus_{k \in \mathcal{K}}$, and read out by $\mathbf{W}_{\text{out}}$ to form the Stage II embedding $\mathbf{Y}_{\text{enh}}$ of (10) and its deterministic comparator; we defer the mixture / embedding-level analysis to Appendix G, where it is needed for the end-to-end bound.

**What is certified, and why $\tau_{\text{g}}$ is a certification knob.** The certificate controls $\|\mathbf{z}_{j,k}^{\text{ts}} - \mathbf{z}_{j,k}^{\text{det}}\|_2$ uniformly over all aggregated items $j$ and degrees $k$. Two parameters appear with distinct roles: the parameter $D_k$ controls the *probability* of the sketch event (larger $D_k$ yields tighter concentration), and the parameter $\tau_{\text{g}}$ is the Stage II norm-control temperature in (7), which is *deterministic* and monotonically damps *both* the sketched features and the deterministic reference features, thereby tightening the discrepancy bound. Theorem 3.1 quantifies this Stage II discrepancy and serves as the random component of the end-to-end certificate in Theorem 3.2 (Section 3.2), whose certification rate Figure 2 (Section 3.3) tests empirically.

**Theorem 3.1** (Stage II feature discrepancy (uniform, high probability)). *Let* $\mathbf{Q} \in \mathbb{R}^{N_q \times d_k}, \mathbf{K} \in \mathbb{R}^{N_k \times d_k}, \mathbf{V} \in \mathbb{R}^{N_k \times d_v}$ *be the inputs, and let* $\eta, \delta \in (0, 1)$. *The randomness comes from the TensorSketch (Pham & Pagh, 2013; Avron et al., 2014; Ahle et al., 2020) hash / sign family* $\{(h_t^{(k)}, s_t^{(k)})\}_{k \in \mathcal{K}, t \in [k]}$, *which satisfies the limited-independence condition in Assumption C.2 (with the checkable instantiation in Construction C.3). The per-degree budgets* $\{\delta_k\}_{k \in \mathcal{K}} \subset (0, 1)$ *satisfy* $\sum_{k \in \mathcal{K}} \delta_k \leq \delta$, *and the sketch sizes* $\{D_k\}_{k \in \mathcal{K}} \subset \mathbb{Z}_{>0}$ *satisfy* $D_k \geq 2C_k M/(\eta^2 \delta_k)$ *for all* $k \in \mathcal{K}$, *where* $C_k \geq 1$ *is a degree-$k$ constant with* $C_1 = 1$. *Then, with probability at least* $1 - \delta$ *over the sketch randomness, simultaneously for all* $j \in [M]$ *and* $k \in \mathcal{K}$, *it holds that*

$$\|\mathbf{z}_{j,k}^{\text{ts}} - \mathbf{z}_{j,k}^{\text{det}}\|_2 \leq \left( \sqrt{1+\eta} + D_k^{\frac{k-1}{2}} \right) \cdot \tau_{\text{g}}^{-k}. \qquad (13)$$

*Consequently, for any target tolerance* $\epsilon_{\text{feat}} > 0$, *the choice* $\tau_{\text{g}} \geq \max_{k \in \mathcal{K}}\left( (\sqrt{1+\eta} + D_k^{(k-1)/2})/\epsilon_{\text{feat}} \right)^{1/k}$ *implies* $\|\mathbf{z}_{j,k}^{\text{ts}} - \mathbf{z}_{j,k}^{\text{det}}\|_2 \leq \epsilon_{\text{feat}}$ *uniformly over all* $(j, k)$ *on the same probability-$(1-\delta)$ event.* $\qquad \square$

**How to use the Stage II theorem in practice.** The bound in (13) separates two knobs: $\{D_k\}_{k \in \mathcal{K}}$ controls the concentration event (via $D_k \geq 2C_k M/(\eta^2 \delta_k)$), while $\tau_{\text{g}}$ controls the bound magnitude through $\tau_{\text{g}}^{-k}$; our runs (Sections 3.3, 4) use $\mathcal{K} \subseteq \{1, 2\}$, so the $D_k^{(k-1)/2}$ exponent is at most $1/2$.

### 3.2. End-to-End Certificate: From Stage II to the Output

Randomness in PLASH sits entirely in Stage II. Every subsequent computation is deterministic given the inputs and learned parameters. We factor the analysis accordingly: a

randomized feature transformation followed by a deterministic pipeline.

**Theorem 3.2** (Forward-pass computable $(\epsilon_{\mathrm{out}}, \delta)$-certificate). *Let $\epsilon_{\mathrm{out}} \in \mathbb{R}_{>0}$ and $\delta \in (0,1)$. Let $\mathbf{Y}_{\mathrm{soft}} = \mathrm{Atten}(\mathbf{Q};\mathbf{K},\mathbf{V}) \in \mathbb{R}^{N_q \times d_v}$ and let $\mathbf{Y}_{\mathrm{PLASH}} \in \mathbb{R}^{N_q \times d_v}$ be the PLASH output on the same $(\mathbf{Q},\mathbf{K},\mathbf{V})$, with finite degree set $\mathcal{K} \subset \mathbb{Z}_{\geq 1}$ and mixture weights $\{\beta_k\}_{k \in \mathcal{K}} \subset \mathbb{R}_{\geq 0}$ as specified in Algorithm 1 (Appendix B). We assume the Stage II sketch sizes satisfy the sizing rule in Theorem 3.1 (Section 3.1, with budgets $\{\delta_k\}_{k \in \mathcal{K}} \subset (0,1)$ summing to $\delta$) and stabilized normalization so that the post-processing constant $L_{\mathrm{post}}(\mathcal{S}) \in \mathbb{R}_{>0}$ (Lemma H.1, Appendix H) is finite on the line segment $\mathcal{S} \subseteq \mathbb{R}^{M \times d}$. Then, with probability at least $1 - \delta$, $\|\mathbf{Y}_{\mathrm{soft}} - \mathbf{Y}_{\mathrm{PLASH}}\|_F \leq \epsilon_{\mathrm{I}} + \epsilon_{\mathrm{det}} + \epsilon_{\mathrm{II}}$ with $\epsilon_{\mathrm{I}}, \epsilon_{\mathrm{det}}, \epsilon_{\mathrm{II}} \in \mathbb{R}_{\geq 0}$ the deterministic Stage I compression error, the deterministic Stage II reference bias, and the Stage II randomized sketch error after deterministic post-processing, respectively. A sufficient condition on $\tau_{\mathrm{g}}$ that guarantees $\|\mathbf{Y}_{\mathrm{soft}} - \mathbf{Y}_{\mathrm{PLASH}}\|_F \leq \epsilon_{\mathrm{out}}$ is given by* (91) *in Appendix H.* □

**Reading the theorem in the simulated regime $\mathcal{K} = \{1\}$.** All experiments in Figure 2 (details in Section 3.3) use $\mathcal{K} = \{1\}$. In this case, Stage II uses only CountSketch (TensorSketch with $k = 1$), and the deterministic reference reduces to $\mathbf{z}_{j,1}^{\mathrm{det}} = \widetilde{\mathbf{G}}_{j,:}$. Consequently, the end-to-end certificate in Theorem 3.2 (Section 3.2) can be read as an explicit approximation guarantee for $\mathbf{Y}_{\mathrm{soft}} = \mathrm{Atten}(\mathbf{Q};\mathbf{K},\mathbf{V})$. Moreover, since $D_1^{(1-1)/2} = 1$, (13) becomes

$$\|\mathbf{z}_{j,1}^{\mathrm{ts}} - \widetilde{\mathbf{G}}_{j,:}\|_2 \leq \left(\sqrt{1+\eta} + 1\right) \cdot \tau_{\mathrm{g}}^{-1}, \ \forall j \in [M], \ (14)$$

with probability at least $1 - \delta$ under the sizing rule $D_1 \geq 2M/(\eta^2 \delta)$ (taking $\delta_1 = \delta$). This is the precise sense in which $\tau_{\mathrm{g}}$ acts as a deterministic knob in the simulations of Section 3.3: increasing $\tau_{\mathrm{g}}$ monotonically tightens the Stage II discrepancy bound while leaving the downstream architecture unchanged. At the same time, increasing $M$ typically reduces $\epsilon_{\mathrm{I}}$ (compression becomes less lossy), increasing the room available for certification (see Appendix F for a quantitative analysis). These are the monotone trends tested empirically by the heatmaps in Figure 2 (details in Section 3.3).

### 3.3. Empirical Prevalence of the Certificate for $\mathcal{K} = \{1\}$

**Certification rate.** We set $\mathcal{K} = \{1\}$ and measure how often the forward-pass sufficient condition succeeds. Given $T \in \mathbb{Z}_{>0}$ independent and identically distributed (i.i.d.) draws of $(\mathbf{Q},\mathbf{K},\mathbf{V})$, we define $\mathrm{CertRate}(\epsilon_{\mathrm{out}})$ as the fraction of trials for which Theorem 3.2 (Section 3.2) certifies $\|\mathbf{Y}_{\mathrm{soft}} - \mathbf{Y}_{\mathrm{PLASH}}\|_F \leq \epsilon_{\mathrm{out}}$. Concretely, a trial is *certified* when the forward-pass slack is positive and the sufficient inequality (91) in Appendix H holds; this follows the standard

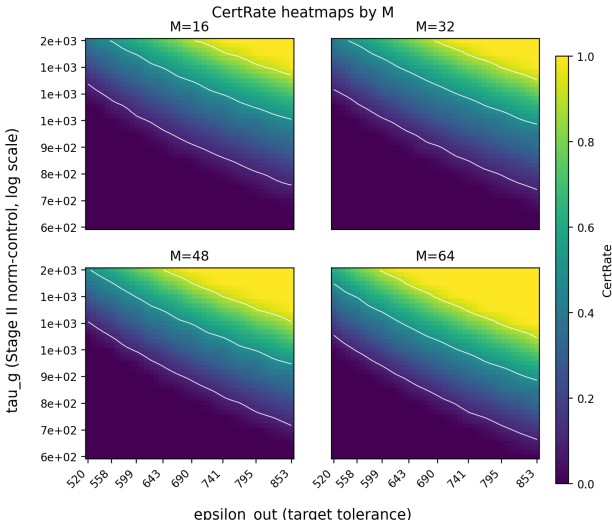

*Figure 2.* Empirical certification rate for PLASH under $\mathcal{K} = \{1\}$. Each cell is the fraction of trials certified at $(\tau_{\mathrm{g}}, \epsilon_{\mathrm{out}})$.

practice of reporting certified rates in robustness certification (e.g., randomized smoothing; Cohen et al., 2019). See Appendix K for simulation details.

Figure 2 sweeps $(\tau_{\mathrm{g}}, \epsilon_{\mathrm{out}})$ and reports $\mathrm{CertRate}$ for several values of $M$. The monotone dependence on $\tau_{\mathrm{g}}$ follows from (14): increasing $\tau_{\mathrm{g}}$ deterministically shrinks the Stage II discrepancy bound and thus tightens the certified approximation error. The trend in $M$ is driven by Stage I: larger $M$ typically reduces the deterministic compression error (and thus $\epsilon_{\mathrm{I}}$), leaving more room at a fixed $\epsilon_{\mathrm{out}}$. Overall, these heatmaps serve as a *non-vacuity diagnostic*: they show, on representative inputs, how often the theory's explicit knobs yield a *nontrivial* (i.e., triggered) forward-pass certificate.

### 3.4. Complexity: Bottleneck and Overhead

PLASH divides the standard quadratic-cost interaction (of size $N_q \times N_k$) into two smaller, structured operations: one of size $N_k \times M$ (Stage I) and another of size $N_q \times M$ (Stage III). The intermediate Stage II computations depend only on $M$, which is much smaller than the sequence length. This separation makes the overall scaling explicit: the dominant cost grows linearly with sequence length, and any additional overhead from improving approximation accuracy is controlled only by the internal sketch dimensions.

**Theorem 3.3** (Baseline complexity of Stages I–III). *For Algorithm 1 with length $M \ll N_k$, assume Stage II implements TensorSketch (Pham & Pagh, 2013; Avron et al., 2014; Ahle et al., 2020) via CountSketch (Charikar et al., 2002) and Fast Fourier Transform (FFT)-based convolutions. If $(d, d', d_k, d_v, \mathcal{K}, \{D_k\}_{k \in \mathcal{K}}, L_{\mathrm{mix}})$ are fixed independently of $(N_q, N_k)$, then the run time satisfies $T_{\mathrm{PLASH}} = O\big((N_q + N_k) \cdot M\big)$. If we store the Stage I soft routing matrix $\mathbf{A} \in \mathbb{R}^{N_k \times M}$ from (3) and the Stage III readout weights in*

*Table 1.* EMA of training loss at $N$=16,384 on PG-19 (Qwen3-4B). Lower is better.

| Method | Type | EMA |
|---|---|---|
| **PLASH $M$=16 (ours)** | Prototype-sketch | **23.5** |
| Local | Sparse (window) | 24.4 |
| Qwen3 GQA | Grouped-query | 24.4 |
| PLASH $M$=32 (ours) | Prototype-sketch | 24.5 |
| Standard | Full quadratic | 25.2 |
| Longformer | Window-global | 25.5 |
| PLASH $M$=8 (ours) | Prototype-sketch | 30.1 |
| ABC | Latent bottleneck | 35.1 |

$\mathbb{R}^{N_q \times M}$, *then the peak memory is* $O\big((N_q + N_k) \cdot M\big)$; *both matrices can be avoided by streaming.* □

The proof of Theorem 3.3 is in Appendix I. Imposing the certificate of Theorem 3.2 (Section 3.2) only constrains the sketch sizes $\{D_k\}_{k \in \mathcal{K}}$; the $O(N_k M)$ and $O(N_q M)$ structure of Stages I and III is unchanged, and $\tau_g$ tightens the bound without affecting arithmetic cost.

## 4. Empirical Validation

We evaluate PLASH on two complementary tasks: long-context language modeling on PG-19 (Section 4.1) and long-sequence time-series forecasting on ETT, ECL, and Weather (Section 4.2). Each subsection lists its own backbone, baselines, and training recipe; Section 4.3 reports memory and latency scaling.

### 4.1. Long-Context Language Modeling on PG-19 (Qwen3-4B)

**Backbone, protocol, and hyper-parameters.** We train `Qwen3-4B` (Yang et al., 2025) (Grouped-Query Attention (GQA) with 32 query / 8 key-value heads, SwiGLU Feed-Forward Network (FFN), Rotary Position Embedding (RoPE), $d$=2560) on PG-19 (Rae et al., 2020) at sequence lengths $N \in \{16{,}384, 32{,}768\}$. All methods share AdamW, $\mathrm{lr} = 10^{-4}$, effective batch 16; only the attention module differs. We report the Exponential Moving Average (EMA, decay 0.999) of training loss; all runs use Fully Sharded Data Parallel (FSDP) with bfloat16 (bf16) on A800-80GB GPUs.

**Baselines.** Five efficient-attention baselines span four paradigms: *Standard* via FlashAttention-2 (Dao, 2024), *Qwen3 GQA* (also FlashAttention-2), *Local* (fixed-window sparse), *Longformer* (Beltagy et al., 2020) (sliding window with global tokens), and *ABC* (Peng et al., 2022) (latent bottleneck). We compare against PLASH with $M \in \{8, 16, 32\}$ prototypes.

**Best modeling loss at long context.** At $N$=16,384 (Table 1), PLASH $M$=16 achieves 23.5, the *lowest* EMA loss among all eight methods, beating both *Local* and *Qwen3*

*Table 2.* EMA loss at $N$=32,768 on PG-19 (Qwen3-4B). Lower is better.

| Method | Type | EMA |
|---|---|---|
| **PLASH $M$=32 (ours)** | Prototype-sketch | **40.1** |
| Standard | Full quadratic | 40.1 |
| PLASH $M$=8 (ours) | Prototype-sketch | 41.2 |
| Local | Sparse (window) | 42.7 |
| PLASH $M$=16 (ours) | Prototype-sketch | 43.4 |
| Qwen3 GQA | Grouped-query | 43.6 |
| ABC | Latent bottleneck | 44.0 |
| Longformer | Window-global | 45.0 |

*Table 3.* Forward-pass time (ms per layer) and speedup over Standard at $N$=32K. Qwen3-4B, A800-80GB, batch 1, bf16; both Standard and Qwen3 GQA use FlashAttention-2.

| Method | $N$=4K | $N$=16K | $N$=32K | 32K speed-up |
|---|---|---|---|---|
| **PLASH $M$=8 (ours)** | 237 | 909 | **1809** | 2.4× |
| PLASH $M$=16 (ours) | 273 | 1030 | 2049 | 2.1× |
| Local | 302 | 1179 | 2370 | 1.8× |
| PLASH $M$=32 (ours) | 335 | 1273 | 2536 | 1.7× |
| ABC | 386 | 1519 | 3048 | 1.4× |
| Longformer | 699 | 1688 | 3430 | 1.3× |
| Qwen3 GQA | 244 | 1471 | 4348 | 1.0× |
| Standard | 243 | 1463 | 4358 | 1.0× |

*GQA* (each 24.4) by 3.8% and *Standard* FlashAttention-2 (25.2) by 6.7%; *ABC* diverges (best 35.1). At $N$=32,768 (Table 2), PLASH $M$=32 ties Standard at 40.1 while every efficient baseline falls behind (Local 42.7, Qwen3 GQA 43.6, ABC 44.0, Longformer 45.0).

**Fastest efficient method and drop-in causal variant.** At $N$=32,768 (Table 3), PLASH $M$=8 is 2.4× faster than Standard FlashAttention-2 (1809 vs 4358 ms per layer) and the fastest of all efficient baselines (1.3× over Local, 1.7× over ABC, 1.9× over Longformer). PLASH is the only method both faster *and* more accurate than every efficient baseline at 32K, and the speed-up over Standard widens with $N$ (1.6× at $N$=16K to 2.4× at $N$=32K) reflecting the linear-in-$N_k$ cost. The construction is also drop-in for autoregressive language modeling via *Block-Causal* PLASH (Appendix J), which preserves causality up to floating-point noise (max output difference $\leq 3.24 \times 10^{-5}$ across 40 verification tests) and outperforms ABC at GPT-2 (Radford et al., 2019) scales of 124M and 1.5B on TinyStories (Eldan & Li, 2023).

### 4.2. Time-Series Forecasting Accuracy

**Backbone, protocol, and hyper-parameters.** We study Long-Sequence Time-series Forecasting (LSTF) on ETT (ETTh1 / ETTh2 / ETTm1), ECL, and WTH under the multivariate Informer protocol (Zhou et al., 2021; Zhang et al., 2023) (same data splits and metrics). We use Informer as the backbone and replace *only* its attention block with PLASH, leaving embeddings, feed-forward sublayers, normalization,

and the training recipe unchanged: batch size 32, 6 epochs, Adam (betas $= (0.9, 0.999)$), peak $\text{lr} = 1 \times 10^{-4}$ with exponential decay, attention dropout 0.05, no weight decay, no gradient clipping, no warm-up (Weather caps tokens per batch at 3). We report Mean Squared Error (MSE) and Mean Absolute Error (MAE) on the test set, averaged over the protocol's prediction horizons.

**Baselines.** We compare against seven efficient-attention families: (i) sparse / windowed attention (Local; (Child et al., 2019; Beltagy et al., 2020; Zaheer et al., 2020)), (ii) random-feature / kernel linearization (Performer (Choromanski et al., 2021) and CosFormer (Qin et al., 2022)), (iii) randomized softmax estimators (LARA (Zheng et al., 2022)), (iv) low-rank approximations (Nystromformer (Xiong et al., 2021)), (v) bounded-context attention (ABC (Peng et al., 2022)), (vi) hybrid local-with-global attention (LongShort (Zhu et al., 2021)), and (vii) state-space sequence layers (S4D (Gu et al., 2022b)). Vanilla (standard softmax attention) is the reference.

**Main result.** Table 4 shows that PLASH is consistently competitive with strong efficient-attention baselines, achieving the best or second-best result on most datasets. This result supports the PLASH design, in which we compress $(\mathbf{K}, \mathbf{V})$ to length $M \ll N_k$, enrich the compressed representation with selective higher-order feature sketches, and keep the final softmax readout from $\mathbf{Q}$ to $(\mathbf{K}_g, \mathbf{V}_g)$ exact.

**Where PLASH wins.** PLASH sets the best WTH score (MSE 0.3009, MAE 0.3492) and yields strong improvements on ETTh2 and ETTm1. On ECL and ETTh1, full attention / LongShort remain best, and PLASH is close and typically ranks second. Concretely: on WTH, PLASH with $M = 512$ and $D_k = 128$ is best overall (beating Performer); on ETTh2, PLASH with $M = 512$ and $D_k = 128$ improves over the best baseline; on ETTm1, PLASH with $M = 512$ and $D_k = 256$ is best overall. These gains are consistent with (i) compressed KV summaries that reduce irrelevant token mixing and (ii) higher-order enrichment that restores expressive interactions. The linear-in-$N_k$ cost also unlocks longer inputs: at $L$=336 (vs $L$=96 in Table 4), PLASH outperforms standard attention on all four datasets, demonstrating that the linear-complexity advantage translates to better accuracy when more context is admitted (Appendix O). Empirically, the deviation certificate of Theorem 3.2 (Section 3.2) holds on 100% of 35,058 real test samples across all four datasets with zero violations, providing a per-input reliability monitor at inference time without ground-truth labels (Appendix L).

**Why accuracy is not monotone in $M$ and $D_k$.** PLASH has two main knobs: $M$ (compressed KV length) and $D_k$ (sketch size). Test accuracy need not improve monotonically with either knob because they change both approximation error and training dynamics. Smaller $D_k$ produces

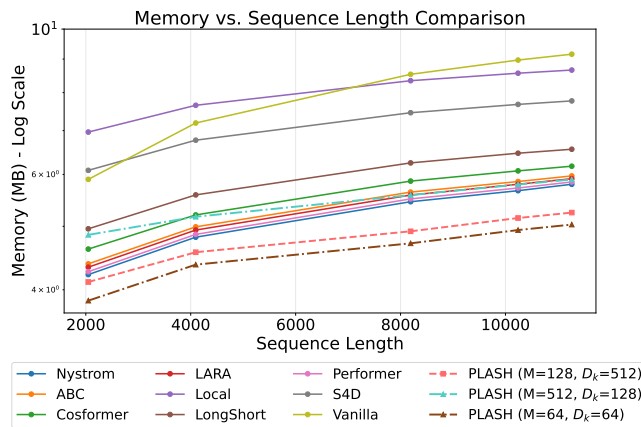

*(a)* Peak storage across methods.

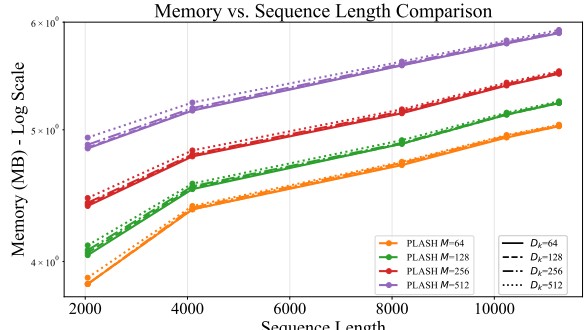

*(b)* Peak storage of PLASH across $M$ and $D_k$.

*Figure 3.* Peak storage comparisons.

noisier sketches; this can help as implicit regularization or hurt by perturbing gradients, so larger $D_k$ does not guarantee better generalization (Pham & Pagh, 2013; Woodruff, 2014). Increasing $M$ reduces compression bias, but it also reduces averaging within each compressed item, which can make training more sensitive to noise and routing changes (Wang et al., 2020; Xiong et al., 2021). Under a fixed backbone and training schedule, $(M, D_k)$ therefore behaves like a standard hyperparameter choice, and the best setting is dataset-dependent (Tay et al., 2022).

### 4.3. Memory and Latency Scaling

**Protocol.** We evaluate efficiency using the standard attention-layer benchmark protocol used in prior efficient-attention comparisons (Zhang et al., 2023). We feed synthetic sequences of lengths $\{2048, 4096, 8192, 10240, 11264\}$ into each attention block, with embedding dimension 512, 4 heads, and batch size 1. We run 100 forward passes per configuration and report average latency and peak storage.

**Peak storage.** Figure 3 shows that Vanilla storage grows quickly with sequence length, while PLASH stays substantially lower and is comparable to other efficient baselines at long contexts. This matches the PLASH design: it re-

*Table 4.* Performance Comparison of Methods Across Multiple Datasets. The top-performing and the second-best methods in each dataset are highlighted in red and blue, respectively. Rankings are determined primarily by MSE, with MAE used as a secondary metric.

| Method | ECL | | WTH | | ETTh1 | | ETTh2 | | ETTm1 | |
|---|---|---|---|---|---|---|---|---|---|---|
| | MSE | MAE | MSE | MAE | MSE | MAE | MSE | MAE | MSE | MAE |
| **Vanilla** | 0.2370 | 0.3452 | 0.3121 | 0.3616 | 0.5553 | 0.5480 | 0.9789 | 0.7961 | 0.3318 | 0.3653 |
| **ABC** | 0.2458 | 0.3527 | 0.3192 | 0.3761 | 0.6870 | 0.6361 | 1.0734 | 0.8173 | 0.3636 | 0.4037 |
| **Performer** | 0.2548 | 0.3629 | 0.3075 | 0.3618 | 0.5065 | 0.5192 | 1.0104 | 0.8037 | 0.3124 | 0.3671 |
| **Local** | 0.2542 | 0.3654 | 0.3185 | 0.3591 | 0.6091 | 0.5824 | 1.3138 | 0.9227 | 0.3144 | 0.3673 |
| **Nystromformer** | 0.2474 | 0.3520 | 0.3141 | 0.3721 | 0.5752 | 0.5601 | 0.9888 | 0.8169 | 0.3691 | 0.3875 |
| **LARA** | 0.2839 | 0.3824 | 0.3104 | 0.3656 | 0.7711 | 0.6677 | 1.0763 | 0.8273 | 0.3906 | 0.4114 |
| **CosFormer** | 0.2775 | 0.3848 | 0.3394 | 0.3862 | 0.4965 | 0.5242 | 1.1316 | 0.8790 | 0.4014 | 0.4437 |
| **LongShort** | 0.2708 | 0.3844 | 0.3413 | 0.4007 | 0.4125 | 0.4399 | 0.5683 | 0.6067 | 0.3869 | 0.4248 |
| **S4D** | 0.2480 | 0.3559 | 0.3129 | 0.3570 | 0.5038 | 0.5080 | 1.2196 | 0.8994 | 0.3508 | 0.3879 |
| **PLASH ($M$=64)** | | | | | | | | | | |
| $D_k$= 64 | 0.2576 | 0.3625 | 0.3166 | 0.3642 | 0.5667 | 0.5606 | 0.6937 | 0.6821 | 0.3112 | 0.3855 |
| $D_k$=128 | 0.2430 | 0.3527 | 0.3137 | 0.3650 | 0.5747 | 0.5585 | 1.0193 | 0.7976 | 0.3217 | 0.3669 |
| $D_k$=256 | 0.2482 | 0.3561 | 0.3082 | 0.3595 | 0.5075 | 0.5027 | 0.9820 | 0.7935 | 0.3314 | 0.3896 |
| $D_k$=512 | 0.2483 | 0.3561 | 0.3106 | 0.3570 | 0.5075 | 0.5027 | 0.9667 | 0.8158 | 0.3467 | 0.3919 |
| **PLASH ($M$=128)** | | | | | | | | | | |
| $D_k$=64 | 0.2435 | 0.3527 | 0.3057 | 0.3596 | 0.5636 | 0.5684 | 0.8993 | 0.7624 | 0.3292 | 0.3692 |
| $D_k$=128 | 0.2473 | 0.3537 | 0.3061 | 0.3580 | 0.4974 | 0.5089 | 0.7668 | 0.7271 | 0.3226 | 0.3872 |
| $D_k$=256 | 0.2453 | 0.3568 | 0.3096 | 0.3650 | 0.5074 | 0.5026 | 0.7328 | 0.6803 | 0.3083 | 0.3685 |
| $D_k$=512 | 0.2444 | 0.3543 | 0.3009 | 0.3507 | 0.5508 | 0.5497 | 0.5236 | 0.5790 | 0.3280 | 0.3917 |
| **PLASH ($M$=256)** | | | | | | | | | | |
| $D_k$=64 | 0.2474 | 0.3520 | 0.3141 | 0.3721 | 0.5199 | 0.5236 | 0.9888 | 0.8169 | 0.3691 | 0.3875 |
| $D_k$=128 | 0.2596 | 0.3700 | 0.3072 | 0.3621 | 0.4711 | 0.4872 | 0.7771 | 0.7045 | 0.3089 | 0.3657 |
| $D_k$=256 | 0.2496 | 0.3565 | 0.3115 | 0.3648 | 0.5074 | 0.5027 | 0.8308 | 0.7167 | 0.3156 | 0.3816 |
| $D_k$=512 | 0.2484 | 0.3590 | 0.3058 | 0.3638 | 0.4553 | 0.4960 | 0.6421 | 0.6312 | 0.3047 | 0.3649 |
| **PLASH ($M$=512)** | | | | | | | | | | |
| $D_k$=64 | 0.2546 | 0.3632 | 0.3067 | 0.3558 | 0.4769 | 0.5055 | 0.6989 | 0.6849 | 0.2945 | 0.3616 |
| $D_k$=128 | 0.2478 | 0.3567 | 0.3009 | 0.3492 | 0.4873 | 0.5024 | 0.4700 | 0.5686 | 0.3057 | 0.3682 |
| $D_k$=256 | 0.2578 | 0.3668 | 0.3015 | 0.3510 | 0.4722 | 0.5072 | 0.5987 | 0.6010 | 0.2896 | 0.3550 |
| $D_k$=512 | 0.2632 | 0.3737 | 0.3081 | 0.3572 | 0.4725 | 0.4944 | 0.8952 | 0.7459 | 0.3452 | 0.4021 |

places the dense $N_q \times N_k$ interaction by two rectangular interfaces, $N_k \times M$ (Stage I) and $N_q \times M$ (Stage III), plus Stage II work that depends only on $M$ and $\{D_k\}_{k\in\mathcal{K}}$ (Theorem 3.3). Within PLASH, $M$ is the primary storage knob, while $\{D_k\}_{k\in\mathcal{K}}$ adds a smaller overhead from the feature sketches.

**Run-time scaling.** Table 5 shows the practical impact of removing the quadratic $N_q N_k$ term (Theorem 3.3); times are in milliseconds. As sequence length grows from 2048 to 11,264, Vanilla latency rises from 2.43 to 58.95 ms while length-sensitive baselines grow sharply (Local: $4.72 \rightarrow 25.58$; S4D: $3.79 \rightarrow 31.22$). PLASH at $M$=64 stays within 2.4–2.8 ms across all lengths ($\approx 25\times$ Vanilla at 11,264); at $M$=512 it grows from 2.3 to 4.9 ms, comparable to the fastest efficient baseline (ABC at 4.35 ms, $L$=11,264). Small non-monotone fluctuations across $(M, D_k)$ reflect GPU kernel selection and FFT padding, not the arithmetic-count scaling (Dao et al., 2022; NVIDIA, 2023; 2024).

## 5. Conclusion

PLASH is a provably linear-time $(\mathbf{Q}, \mathbf{K}, \mathbf{V})$ attention block that compresses keys and values to $M \ll N_k$ prototypes, enriches them with randomized higher-order sketches, and

*Table 5.* Run-time Comparison across Sequence Length. The fastest and the second-fastest methods at each sequence length are highlighted in red and blue, respectively.

| Method | Sequence Length | | | | |
|---|---|---|---|---|---|
| | 2048 | 4096 | 8192 | 10240 | 11264 |
| Nystrom | 3.55 | 3.50 | 4.27 | 4.09 | 4.56 |
| ABC | 1.12 | 1.65 | 3.00 | 4.07 | 4.35 |
| Cosformer | 2.22 | 2.84 | 5.12 | 6.33 | 6.95 |
| LARA | 2.18 | 2.33 | 4.11 | 5.12 | 5.65 |
| Local | 4.72 | 9.34 | 18.60 | 23.24 | 25.58 |
| LongShort | 2.58 | 3.92 | 7.40 | 9.21 | 10.19 |
| Performer | 1.79 | 1.92 | 4.45 | 5.52 | 6.10 |
| S4D | 3.79 | 8.00 | 16.34 | 24.68 | 31.22 |
| Vanilla | 2.43 | 8.33 | 31.84 | 48.97 | 58.95 |
| **PLASH ($M$=64)** | | | | | |
| $D_k$= 64 | 2.49 | 2.52 | 2.71 | 2.51 | 2.69 |
| $D_k$=128 | 2.49 | 2.56 | 2.42 | 2.62 | 2.36 |
| $D_k$=256 | 2.59 | 2.59 | 2.65 | 2.79 | 2.63 |
| $D_k$=512 | 2.43 | 2.51 | 2.76 | 2.66 | 2.39 |
| **PLASH ($M$=512)** | | | | | |
| $D_k$= 64 | 2.38 | 2.86 | 3.63 | 4.51 | 4.89 |
| $D_k$=128 | 2.56 | 2.56 | 3.62 | 4.44 | 4.90 |
| $D_k$=256 | 2.34 | 2.72 | 3.60 | 4.52 | 4.89 |
| $D_k$=512 | 2.34 | 2.80 | 3.94 | 4.55 | 4.94 |

reads out via an exact softmax compatible with FlashAttention (Dao et al., 2022), with a kernel-agnostic per-input deviation certificate. Experiments confirm competitive accuracy at markedly milder latency growth.

## Acknowledgements

The work described in this paper was partially supported by the Hong Kong University of Science and Technology (Guangzhou) under Grant (G0101000351) and Lingnan University under Grants (SDS24A16).

## Impact Statement

This paper presents work whose goal is to advance the field of Machine Learning by improving the efficiency of attention mechanisms for long-context modeling. The primary expected benefit is reduced computation and energy use for training and inference, which may lower the cost of deploying long-context models. As with many advances in efficient modeling, the technique could also be used to scale up applications that process sensitive data; responsible use should therefore follow standard practices for privacy, security, and data governance. We do not anticipate additional ethical concerns beyond those already well established for deploying large-scale sequence models.

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

# A. Notation and Basic Operators

**Index set.** For any positive integer $M \in \mathbb{Z}_{>0}$, we define

$$[M] \triangleq \{1, 2, \ldots, M\}.$$

**Definition A.1** (Concatenation operator $\bigoplus$ (vector stacking / external direct sum)). Let $m_1, m_2, M \in \mathbb{Z}_{>0}$. The symbol $\bigoplus$ denotes vector concatenation: for $\mathbf{a} \in \mathbb{R}^{m_1}$ and $\mathbf{b} \in \mathbb{R}^{m_2}$,

$$\mathbf{a} \bigoplus \mathbf{b} \triangleq \begin{pmatrix} \mathbf{a} \\ \mathbf{b} \end{pmatrix} \in \mathbb{R}^{m_1 + m_2}.$$

The indexed form $\bigoplus_{k \in \mathcal{K}} \mathbf{a}_k \in \mathbb{R}^{\sum_{k \in \mathcal{K}} m_k}$ iterates this construction over the ordered index set $\mathcal{K}$. For matrices with the same number of rows, concatenation is applied row-wise: if $\mathbf{A} \in \mathbb{R}^{M \times m_1}$ and $\mathbf{B} \in \mathbb{R}^{M \times m_2}$, then $\mathbf{A} \bigoplus \mathbf{B} \in \mathbb{R}^{M \times (m_1 + m_2)}$ has $i$-th row $(\mathbf{A}_{i,:} \bigoplus \mathbf{B}_{i,:})^\top$. △

**Norm conventions.** For a matrix $\mathbf{X} \in \mathbb{R}^{m \times n}$ and a vector $\mathbf{u} \in \mathbb{R}^m$, we use:

$$\|\mathbf{X}\|_{2,\infty} \triangleq \max_{i \in [m]} \|\mathbf{X}_{i,:}\|_2, \qquad \|\mathbf{u}\|_\infty \triangleq \max_{i \in [m]} |u_i|, \qquad \|\mathbf{X}\|_F \triangleq \left( \sum_{i=1}^m \sum_{j=1}^n X_{ij}^2 \right)^{1/2}.$$

**Definition A.2** (Temperature-scaled row softmax). Throughout the paper, $\mathrm{softmax}(\cdot)$ applied to a matrix denotes the row-wise softmax: for $n, m \in \mathbb{Z}_{>0}$ and $\mathbf{S} \in \mathbb{R}^{n \times m}$, the $i$-th row of $\mathrm{softmax}(\mathbf{S})$ is $\mathrm{softmax}(\mathbf{S}_{i,:})$ as defined for vectors in (1). For temperature $\tau \in \mathbb{R}_{>0}$, the temperature-scaled row softmax $\mathrm{softmax}(\mathbf{S}/\tau)$ rescales the logits by $1/\tau$ before the row-wise softmax. △

**Definition A.3** (Spectral / Operator Norm). For any $\mathbf{W} \in \mathbb{R}^{N \times N}$ and any $N \in \mathbb{Z}_{>0}$, the spectral (operator) norm induced by the Euclidean norm is

$$\|\mathbf{W}\|_{\mathrm{op}} \triangleq \sup_{\mathbf{x} \neq \mathbf{0}} \frac{\|\mathbf{W} \cdot \mathbf{x}\|_2}{\|\mathbf{x}\|_2} = \sup_{\|\mathbf{x}\|_2 = 1} \|\mathbf{W} \cdot \mathbf{x}\|_2.$$

Equivalently, $\|\mathbf{W}\|_{\mathrm{op}} = \sqrt{\lambda_{\max}(\mathbf{W}^\top \cdot \mathbf{W})}$. △

**Definition A.4** (Pre-map and stabilized row normalization). Let $d_m, d', M \in \mathbb{Z}_{>0}$. A *pre-map* is a differentiable function $\psi : \mathbb{R}^{d_m} \to \mathbb{R}^{d'}$ applied row-wise to matrices in $\mathbb{R}^{M \times d_m}$.

For $\varepsilon_g \in \mathbb{R}_{>0}$, we define the stabilized normalization map $\mathrm{Norm}_{\varepsilon_g} : \mathbb{R}^{d'} \to \mathbb{R}^{d'}$ by

$$\mathrm{Norm}_{\varepsilon_g}(\mathbf{g}) \triangleq \frac{\mathbf{g}}{\max\{\|\mathbf{g}\|_2, \varepsilon_g\}}.$$

Thus $\|\mathrm{Norm}_{\varepsilon_g}(\mathbf{g})\|_2 \leq 1$, with denominator bounded below by $\varepsilon_g$. △

**Definition A.5** (Layer normalization). Let $d \in \mathbb{Z}_{>0}$, $\gamma, \beta \in \mathbb{R}^d$ be trainable parameters, and let $\varepsilon_{\mathrm{ln}} \in \mathbb{R}_{>0}$. For $\mathbf{u} \in \mathbb{R}^d$, we define

$$\mu(\mathbf{u}) \triangleq \frac{1}{d} \sum_{r=1}^d u_r, \qquad \mathrm{Var}(\mathbf{u}) \triangleq \frac{1}{d} \sum_{r=1}^d \big(u_r - \mu(\mathbf{u})\big)^2, \qquad \mathrm{LayerNorm}(\mathbf{u}) \triangleq \gamma \odot \frac{\mathbf{u} - \mu(\mathbf{u}) \cdot \mathbf{1}}{\sqrt{\mathrm{Var}(\mathbf{u}) + \varepsilon_{\mathrm{ln}}}} + \beta,$$

where $\odot$ denotes elementwise multiplication and $\mathbf{1} \in \mathbb{R}^d$ is the all-ones vector. For $\mathbf{U} \in \mathbb{R}^{M \times d}$, $\mathrm{LayerNorm}(\mathbf{U})$ is applied row-wise (Ba et al., 2016). △

**Definition A.6** (TensorSketch primitive (Pham & Pagh, 2013; Avron et al., 2014; Ahle et al., 2020) (abstract form))**.** Let $k, D_k, d' \in \mathbb{Z}_{>0}$. The degree-$k$ TensorSketch map is a randomized feature map

$$\mathrm{TS}_k(\cdot; D_k) : \mathbb{R}^{d'} \to \mathbb{R}^{D_k},$$

implemented via CountSketch hashing and FFT-based convolution (Charikar et al., 2002; Cooley & Tukey, 1965; Pham & Pagh, 2013); see Section C for the explicit construction and concentration guarantees. $\triangle$

**Definition A.7** (Mixer$_{L_{\mathrm{mix}}}$ (Transformer encoder on length $M$))**.** Let $M, d, L_{\mathrm{mix}} \in \mathbb{Z}_{>0}$, and let $\mathbf{Y}^{(0)} \in \mathbb{R}^{M \times d}$ be the input length-$M$ sequence. Let $H \in \mathbb{Z}_{>0}$ denote the number of heads and $d_h \in \mathbb{Z}_{>0}$ the head width, with $Hd_h = d$.

For each layer $\ell \in \{1, \ldots, L_{\mathrm{mix}}\}$ and head $h \in \{1, \ldots, H\}$, let $\mathbf{W}_Q^{(\ell,h)}, \mathbf{W}_K^{(\ell,h)}, \mathbf{W}_V^{(\ell,h)} \in \mathbb{R}^{d \times d_h}$ and $\mathbf{W}_O^{(\ell)} \in \mathbb{R}^{d \times d}$ be learnable parameters. We define the multi-head self-attention (MHSA) map $\mathrm{MHSA}^{(\ell)} : \mathbb{R}^{M \times d} \to \mathbb{R}^{M \times d}$ by

$$\mathrm{MHSA}^{(\ell)}(\mathbf{U}) \triangleq \left( \bigoplus_{h=1}^H \mathrm{softmax}\left( \frac{1}{\sqrt{d_h}} \cdot \left( \mathbf{U} \cdot \mathbf{W}_Q^{(\ell,h)} \right) \cdot \left( \mathbf{U} \cdot \mathbf{W}_K^{(\ell,h)} \right)^\top \right) \cdot \left( \mathbf{U} \cdot \mathbf{W}_V^{(\ell,h)} \right) \right) \cdot \mathbf{W}_O^{(\ell)}.$$

Here $\bigoplus_{h=1}^H$ denotes concatenation along the feature dimension (so the result has width $Hd_h = d$).

Let $d_{\mathrm{ff}} \in \mathbb{Z}_{>0}$ be the FFN hidden width, and let $\mathbf{W}_1^{(\ell)} \in \mathbb{R}^{d \times d_{\mathrm{ff}}}$, $\mathbf{W}_2^{(\ell)} \in \mathbb{R}^{d_{\mathrm{ff}} \times d}$, $\mathbf{b}_1^{(\ell)} \in \mathbb{R}^{d_{\mathrm{ff}}}$, and $\mathbf{b}_2^{(\ell)} \in \mathbb{R}^d$ be learnable parameters. We define the position-wise feed-forward map $\mathrm{FFN}^{(\ell)} : \mathbb{R}^{M \times d} \to \mathbb{R}^{M \times d}$ by

$$\mathrm{FFN}^{(\ell)}(\mathbf{U}) \triangleq \sigma\left( \mathbf{U} \cdot \mathbf{W}_1^{(\ell)} + \mathbf{1} \cdot \left( \mathbf{b}_1^{(\ell)} \right)^\top \right) \cdot \mathbf{W}_2^{(\ell)} + \mathbf{1} \cdot \left( \mathbf{b}_2^{(\ell)} \right)^\top, \tag{15}$$

where $\sigma : \mathbb{R} \to \mathbb{R}$ is applied elementwise.

**Activation regularity for the stability lemmas.** The stability lemmas use that $\sigma$ is Lipschitz (globally or on a bounded interval): ReLU and $\tanh$ are globally 1-Lipschitz, sigmoid is globally $(1/4)$-Lipschitz (Lemma A.8), and GELU / SiLU are locally Lipschitz with constants invoked in Section A.1.

The post-LayerNorm encoder-layer update uses LayerNorm from Definition A.5:

$$\hat{\mathbf{Y}}^{(\ell)} \triangleq \mathrm{LayerNorm}\left( \mathbf{Y}^{(\ell-1)} + \mathrm{MHSA}^{(\ell)}\left( \mathbf{Y}^{(\ell-1)} \right) \right), \qquad \mathbf{Y}^{(\ell)} \triangleq \mathrm{LayerNorm}\left( \hat{\mathbf{Y}}^{(\ell)} + \mathrm{FFN}^{(\ell)}(\hat{\mathbf{Y}}^{(\ell)}) \right). \tag{16}$$

The mixer output is

$$\mathrm{Mixer}_{L_{\mathrm{mix}}}(\mathbf{Y}^{(0)}) \triangleq \mathbf{Y}^{(L_{\mathrm{mix}})}.$$

$\triangle$

## A.1. Lipschitz Facts for $\sigma$

**Lemma A.8** (Activation Lipschitz constants)**.** *A differentiable $\sigma : \mathbb{R} \to \mathbb{R}$ with $\sup_{x \in \mathcal{I}} |\sigma'(x)| \leq L_\sigma(\mathcal{I})$ is $L_\sigma(\mathcal{I})$-Lipschitz on $\mathcal{I}$ by the mean-value theorem; smooth activations such as GELU and SiLU are therefore Lipschitz on bounded intervals. In particular ReLU and $\tanh$ are 1-Lipschitz and sigmoid is $\frac{1}{4}$-Lipschitz on $\mathbb{R}$ (Goodfellow et al., 2016). A scalar $L_\sigma$-Lipschitz map applied elementwise to $\mathbb{R}^{M \times d}$ is $L_\sigma$-Lipschitz in $\| \cdot \|_{2,\infty}$ (coordinatewise bound, then maximum over rows).* $\square$

## B. The PLASH Algorithm

Algorithm 1 presents the PLASH block in the $(\mathbf{Q}, \mathbf{K}, \mathbf{V})$ interface, implementing the three-stage pipeline: Stage I compresses $(\mathbf{K}, \mathbf{V})$ from length $N_k$ to $M \ll N_k$ via key-based routing; Stage II enriches each compressed item with selective higher-order feature sketches, a linear readout, and a short-sequence mixer across the $M$ items; Stage III projects to $(\mathbf{K}_g, \mathbf{V}_g)$ and performs the exact scaled dot-product softmax readout.

**Lemma B.1** (Row-stochastic routing matrix)**.** *Let $\mathbf{A} \triangleq \mathrm{softmax}(\mathbf{S}/\tau)$ be defined row-wise as in Algorithm 1. Then each row of $\mathbf{A}$ is a probability vector: for all $i \in [N_k]$, we have $\mathbf{A}_{i,j} \geq 0$ for all $j \in [M]$ and $\sum_{j=1}^M \mathbf{A}_{i,j} = 1$.*

*Proof.* For any row $i$, by the definition of softmax, $\mathbf{A}_{i,j} = \exp(\mathbf{S}_{i,j}/\tau)/\left(\sum_{\ell=1}^{M} \exp(\mathbf{S}_{i,\ell}/\tau)\right)$. Exponentials are nonnegative, so $\mathbf{A}_{i,j} \geq 0$. Summing over $j$ yields $\sum_{j=1}^{M} \mathbf{A}_{i,j} = \sum_{j=1}^{M} \exp(\mathbf{S}_{i,j}/\tau)/\left(\sum_{\ell=1}^{M} \exp(\mathbf{S}_{i,\ell}/\tau)\right) = 1$. $\qquad\square$

**Lemma B.2** (Deterministic norm control in Stage II). *In Algorithm 1, for every $j \in [M]$,*

$$\widetilde{\mathbf{G}}_{j,:} = \frac{\mathbf{G}_{j,:}}{\max\{\|\mathbf{G}_{j,:}\|_2, \varepsilon_g\} \cdot \tau_{\mathrm{g}}}.$$

*Then $\|\widetilde{\mathbf{G}}_{j,:}\|_2 \leq 1/\tau_{\mathrm{g}}$. Moreover, if $\|\mathbf{G}_{j,:}\|_2 \geq \varepsilon_g$, then $\|\widetilde{\mathbf{G}}_{j,:}\|_2 = 1/\tau_{\mathrm{g}}$.*

*Proof.* With $c_j \triangleq \max\{\|\mathbf{G}_{j,:}\|_2, \varepsilon_g\} \cdot \tau_{\mathrm{g}} > 0$,

$$\|\widetilde{\mathbf{G}}_{j,:}\|_2 = \frac{\|\mathbf{G}_{j,:}\|_2}{\max\{\|\mathbf{G}_{j,:}\|_2, \varepsilon_g\} \cdot \tau_{\mathrm{g}}} \leq \frac{1}{\tau_{\mathrm{g}}}.$$

If $\|\mathbf{G}_{j,:}\|_2 \geq \varepsilon_g$, then $\max\{\|\mathbf{G}_{j,:}\|_2, \varepsilon_g\} = \|\mathbf{G}_{j,:}\|_2$, so the fraction equals $1/\tau_{\mathrm{g}}$. $\qquad\square$

**Proposition B.3** (Randomness localization in Algorithm 1). *With the parameters of Algorithm 1 fixed ($\mathbf{P}$, $\tau$, $\tau_{\mathrm{g}}$, $\varepsilon_g$, $\psi$, $\mathbf{W}_{\mathrm{out}}$, $\mathrm{Mixer}_{L_{\mathrm{mix}}}$, $\mathbf{W}_K$, $\mathbf{W}_V$), the TensorSketch seeds $\{(h_t^{(k)}, s_t^{(k)})\}_{k \in \mathcal{K}, t \in [k]}$ are by construction the only sampled objects. Then every intermediate quantity produced by Stages I and III, and by the mixer at Stage II, is a deterministic function of $(\mathbf{Q}, \mathbf{K}, \mathbf{V})$ and the fixed parameters. The only random intermediate variables are the Stage II sketch outputs $\{\mathbf{z}_k\}_{k \in \mathcal{K}}$ (and hence $\{\mathbf{v}_j\}_{j \in [M]}$ and $\mathbf{Y}_{\mathrm{enh}}$) generated by $\mathrm{TensorSketch}_k(\cdot)$.*

*Proof.* Under Assumption C.2, the hash / sign seeds across distinct $(k, t)$ pairs are mutually independent and form the entire random component.

**Stage I.** The matrix product $\mathbf{S} = \mathbf{K} \cdot \mathbf{P}^\top$ is deterministic. Applying softmax row-wise with a fixed temperature $\tau$ is deterministic, so $\mathbf{A}$ is deterministic. Matrix multiplications $\widetilde{\mathbf{K}} = \mathbf{A}^\top \cdot \mathbf{K}$ and $\widetilde{\mathbf{V}} = \mathbf{A}^\top \cdot \mathbf{V}$ are deterministic.

**Stage II (before sketching).** The concatenation $\mathbf{U} = \widetilde{\mathbf{K}} \bigoplus \widetilde{\mathbf{V}}$, the row-wise pre-map $\psi$, and the normalization $\widetilde{\mathbf{G}}_{j,:}$ are all deterministic given $\varepsilon_g, \tau_{\mathrm{g}}$.

**Stage II (sketching).** The seeds $\{(h_t^{(k)}, s_t^{(k)})\}_{k \in \mathcal{K}, t \in [k]}$ are referenced only in the call $\mathbf{z}_k \leftarrow \mathrm{TensorSketch}_k(\widetilde{\mathbf{G}}_{j,:}; D_k, \cdot)$. The downstream concatenation $\mathbf{v}_j \leftarrow \mathbf{v}_j \bigoplus(\beta_k \cdot \mathbf{z}_k)$ and linear map $(\mathbf{Y}_{\mathrm{enh}})_{j,:} \leftarrow \mathbf{W}_{\mathrm{out}} \cdot \mathbf{v}_j$ are deterministic given $\mathbf{z}_k$.

**The mixer at Stage II and Stage III.** The mixer, the projections $\mathbf{K}_g = \mathbf{Z}_{\mathrm{out}} \cdot \mathbf{W}_K$ and $\mathbf{V}_g = \mathbf{Z}_{\mathrm{out}} \cdot \mathbf{W}_V$, and the readout $\mathbf{Y} = \mathrm{softmax}(\mathbf{Q} \cdot \mathbf{K}_g^\top / \sqrt{d_k}) \cdot \mathbf{V}_g$ are all deterministic conditional on the Stage II sketches. $\qquad\square$

## C. TensorSketch (Pham & Pagh, 2013; Avron et al., 2014; Ahle et al., 2020): Construction and Kernel-Estimation Facts

The target is the degree-$k$ polynomial feature map: for $\mathbf{g} \in \mathbb{R}^{d'}$, the exact lift $\mathrm{vec}(\mathbf{g}^{\otimes k}) \in \mathbb{R}^{(d')^k}$ induces the kernel $\langle \mathbf{g}, \mathbf{g}' \rangle^k$, where $\mathbf{g}^{\otimes k}$ is the $k$-fold Kronecker product and $\mathrm{vec}(\cdot)$ denotes column-major vectorization. TensorSketch produces a $D_k$-dimensional sketch whose inner product is an unbiased estimator of $\langle \mathbf{g}, \mathbf{g}' \rangle^k$, with distortion decreasing in $D_k$ (Charikar et al., 2002; Pham & Pagh, 2013).

These definitions apply to generic $\mathbf{g} \in \mathbb{R}^{d'}$; in PLASH, $\mathbf{g} = \widetilde{\mathbf{G}}_{j,:}$ from (7) with $\|\mathbf{g}\|_2$ uniformly bounded.

**Definition C.1** (CountSketch and TensorSketch (explicit maps)). Let $d', D \in \mathbb{Z}_{>0}$, and write $[D] \triangleq \{0, 1, \ldots, D-1\}$ and $[d'] \triangleq \{1, 2, \ldots, d'\}$. Let $h : [d'] \to [D]$ and $s : [d'] \to \{\pm 1\}$ be random hash functions. As in (Charikar et al., 2002; Pham & Pagh, 2013), we assume $h$ is 2-wise independent and uniform on $[D]$, and $s$ is 2-wise independent and uniform on $\{\pm 1\}$.

The CountSketch matrix $\mathbf{C} \in \mathbb{R}^{D \times d'}$ is defined entrywise by

$$\mathbf{C}_{b,i} \triangleq s(i) \cdot \mathbf{1}\{h(i) = b\}, \qquad b \in [D], \ i \in [d']. \tag{17}$$

---

**Algorithm 1** PLASH block in $(\mathbf{Q}, \mathbf{K}, \mathbf{V})$

---

1: **Input:** $\mathbf{Q} \in \mathbb{R}^{N_q \times d_k}$, $\mathbf{K} \in \mathbb{R}^{N_k \times d_k}$, $\mathbf{V} \in \mathbb{R}^{N_k \times d_v}$.
2: **Input:** length $M \ll N_k$; prototypes $\mathbf{P} \in \mathbb{R}^{M \times d_k}$; temperatures $\tau, \tau_{\mathrm{g}} > 0$; stability $\varepsilon_g > 0$.
3: **Input:** pre-map $\psi : \mathbb{R}^{d_k + d_v} \to \mathbb{R}^{d'}$ (row-wise).
4: **Input:** degrees $\mathcal{K} \subset \mathbb{Z}_{\geq 1}$; weights $\{\beta_k\}_{k \in \mathcal{K}}$; sketch sizes $\{D_k\}_{k \in \mathcal{K}}$; fixed TensorSketch seeds $\{(h_t^{(k)}, s_t^{(k)})\}_{k \in \mathcal{K},\, t \in [k]}$.
5: **Input:** $\mathbf{W}_{\mathrm{out}} \in \mathbb{R}^{d \times D_{\mathrm{tot}}}$ with $D_{\mathrm{tot}} = \sum_{k \in \mathcal{K}} D_k$; mixer $\mathrm{Mixer}_{L_{\mathrm{mix}}}$.
6: **Input:** projections $\mathbf{W}_K \in \mathbb{R}^{d \times d_k}$, $\mathbf{W}_V \in \mathbb{R}^{d \times d_v}$.
7: **Output:** $\mathbf{Y} \in \mathbb{R}^{N_q \times d_v}$.
   **Stage I: KV Compression**
9: $\mathbf{S} \leftarrow \mathbf{K} \cdot \mathbf{P}^\top$                                                     // $\mathbf{S} \in \mathbb{R}^{N_k \times M}$
10: $\mathbf{A} \leftarrow \mathrm{softmax}(\mathbf{S}/\tau)$                                           // row-wise over $M$
11: $\widetilde{\mathbf{K}} \leftarrow \mathbf{A}^\top \cdot \mathbf{K}; \quad \widetilde{\mathbf{V}} \leftarrow \mathbf{A}^\top \cdot \mathbf{V}$        // both length $M$
   **Stage II: Randomized Higher-Order Feature Lifting**
13: $\mathbf{U} \leftarrow \widetilde{\mathbf{K}} \bigoplus \widetilde{\mathbf{V}}$                          // row-wise concat, $\mathbf{U} \in \mathbb{R}^{M \times (d_k + d_v)}$
14: $\mathbf{G} \leftarrow \psi(\mathbf{U})$                                                    // $\mathbf{G} \in \mathbb{R}^{M \times d'}$
15: **for** $j = 1$ **to** $M$ **do**
16: $\quad \widetilde{\mathbf{G}}_{j,:} \leftarrow \mathbf{G}_{j,:} / \big(\max\{\|\mathbf{G}_{j,:}\|_2, \varepsilon_g\} \cdot \tau_{\mathrm{g}}\big)$   // stabilized norm control
17: $\quad$ **for each** $k \in \mathcal{K}$ **do**
18: $\quad\quad \mathbf{z}_k \leftarrow \mathrm{TensorSketch}_k(\widetilde{\mathbf{G}}_{j,:}; D_k, \{(h_t^{(k)}, s_t^{(k)})\}_{t=1}^k) \in \mathbb{R}^{D_k}$
19: $\quad\quad \mathbf{v}_j \leftarrow \mathbf{v}_j \bigoplus (\beta_k \cdot \mathbf{z}_k)$
20: $\quad$ **end for**
21: $\quad (\mathbf{Y}_{\mathrm{enh}})_{j,:} \leftarrow \mathbf{W}_{\mathrm{out}} \cdot \mathbf{v}_j$                         // $\mathbf{Y}_{\mathrm{enh}} \in \mathbb{R}^{M \times d}$
22: **end for**
23: $\mathbf{Z}_{\mathrm{out}} \leftarrow \mathrm{Mixer}_{L_{\mathrm{mix}}}(\mathbf{Y}_{\mathrm{enh}})$                            // $\mathbf{Z}_{\mathrm{out}} \in \mathbb{R}^{M \times d}$
   **Stage III: Final Output**
25: $\mathbf{K}_g \leftarrow \mathbf{Z}_{\mathrm{out}} \cdot \mathbf{W}_K; \quad \mathbf{V}_g \leftarrow \mathbf{Z}_{\mathrm{out}} \cdot \mathbf{W}_V$   // $\mathbf{K}_g \in \mathbb{R}^{M \times d_k}$, $\mathbf{V}_g \in \mathbb{R}^{M \times d_v}$
26: $\mathbf{Y} \leftarrow \mathrm{softmax}\big(\mathbf{Q} \cdot \mathbf{K}_g^\top / \sqrt{d_k}\big) \cdot \mathbf{V}_g$             // exact attention
27: **return** $\mathbf{Y}$

---

For $\mathbf{g} \in \mathbb{R}^{d'}$, its CountSketch is $\mathbf{c}(\mathbf{g}) \triangleq \mathbf{C} \cdot \mathbf{g} \in \mathbb{R}^D$, i.e.,

$$\mathbf{c}(\mathbf{g})_b = \sum_{i:\, h(i)=b} s(i) \cdot g_i, \qquad b \in [D]. \tag{18}$$

Let $k, D_k \in \mathbb{Z}_{>0}$. For each $t \in \{1, \ldots, k\}$, we sample an independent pair $h_t : [d'] \to [D_k]$ and $s_t : [d'] \to \{\pm 1\}$ with the same assumptions, and let $\mathbf{C}_t$ be the associated CountSketch matrix. We define $\mathbf{c}_t(\mathbf{g}) \triangleq \mathbf{C}_t \cdot \mathbf{g} \in \mathbb{R}^{D_k}$. TensorSketch is the $k$-fold circular convolution of these sketches (Pham & Pagh, 2013; Avron et al., 2014; Ahle et al., 2020):

$$\mathrm{TS}_k(\mathbf{g}; D_k) \triangleq \mathbf{c}_1(\mathbf{g}) * \mathbf{c}_2(\mathbf{g}) * \cdots * \mathbf{c}_k(\mathbf{g}) \in \mathbb{R}^{D_k}, \tag{19}$$

$$\mathrm{TensorSketch}_k(\mathbf{g}; D_k, \{(h_t, s_t)\}_{t=1}^k) \triangleq \mathrm{TS}_k(\mathbf{g}; D_k) \in \mathbb{R}^{D_k}, \tag{20}$$

where $*$ denotes circular convolution on $[D_k]$. Using the FFT convolution identity (Cooley & Tukey, 1965),

$$\mathrm{TS}_k(\mathbf{g}; D_k) = \mathrm{IFFT}\big(\mathrm{FFT}(\mathbf{c}_1(\mathbf{g})) \odot \cdots \odot \mathrm{FFT}(\mathbf{c}_k(\mathbf{g}))\big). \tag{21}$$

TensorSketch can also be viewed as a CountSketch applied to the degree-$k$ tensor lifting. We define the combined hash / sign on $[d']^k$ by

$$H(i_1, \ldots, i_k) \triangleq \left(\sum_{t=1}^k h_t(i_t)\right) \bmod D_k, \tag{22}$$

$$S(i_1, \ldots, i_k) \triangleq \prod_{t=1}^k s_t(i_t). \tag{23}$$

Let $\mathbf{C}^{(\otimes k)} \in \mathbb{R}^{D_k \times (d')^k}$ be the corresponding CountSketch matrix on tensor indices,

$$\mathbf{C}^{(\otimes k)}_{b,(i_1,\ldots,i_k)} \triangleq S(i_1,\ldots,i_k) \cdot \mathbf{1}\{H(i_1,\ldots,i_k) = b\}. \tag{24}$$

The convolution construction yields the identity (Pham & Pagh, 2013):

$$\mathrm{TS}_k(\mathbf{g}; D_k) = \mathbf{C}^{(\otimes k)} \cdot \mathrm{vec}(\mathbf{g}^{\otimes k}). \tag{25}$$

$\triangle$

By (25), the implemented feature is a randomized linear projection of $\mathrm{vec}(\mathbf{g}^{\otimes k})$, so it admits the exact deterministic comparator analyzed in Appendix G.

**Assumption C.2** (TensorSketch randomness with limited independence)**.** Let $\mathcal{K} \subset \mathbb{Z}_{\geq 1}$ be a finite degree set, with sketch sizes $\{D_k\}_{k \in \mathcal{K}} \subset \mathbb{Z}_{>0}$ and feature width $d' \in \mathbb{Z}_{>0}$. For each $k \in \mathcal{K}$ and each factor index $t \in [k]$, TensorSketch uses hash / sign maps

$$h_t^{(k)} : [d'] \to [D_k], \qquad s_t^{(k)} : [d'] \to \{\pm 1\},$$

such that:

1. $h_t^{(k)}$ is uniform on $[D_k]$ and 3-wise independent;
2. $s_t^{(k)}$ is uniform on $\{\pm 1\}$ and 4-wise independent;
3. the family $\{(h_t^{(k)}, s_t^{(k)})\}_{k \in \mathcal{K}, \, t \in [k]}$ is mutually independent across all pairs $(k,t)$.

This regime suffices for the second-moment calculation in Proposition C.4 (Pham & Pagh, 2013; Avron et al., 2014). $\triangle$

**Construction C.3** (A checkable instantiation of Assumption C.2)**.** Let $k \in \mathcal{K}$ and let $D_k \in \mathbb{Z}_{>0}$ be the target sketch length.

**Step 1 (3-wise uniform hashing).** We choose a prime $p_k \geq \max\{d', D_k\}$, identify $[d'] \subset \mathbb{F}_{p_k}$, sample $a_0, a_1, a_2 \overset{\text{i.i.d.}}{\sim} \mathrm{Unif}(\mathbb{F}_{p_k})$, and define

$$\tilde{h}(x) = a_0 + a_1 x + a_2 x^2 \pmod{p_k}, \qquad x \in \mathbb{F}_{p_k}.$$

If $D_k = p_k$, then $h(i) \triangleq \tilde{h}(i) \in [D_k]$. If $D_k < p_k$, then $h(i) \triangleq \tilde{h}(i) \bmod D_k$. As a coordinate-wise deterministic function of $\tilde{h}$, the hash $h$ is 3-wise independent: any three values $h(i_1), h(i_2), h(i_3)$ are images of the jointly independent $\tilde{h}(i_1), \tilde{h}(i_2), \tilde{h}(i_3)$. The reduction is uniform on $[D_k]$ up to total-variation distance at most $D_k/p_k$, which is negligible for $p_k \gg D_k$ and vanishes when $D_k \mid p_k$; this is the standard polynomial-hashing realization of the 3-wise uniform family of Assumption C.2 (Stinson, 1994).

**Step 2 (4-wise independent Rademacher signs).** We work over $\mathbb{F}_{2^w}$ with $2^w \geq d'$, identify $[d'] \subset \mathbb{F}_{2^w}$, sample $b_0, b_1, b_2, b_3 \overset{\text{i.i.d.}}{\sim} \mathrm{Unif}(\mathbb{F}_{2^w})$, define

$$\tilde{s}(x) = b_0 + b_1 x + b_2 x^2 + b_3 x^3 \in \mathbb{F}_{2^w},$$

fix any nonzero $\mathbb{F}_2$-linear functional $\ell : \mathbb{F}_{2^w} \to \mathbb{F}_2$, and set

$$s(i) \triangleq (-1)^{\ell(\tilde{s}(i))} \in \{\pm 1\}.$$

Then $s$ is uniform on $\{\pm 1\}$ and 4-wise independent.

**Step 3 (independence across $(k,t)$).** We instantiate $(h_t^{(k)}, s_t^{(k)})$ independently for each $(k,t)$.

Degree-$d$ random polynomials over a field yield $(d+1)$-wise independence by interpolation; the modular reduction $\mathbb{F}_{p_k} \to [D_k]$ preserves this independence and is uniform up to a negligible $O(D_k/p_k)$ bias, and the sign construction is the analogous $\mathbb{F}_{2^w}$ argument (Stinson, 1994). $\triangle$

**Proposition C.4** (Inner-product estimation for the degree-$k$ polynomial kernel)**.** *Let $k, D, d' \in \mathbb{Z}_{>0}$, and let $\mathrm{TS}_k(\cdot; D)$ be as in Definition C.1. We define*

$$\widehat{\kappa}_k(\mathbf{x}, \mathbf{y}) \triangleq \langle \mathrm{TS}_k(\mathbf{x}; D), \mathrm{TS}_k(\mathbf{y}; D) \rangle.$$

*Under Assumption C.2, for any $\mathbf{x}, \mathbf{y} \in \mathbb{R}^{d'}$,*

$$\mathbb{E}(\widehat{\kappa}_k(\mathbf{x}, \mathbf{y})) = \langle \mathbf{x}, \mathbf{y} \rangle^k, \tag{26}$$

*and*

$$\mathrm{Var}\big(\widehat{\kappa}_k(\mathbf{x}, \mathbf{y})\big) \ \leq \ \frac{C_k}{D}\Big(\langle \mathbf{x}, \mathbf{y}\rangle^{2k} + \|\mathbf{x}\|_2^{2k}\|\mathbf{y}\|_2^{2k}\Big), \tag{27}$$

*where $C_k \geq 1$ is a constant depending only on the degree $k$ (the degree-$k$ TensorSketch second-moment constant of Pham & Pagh, 2013; Avron et al., 2014); for $k = 1$ (CountSketch) one has $C_1 = 1$ (Charikar et al., 2002).*

*Proof.* By (25), $\widehat{\kappa}_k(\mathbf{x}, \mathbf{y}) = \langle \mathbf{C}^{(\otimes k)} \cdot \mathbf{u}, \mathbf{C}^{(\otimes k)} \cdot \mathbf{v}\rangle$ with $\mathbf{u} = \mathrm{vec}(\mathbf{x}^{\otimes k})$ and $\mathbf{v} = \mathrm{vec}(\mathbf{y}^{\otimes k})$, where $\mathbf{C}^{(\otimes k)}$ is the degree-$k$ tensor CountSketch of (24). Unbiasedness uses only 2-wise sign independence: the off-diagonal terms $\sum_{p \neq q} s(p)s(q)u_p v_q \mathbf{1}\{h(p) = h(q)\}$ vanish in expectation since $\mathbb{E}(s(p)s(q)) = 0$, so $\mathbb{E}(\widehat{\kappa}_k) = \langle \mathbf{u}, \mathbf{v}\rangle$; with the tensor identities $\langle \mathbf{u}, \mathbf{v}\rangle = \langle \mathbf{x}, \mathbf{y}\rangle^k$, $\|\mathbf{u}\|_2^2 = \|\mathbf{x}\|_2^{2k}$, and $\|\mathbf{v}\|_2^2 = \|\mathbf{y}\|_2^{2k}$ this gives (26). For the variance, the combined hash $H(i_1, \ldots, i_k) = \big(\sum_t h_t(i_t)\big) \bmod D$ and sign $S(i_1, \ldots, i_k) = \prod_t s_t(i_t)$ are correlated across tensor multi-indices that share base coordinates, so $\mathbf{C}^{(\otimes k)}$ is not a fully independent CountSketch on $[d']^k$. The degree-$k$ second-moment analysis of Pham & Pagh (2013); Avron et al. (2014), under the 3-wise hash and 4-wise sign independence of Construction C.3, gives $\mathrm{Var}(\widehat{\kappa}_k) \leq \frac{C_k}{D}\big(\|\mathbf{u}\|_2^2\|\mathbf{v}\|_2^2 + \langle \mathbf{u}, \mathbf{v}\rangle^2\big)$ for a constant $C_k$ depending only on $k$, with $C_1 = 1$ recovering the CountSketch second moment (Charikar et al., 2002). Substituting the tensor identities yields (27). $\square$

**Corollary C.5** (Uniform variance control under norm bounds)**.** *If $\|\mathbf{x}\|_2 \leq R$ and $\|\mathbf{y}\|_2 \leq R$, then*

$$\mathrm{Var}\big(\widehat{\kappa}_k(\mathbf{x}, \mathbf{y})\big) \ \leq \ \frac{2C_k R^{4k}}{D}.$$

*In particular, if $\|\widetilde{\mathbf{G}}_{j,:}\|_2 \leq 1/\tau_{\mathrm{g}}$ for all $j \in [M]$, then the degree-dependent factor in (27) is uniformly controlled over all compressed items.*

*Proof.* By Cauchy–Schwarz, $|\langle \mathbf{x}, \mathbf{y}\rangle| \leq \|\mathbf{x}\|_2\|\mathbf{y}\|_2 \leq R^2$, hence $\langle \mathbf{x}, \mathbf{y}\rangle^{2k} \leq R^{4k}$ and $\|\mathbf{x}\|_2^{2k}\|\mathbf{y}\|_2^{2k} \leq R^{4k}$. Substituting these bounds into (27), we obtain $\mathrm{Var}(\widehat{\kappa}_k(\mathbf{x}, \mathbf{y})) \leq \frac{C_k}{D}(R^{4k} + R^{4k}) = 2C_k R^{4k}/D$. The final statement follows by taking $R = 1/\tau_{\mathrm{g}}$. $\square$

**Degree dependence of $C_k$.** The constant $C_k$ depends only on the degree $k$ and reflects the degree-$k$ TensorSketch second moment, which grows with $k$, a known property of convolution-based polynomial sketches that motivates the improved high-degree constructions of Ahle et al. (2020). PLASH uses $\mathcal{K} \subseteq \{1, 2\}$ in all experiments, so the relevant constants are small, with $C_1 = 1$.

## D. Local Lipschitzness of the KV Mixer in $\|\cdot\|_{2,\infty}$

We consider the post-LayerNorm Transformer-encoder mixer $\mathrm{Mixer}_{L_{\mathrm{mix}}}$ from Definition A.7 under the matrix norm

$$\|\mathbf{U}\|_{2,\infty} \triangleq \max_{i \in [M]} \|\mathbf{U}_{i,:}\|_2,$$

the maximum row-wise $\ell_2$ magnitude. The Stage II certificate controls per-row perturbations in this norm, and the mixer is deterministic given inputs and parameters.

Statements in this section assume inference mode (dropout disabled); under training-mode dropout they hold conditional on a fixed dropout mask.

The comparison between the realized Stage II embedding $\mathbf{Y}_{\mathrm{enh}}$ and its deterministic comparator $\mathbf{Y}_{\mathrm{enh}}^{\mathrm{det}}$ requires controlling the mixer on the line segment

$$\mathcal{S} = \Big\{\mathbf{Y}_{\mathrm{enh}}^{\mathrm{det}} + t \cdot \big(\mathbf{Y}_{\mathrm{enh}} - \mathbf{Y}_{\mathrm{enh}}^{\mathrm{det}}\big) : t \in [0, 1]\Big\} \ \subseteq \ \mathbb{R}^{M \times d},$$

which is the same as (70).

**Lemma D.1** (Right multiplication is Lipschitz in $\|\cdot\|_{2,\infty}$)**.** *For any compatible matrices $\mathbf{X}, \mathbf{X}'$ and any matrix $\mathbf{W}$,*

$$\big\|\mathbf{X} \cdot \mathbf{W} - \mathbf{X}' \cdot \mathbf{W}\big\|_{2,\infty} \ \leq \ \|\mathbf{W}\|_{\mathrm{op}} \cdot \|\mathbf{X} - \mathbf{X}'\|_{2,\infty},$$

*where $\|\mathbf{W}\|_{\mathrm{op}}$ is defined in Definition A.3.*

*Proof.* For each $i \in [M]$,

$$\left\|(\mathbf{X} \cdot \mathbf{W} - \mathbf{X}' \cdot \mathbf{W})_{i,:}\right\|_2 = \left\|(\mathbf{X}_{i,:} - \mathbf{X}'_{i,:}) \cdot \mathbf{W}\right\|_2 \leq \|\mathbf{W}\|_{\mathrm{op}} \cdot \|\mathbf{X}_{i,:} - \mathbf{X}'_{i,:}\|_2.$$

Taking the maximum over $i \in [M]$ yields the claim. $\qquad\square$

**Lemma D.2** (Softmax is 1-Lipschitz from $\ell_\infty$ to $\ell_1$). *For any* $\mathbf{s}, \mathbf{s}' \in \mathbb{R}^N$,

$$\| \operatorname{softmax}(\mathbf{s}) - \operatorname{softmax}(\mathbf{s}') \|_1 \leq \|\mathbf{s} - \mathbf{s}'\|_\infty. \tag{28}$$

*Proof.* The softmax Jacobian $\mathbf{J} = \operatorname{Diag}(\mathbf{a}) - \mathbf{a} \cdot \mathbf{a}^\top$ at any point satisfies $\|\mathbf{J}\|_{\infty \to 1} \leq 1$ (Gao & Pavel, 2017; Gao et al., 2016); integrating along the segment $\mathbf{s}' + t(\mathbf{s} - \mathbf{s}')$, $t \in [0,1]$, gives (28). $\qquad\square$

**Lemma D.3** (Scaled dot-product attention is Lipschitz in each argument under $\|\cdot\|_{2,\infty}$). *Scaled dot-product attention is*

$$\operatorname{Atten}(\mathbf{Q}; \mathbf{K}, \mathbf{V}) = \operatorname{softmax}\left(\tfrac{1}{\sqrt{d_k}} \cdot \mathbf{Q} \cdot \mathbf{K}^\top\right) \cdot \mathbf{V} \in \mathbb{R}^{N_q \times d_v}.$$

*Given* $\mathbf{Q} \in \mathbb{R}^{N_q \times d_k}$, $\mathbf{V}, \mathbf{V}' \in \mathbb{R}^{N \times d_v}$, *and* $\mathbf{K}, \mathbf{K}' \in \mathbb{R}^{N \times d_k}$, *we define*

$$\Gamma_Q \triangleq \frac{\|\mathbf{Q}\|_{2,\infty}}{\sqrt{d_k}}, \qquad \Gamma_V \triangleq \max\{\|\mathbf{V}\|_{2,\infty}, \|\mathbf{V}'\|_{2,\infty}\}.$$

*Then the following deterministic bounds hold:*

$$\left\|\operatorname{Atten}(\mathbf{Q}; \mathbf{K}, \mathbf{V}) - \operatorname{Atten}(\mathbf{Q}; \mathbf{K}, \mathbf{V}')\right\|_{2,\infty} \leq \|\mathbf{V} - \mathbf{V}'\|_{2,\infty},$$
$$\left\|\operatorname{Atten}(\mathbf{Q}; \mathbf{K}, \mathbf{V}) - \operatorname{Atten}(\mathbf{Q}; \mathbf{K}', \mathbf{V})\right\|_{2,\infty} \leq \Gamma_Q \cdot \Gamma_V \cdot \|\mathbf{K} - \mathbf{K}'\|_{2,\infty}.$$

*Consequently,*

$$\left\|\operatorname{Atten}(\mathbf{Q}; \mathbf{K}, \mathbf{V}) - \operatorname{Atten}(\mathbf{Q}; \mathbf{K}', \mathbf{V}')\right\|_{2,\infty} \leq \Gamma_Q \cdot \Gamma_V \cdot \|\mathbf{K} - \mathbf{K}'\|_{2,\infty} + \|\mathbf{V} - \mathbf{V}'\|_{2,\infty}.$$

*Proof.* We write

$$\mathbf{S} = \tfrac{1}{\sqrt{d_k}} \cdot \mathbf{Q} \cdot \mathbf{K}^\top \in \mathbb{R}^{N_q \times N}, \qquad \mathbf{A} = \operatorname{softmax}(\mathbf{S}) \in \mathbb{R}^{N_q \times N},$$

and analogously $\mathbf{S}'$ and $\mathbf{A}'$ for $\mathbf{K}'$. Each row of $\mathbf{A}$ and $\mathbf{A}'$ lies in the probability simplex (nonnegative entries summing to 1).

*Step 1: Lipschitz in $\mathbf{V}$ (with $\mathbf{K}$ fixed).* For any $i \in [N_q]$, since $\mathbf{A}_{i,:}$ is a convex weight vector,

$$\begin{aligned}
\|(\mathbf{A} \cdot \mathbf{V} - \mathbf{A} \cdot \mathbf{V}')_{i,:}\|_2 &= \left\|\sum_{j=1}^N \mathbf{A}_{i,j} \cdot (\mathbf{V}_{j,:} - \mathbf{V}'_{j,:})\right\|_2 \\
&\leq \sum_{j=1}^N \mathbf{A}_{i,j} \cdot \|\mathbf{V}_{j,:} - \mathbf{V}'_{j,:}\|_2 \\
&\leq \|\mathbf{V} - \mathbf{V}'\|_{2,\infty}.
\end{aligned}$$

Taking the maximum over $i \in [N_q]$ gives the first bound.

*Step 2: Lipschitz in $\mathbf{K}$ (with $\mathbf{V}$ fixed).* For any $i \in [N_q]$, let $\mathbf{s} = \mathbf{S}_{i,:}$ and $\mathbf{s}' = \mathbf{S}'_{i,:}$. For any $j \in [N]$,

$$|\mathbf{s}_j - \mathbf{s}'_j| = \left|\tfrac{1}{\sqrt{d_k}}\langle \mathbf{Q}_{i,:}, \mathbf{K}_{j,:} - \mathbf{K}'_{j,:}\rangle\right| \leq \tfrac{1}{\sqrt{d_k}}\|\mathbf{Q}_{i,:}\|_2 \cdot \|\mathbf{K}_{j,:} - \mathbf{K}'_{j,:}\|_2,$$

so

$$\|\mathbf{s} - \mathbf{s}'\|_\infty \leq \Gamma_Q \cdot \|\mathbf{K} - \mathbf{K}'\|_{2,\infty}.$$

Using $\mathbf{A} \cdot \mathbf{V} - \mathbf{A}' \cdot \mathbf{V} = (\mathbf{A} - \mathbf{A}') \cdot \mathbf{V}$, Lemma D.2, and $\|\mathbf{V}\|_{2,\infty} \leq \Gamma_V$,

$$
\begin{aligned}
\left\| \left( (\mathbf{A} - \mathbf{A}') \cdot \mathbf{V} \right)_{i,:} \right\|_2 &\leq \|\mathbf{A}_{i,:} - \mathbf{A}'_{i,:}\|_1 \cdot \|\mathbf{V}\|_{2,\infty} \\
&\leq \|\mathbf{s} - \mathbf{s}'\|_\infty \cdot \Gamma_V \\
&\leq \Gamma_Q \cdot \Gamma_V \cdot \|\mathbf{K} - \mathbf{K}'\|_{2,\infty}.
\end{aligned}
$$

Taking the maximum over $i \in [N_q]$ yields the second bound. The joint bound follows by triangle inequality. $\qquad\square$

**Lemma D.4** (Row-wise LayerNorm is locally Lipschitz on a bounded-variance set). *Let* LayerNorm *be as in Definition A.5 with $\varepsilon_{\ln} > 0$ and trainable $\boldsymbol{\gamma}, \boldsymbol{\beta} \in \mathbb{R}^d$. For a fixed set $\mathcal{T} \subseteq \mathbb{R}^d$, we define*

$$
m_{\mathcal{T}} \triangleq \inf_{\mathbf{u} \in \mathcal{T}} \left( \mathrm{Var}(\mathbf{u}) + \varepsilon_{\ln} \right) \geq \varepsilon_{\ln}.
$$

*Then there exists a deterministic constant $L_{\mathrm{LN}}(\mathcal{T}) > 0$ such that for all $\mathbf{u}, \mathbf{v} \in \mathcal{T}$,*

$$
\|\mathrm{LayerNorm}(\mathbf{u}) - \mathrm{LayerNorm}(\mathbf{v})\|_2 \leq L_{\mathrm{LN}}(\mathcal{T}) \cdot \|\mathbf{u} - \mathbf{v}\|_2.
$$

*One admissible choice is*

$$
L_{\mathrm{LN}}(\mathcal{T}) \triangleq \|\boldsymbol{\gamma}\|_\infty \cdot \left( \frac{2}{\sqrt{m_{\mathcal{T}}}} + \frac{2}{d} \cdot \frac{\left( \sup_{\mathbf{u} \in \mathcal{T}} \|\mathbf{u} - \mu(\mathbf{u})\mathbf{1}\|_2 \right)^2}{m_{\mathcal{T}}^{3/2}} \right).
$$

*Consequently, if* LayerNorm *is applied row-wise to matrices and each row lies in $\mathcal{T}$, then*

$$
\|\mathrm{LayerNorm}(\mathbf{U}) - \mathrm{LayerNorm}(\mathbf{V})\|_{2,\infty} \leq L_{\mathrm{LN}}(\mathcal{T}) \cdot \|\mathbf{U} - \mathbf{V}\|_{2,\infty}.
$$

*Proof.* **Step 0: Rewriting LayerNorm.** We define the centering and inverse-stdev maps

$$
\mathbf{c}(\mathbf{u}) \triangleq \mathbf{u} - \mu(\mathbf{u})\mathbf{1}, \qquad \alpha(\mathbf{u}) \triangleq \left( \mathrm{Var}(\mathbf{u}) + \varepsilon_{\ln} \right)^{-1/2}.
$$

Then Definition A.5 becomes

$$
\mathrm{LayerNorm}(\mathbf{u}) = \boldsymbol{\gamma} \odot \left( \alpha(\mathbf{u}) \cdot \mathbf{c}(\mathbf{u}) \right) + \boldsymbol{\beta}.
$$

The bias cancels under subtraction and $\|\boldsymbol{\gamma} \odot \mathbf{x}\|_2 \leq \|\boldsymbol{\gamma}\|_\infty \|\mathbf{x}\|_2$, so it suffices to bound

$$
\left\| \alpha(\mathbf{u}) \cdot \mathbf{c}(\mathbf{u}) - \alpha(\mathbf{v}) \cdot \mathbf{c}(\mathbf{v}) \right\|_2.
$$

**Step 1: Centering is $2$-Lipschitz.** For any $\mathbf{u}, \mathbf{v}$,

$$
\mathbf{c}(\mathbf{u}) - \mathbf{c}(\mathbf{v}) = (\mathbf{u} - \mathbf{v}) - \frac{1}{d}\langle \mathbf{1}, \mathbf{u} - \mathbf{v} \rangle \mathbf{1}.
$$

Hence, using Cauchy–Schwarz and $\|\mathbf{1}\|_2^2 = d$,

$$
\begin{aligned}
\|\mathbf{c}(\mathbf{u}) - \mathbf{c}(\mathbf{v})\|_2 &\leq \|\mathbf{u} - \mathbf{v}\|_2 + \frac{1}{d} \cdot |\langle \mathbf{1}, \mathbf{u} - \mathbf{v} \rangle| \cdot \|\mathbf{1}\|_2 \\
&\leq \|\mathbf{u} - \mathbf{v}\|_2 + \frac{1}{d} \cdot \|\mathbf{1}\|_2^2 \cdot \|\mathbf{u} - \mathbf{v}\|_2 \\
&= 2\|\mathbf{u} - \mathbf{v}\|_2.
\end{aligned}
$$

**Step 2: Decomposing the difference.** Adding and subtracting $\alpha(\mathbf{u}) \cdot \mathbf{c}(\mathbf{v})$ yields

$$
\|\alpha(\mathbf{u}) \cdot \mathbf{c}(\mathbf{u}) - \alpha(\mathbf{v}) \cdot \mathbf{c}(\mathbf{v})\|_2 \leq \alpha(\mathbf{u}) \|\mathbf{c}(\mathbf{u}) - \mathbf{c}(\mathbf{v})\|_2 + |\alpha(\mathbf{u}) - \alpha(\mathbf{v})| \cdot \|\mathbf{c}(\mathbf{v})\|_2. \tag{29}
$$

**Step 3: Bounding the first term using $m_{\mathcal{T}}$.** For any $\mathbf{u} \in \mathcal{T}$, $\alpha(\mathbf{u}) \leq 1/\sqrt{m_{\mathcal{T}}}$ by definition of $m_{\mathcal{T}}$. Combining with Step 1 gives

$$
\alpha(\mathbf{u})\|\mathbf{c}(\mathbf{u}) - \mathbf{c}(\mathbf{v})\|_2 \leq \frac{2}{\sqrt{m_{\mathcal{T}}}}\|\mathbf{u} - \mathbf{v}\|_2.
$$

**Step 4: Bounding $|\alpha(\mathbf{u}) - \alpha(\mathbf{v})|$ via variance differences.** Let $f(x) = x^{-1/2}$ on $[m_{\mathcal{T}}, \infty)$. Then $|f'(x)| = \frac{1}{2}x^{-3/2} \leq \frac{1}{2}m_{\mathcal{T}}^{-3/2}$. By the mean value theorem,

$$|\alpha(\mathbf{u}) - \alpha(\mathbf{v})| \leq \frac{1}{2}m_{\mathcal{T}}^{-3/2} \cdot |\operatorname{Var}(\mathbf{u}) - \operatorname{Var}(\mathbf{v})|. \tag{30}$$

Next, since $\operatorname{Var}(\mathbf{u}) = \frac{1}{d}\|\mathbf{c}(\mathbf{u})\|_2^2$,

$$
\begin{aligned}
|\operatorname{Var}(\mathbf{u}) - \operatorname{Var}(\mathbf{v})| &= \frac{1}{d}\Big|\|\mathbf{c}(\mathbf{u})\|_2^2 - \|\mathbf{c}(\mathbf{v})\|_2^2\Big| \\
&= \frac{1}{d}\Big|\langle \mathbf{c}(\mathbf{u}) + \mathbf{c}(\mathbf{v}),\ \mathbf{c}(\mathbf{u}) - \mathbf{c}(\mathbf{v})\rangle\Big| \\
&\leq \frac{1}{d}\|\mathbf{c}(\mathbf{u}) + \mathbf{c}(\mathbf{v})\|_2 \cdot \|\mathbf{c}(\mathbf{u}) - \mathbf{c}(\mathbf{v})\|_2 \\
&\leq \frac{2}{d}\Big(\sup_{\mathbf{w}\in\mathcal{T}} \|\mathbf{c}(\mathbf{w})\|_2\Big) \cdot 2\|\mathbf{u} - \mathbf{v}\|_2 \\
&= \frac{4}{d}\Big(\sup_{\mathbf{w}\in\mathcal{T}} \|\mathbf{c}(\mathbf{w})\|_2\Big)\|\mathbf{u} - \mathbf{v}\|_2,
\end{aligned}
$$

where we used Step 1 and $\|\mathbf{c}(\mathbf{u}) + \mathbf{c}(\mathbf{v})\|_2 \leq 2\sup_{\mathbf{w}\in\mathcal{T}} \|\mathbf{c}(\mathbf{w})\|_2$. Substituting this into (30) yields

$$|\alpha(\mathbf{u}) - \alpha(\mathbf{v})| \leq \frac{2}{d} \cdot \frac{\sup_{\mathbf{w}\in\mathcal{T}} \|\mathbf{c}(\mathbf{w})\|_2}{m_{\mathcal{T}}^{3/2}} \cdot \|\mathbf{u} - \mathbf{v}\|_2. \tag{31}$$

**Step 5: Bounding the second term and combining.** Since $\|\mathbf{c}(\mathbf{v})\|_2 \leq \sup_{\mathbf{w}\in\mathcal{T}} \|\mathbf{c}(\mathbf{w})\|_2$, combining with (31) gives

$$|\alpha(\mathbf{u}) - \alpha(\mathbf{v})| \cdot \|\mathbf{c}(\mathbf{v})\|_2 \leq \frac{2}{d} \cdot \frac{\big(\sup_{\mathbf{w}\in\mathcal{T}} \|\mathbf{c}(\mathbf{w})\|_2\big)^2}{m_{\mathcal{T}}^{3/2}} \cdot \|\mathbf{u} - \mathbf{v}\|_2.$$

Substituting the Step-3 and Step-5 bounds into (29) and multiplying by $\|\gamma\|_\infty$ yields the claim.

**Row-wise matrix version.** If each row of $\mathbf{U}, \mathbf{V}$ lies in $\mathcal{T}$, applying the vector bound row-wise and taking the maximum over $i$ gives the $\|\cdot\|_{2,\infty}$ form. $\qquad\square$

**Theorem D.5** (Local Lipschitzness of $\operatorname{Mixer}_{L_{\mathrm{mix}}}$ on $\mathcal{S}$ in $\|\cdot\|_{2,\infty}$)**.** *Let $\operatorname{Mixer}_{L_{\mathrm{mix}}}$ be as in Definition A.7, and assume the pointwise nonlinearity $\sigma$ used in $\mathrm{FFN}^{(\ell)}$ is $L_\sigma$-Lipschitz and $\varepsilon_{\mathrm{ln}} > 0$ (see Section A.1). Let $\mathcal{S} \subseteq \mathbb{R}^{M\times d}$ be the segment defined in (70). Then there exists a deterministic constant $L_{\mathrm{mix}}(\mathcal{S}) > 0$ such that for all $\mathbf{U}, \mathbf{U}' \in \mathcal{S}$,*

$$\big\|\operatorname{Mixer}_{L_{\mathrm{mix}}}(\mathbf{U}) - \operatorname{Mixer}_{L_{\mathrm{mix}}}(\mathbf{U}')\big\|_{2,\infty} \leq L_{\mathrm{mix}}(\mathcal{S}) \cdot \|\mathbf{U} - \mathbf{U}'\|_{2,\infty}.$$

*Moreover, one admissible choice is*

$$L_{\mathrm{mix}}(\mathcal{S}) \triangleq \prod_{\ell=1}^{L_{\mathrm{mix}}} L_\ell(\mathcal{S}),$$

*where $L_\ell(\mathcal{S})$ is any Lipschitz constant of the $\ell$th post-LN encoder layer restricted to the corresponding input segment induced by $\mathcal{S}$.*

*Proof.* **Step 0: Compact segments induced by $\mathcal{S}$.** For $\ell \in \{0, 1, \ldots, L_{\mathrm{mix}}\}$, let $\mathbf{Y}^{(\ell)}$ be the $\ell$th-layer hidden state produced by the update equations in Definition A.7 starting from $\mathbf{Y}^{(0)} \in \mathcal{S}$, and define

$$\mathcal{S}_0 \triangleq \mathcal{S}, \qquad \mathcal{S}_\ell \triangleq \{\mathbf{Y}^{(\ell)} :\ \mathbf{Y}^{(0)} \in \mathcal{S}\} \subseteq \mathbb{R}^{M\times d}.$$

$\mathcal{S}_\ell$ is compact as the continuous image of the compact set $\mathcal{S}$.

For a layer index $\ell \in \{1, \ldots, L_{\mathrm{mix}}\}$, we write the $\ell$th post-LN encoder layer as (cf. Definition A.7)

$$\mathbf{H}^{(\ell)}(\mathbf{Y}) \triangleq \underbrace{\operatorname{LayerNorm}\Big(\mathbf{A} + \mathrm{FFN}^{(\ell)}(\mathbf{A})\Big)}_{\mathbf{G}^{(\ell)}(\mathbf{A})}\Big|_{\mathbf{A}=\mathbf{F}^{(\ell)}(\mathbf{Y})}, \qquad \mathbf{F}^{(\ell)}(\mathbf{Y}) \triangleq \operatorname{LayerNorm}\Big(\mathbf{Y} + \mathrm{MHSA}^{(\ell)}(\mathbf{Y})\Big).$$

Then $\mathbf{Y}^{(\ell)} = \mathbf{H}^{(\ell)}(\mathbf{Y}^{(\ell-1)})$.

**Step 1: The map $\mathbf{F}^{(\ell)}$ is Lipschitz on $\mathcal{S}_{\ell-1}$.** We define the set of row vectors that occur as inputs to the first LayerNorm in layer $\ell$:

$$\mathcal{T}_{\ell,F} \triangleq \left\{ \left(\mathbf{Y} + \mathrm{MHSA}^{(\ell)}(\mathbf{Y})\right)_{i,:} : \ \mathbf{Y} \in \mathcal{S}_{\ell-1}, \ i \in [M] \right\} \subseteq \mathbb{R}^d.$$

Since $\mathcal{S}_{\ell-1}$ is compact and $\mathbf{Y} \mapsto \mathbf{Y} + \mathrm{MHSA}^{(\ell)}(\mathbf{Y})$ is continuous, $\mathcal{T}_{\ell,F}$ is compact; moreover $\mathrm{Var}(\cdot) + \varepsilon_{\mathrm{ln}} \geq \varepsilon_{\mathrm{ln}} > 0$. Thus Lemma D.4 applies row-wise on $\mathcal{T}_{\ell,F}$:

$$\begin{aligned}
\|\mathbf{F}^{(\ell)}(\mathbf{Y}) - \mathbf{F}^{(\ell)}(\mathbf{Y}')\|_{2,\infty} &\leq L_{\mathrm{LN}}(\mathcal{T}_{\ell,F}) \cdot \left\| \left(\mathbf{Y} + \mathrm{MHSA}^{(\ell)}(\mathbf{Y})\right) - \left(\mathbf{Y}' + \mathrm{MHSA}^{(\ell)}(\mathbf{Y}')\right) \right\|_{2,\infty} \\
&\leq L_{\mathrm{LN}}(\mathcal{T}_{\ell,F}) \cdot \left( \|\mathbf{Y} - \mathbf{Y}'\|_{2,\infty} + \|\mathrm{MHSA}^{(\ell)}(\mathbf{Y}) - \mathrm{MHSA}^{(\ell)}(\mathbf{Y}')\|_{2,\infty} \right).
\end{aligned} \tag{32}$$

In order to control the MHSA term, we decompose it into: (i) linear projections to $\mathbf{Q}, \mathbf{K}, \mathbf{V}$ per head, (ii) head-level attention, and (iii) the output projection. We apply Lemma D.1 to the linear maps $\mathbf{Y} \mapsto \mathbf{Y} \cdot \mathbf{W}_Q^{(\ell,h)}, \mathbf{Y} \mapsto \mathbf{Y} \cdot \mathbf{W}_K^{(\ell,h)}$, and $\mathbf{Y} \mapsto \mathbf{Y} \cdot \mathbf{W}_V^{(\ell,h)}$, then Lemma D.3 to each head-level attention map (with $N_q = N = M$ inside the mixer and $d_k = d_h$ per head), and finally Lemma D.1 to the output projection $\mathbf{W}_O^{(\ell)}$. This yields a deterministic constant $L_{\mathrm{MHSA}}^{(\ell)}(\mathcal{S}_{\ell-1})$ such that for all $\mathbf{Y}, \mathbf{Y}' \in \mathcal{S}_{\ell-1}$,

$$\|\mathrm{MHSA}^{(\ell)}(\mathbf{Y}) - \mathrm{MHSA}^{(\ell)}(\mathbf{Y}')\|_{2,\infty} \leq L_{\mathrm{MHSA}}^{(\ell)}(\mathcal{S}_{\ell-1}) \cdot \|\mathbf{Y} - \mathbf{Y}'\|_{2,\infty}. \tag{33}$$

Substituting (33) into (32) gives

$$\|\mathbf{F}^{(\ell)}(\mathbf{Y}) - \mathbf{F}^{(\ell)}(\mathbf{Y}')\|_{2,\infty} \leq L_{\mathrm{LN}}(\mathcal{T}_{\ell,F}) \cdot \left(1 + L_{\mathrm{MHSA}}^{(\ell)}(\mathcal{S}_{\ell-1})\right) \cdot \|\mathbf{Y} - \mathbf{Y}'\|_{2,\infty}. \tag{34}$$

**Step 2: The map $\mathbf{G}^{(\ell)}$ is Lipschitz on $\mathbf{F}^{(\ell)}(\mathcal{S}_{\ell-1})$.** We define the set of row vectors that occur as inputs to the second LayerNorm in layer $\ell$:

$$\mathcal{T}_{\ell,G} \triangleq \left\{ \left(\mathbf{A} + \mathrm{FFN}^{(\ell)}(\mathbf{A})\right)_{i,:} : \ \mathbf{A} \in \mathbf{F}^{(\ell)}(\mathcal{S}_{\ell-1}), \ i \in [M] \right\} \subseteq \mathbb{R}^d.$$

The set $\mathbf{F}^{(\ell)}(\mathcal{S}_{\ell-1})$ is compact by continuity of $\mathbf{F}^{(\ell)}$, so $\mathcal{T}_{\ell,G}$ is compact. Hence Lemma D.4 applies row-wise on $\mathcal{T}_{\ell,G}$.

Using Lemma D.1 and $L_\sigma$-Lipschitzness of $\sigma$ (cf. (15)), we have for all $\mathbf{A}, \mathbf{A}'$,

$$\|\mathrm{FFN}^{(\ell)}(\mathbf{A}) - \mathrm{FFN}^{(\ell)}(\mathbf{A}')\|_{2,\infty} \leq \|\mathbf{W}_2^{(\ell)}\|_{\mathrm{op}} \cdot L_\sigma \cdot \|\mathbf{W}_1^{(\ell)}\|_{\mathrm{op}} \cdot \|\mathbf{A} - \mathbf{A}'\|_{2,\infty}. \tag{35}$$

Therefore, by Lemma D.4 and the residual addition,

$$\|\mathbf{G}^{(\ell)}(\mathbf{A}) - \mathbf{G}^{(\ell)}(\mathbf{A}')\|_{2,\infty} \leq L_{\mathrm{LN}}(\mathcal{T}_{\ell,G}) \cdot \left(1 + \|\mathbf{W}_2^{(\ell)}\|_{\mathrm{op}} \cdot L_\sigma \cdot \|\mathbf{W}_1^{(\ell)}\|_{\mathrm{op}}\right) \cdot \|\mathbf{A} - \mathbf{A}'\|_{2,\infty}. \tag{36}$$

**Step 3: One layer is Lipschitz on $\mathcal{S}_{\ell-1}$.** Let $\mathbf{A} = \mathbf{F}^{(\ell)}(\mathbf{Y})$ and $\mathbf{A}' = \mathbf{F}^{(\ell)}(\mathbf{Y}')$ in (36) and then apply (34). This yields, for all $\mathbf{Y}, \mathbf{Y}' \in \mathcal{S}_{\ell-1}$,

$$\|\mathbf{H}^{(\ell)}(\mathbf{Y}) - \mathbf{H}^{(\ell)}(\mathbf{Y}')\|_{2,\infty} \leq L_\ell(\mathcal{S}) \cdot \|\mathbf{Y} - \mathbf{Y}'\|_{2,\infty},$$

with the admissible choice

$$L_\ell(\mathcal{S}) \triangleq L_{\mathrm{LN}}(\mathcal{T}_{\ell,G}) \cdot \left(1 + \|\mathbf{W}_2^{(\ell)}\|_{\mathrm{op}} \cdot L_\sigma \cdot \|\mathbf{W}_1^{(\ell)}\|_{\mathrm{op}}\right) \cdot L_{\mathrm{LN}}(\mathcal{T}_{\ell,F}) \cdot \left(1 + L_{\mathrm{MHSA}}^{(\ell)}(\mathcal{S}_{\ell-1})\right).$$

**Step 4: Propagating across layers.** For $\mathbf{U}, \mathbf{U}' \in \mathcal{S}$, let $\mathbf{Y}^{(0)} = \mathbf{U}$ and $\mathbf{Y}'^{(0)} = \mathbf{U}'$. Iterating the one-layer Lipschitz bound over $\ell = 1, \ldots, L_{\mathrm{mix}}$ gives

$$\|\mathbf{Y}^{(L_{\mathrm{mix}})} - \mathbf{Y}'^{(L_{\mathrm{mix}})}\|_{2,\infty} \leq \left( \prod_{\ell=1}^{L_{\mathrm{mix}}} L_\ell(\mathcal{S}) \right) \cdot \|\mathbf{U} - \mathbf{U}'\|_{2,\infty}.$$

Setting $L_{\mathrm{mix}}(\mathcal{S}) \triangleq \prod_{\ell=1}^{L_{\mathrm{mix}}} L_\ell(\mathcal{S})$ proves the theorem. $\qquad\square$

### D.1. From Row-Wise LayerNorm Stability to a Checkable Layer Constant

**Definition D.6** (Pre-LN row-sets induced by a segment). For a segment $\mathcal{S} \subseteq \mathbb{R}^{M \times d}$ and encoder layer $\ell \in \{1, \ldots, L_{\mathrm{mix}}\}$, let $\mathcal{S}_{\ell-1}$ denote the set of possible inputs to layer $\ell$ induced by $\mathcal{S}$ (i.e., the image of $\mathcal{S}$ under the first $\ell - 1$ layers of $\mathrm{Mixer}_{L_{\mathrm{mix}}}$, with $\mathcal{S}_0 \triangleq \mathcal{S}$).

We define the two pre-LN row-sets:

$$\mathcal{T}_{\ell,F} \triangleq \left\{ \mathbf{u} \in \mathbb{R}^d : \exists\, \mathbf{Y} \in \mathcal{S}_{\ell-1},\ \exists\, i \in [M] \text{ s.t. } \mathbf{u} = \left( \mathbf{Y} + \mathrm{MHSA}^{(\ell)}(\mathbf{Y}) \right)_{i,:} \right\},$$

$$\mathcal{T}_{\ell,G} \triangleq \left\{ \mathbf{u} \in \mathbb{R}^d : \exists\, \mathbf{A} \in \mathbf{F}^{(\ell)}(\mathcal{S}_{\ell-1}),\ \exists\, i \in [M] \text{ s.t. } \mathbf{u} = \left( \mathbf{A} + \mathrm{FFN}^{(\ell)}(\mathbf{A}) \right)_{i,:} \right\},$$

where $\mathbf{F}^{(\ell)}(\mathbf{Y}) = \mathrm{LayerNorm}\left( \mathbf{Y} + \mathrm{MHSA}^{(\ell)}(\mathbf{Y}) \right)$ as in the proof of Theorem D.5. △

**Lemma D.7** (Per-layer Lipschitz constant decomposes into LayerNorm and sublayer constants). *Let* $\ell \in \{1, \ldots, L_{\mathrm{mix}}\}$, *and let* $\mathcal{S} \subseteq \mathbb{R}^{M \times d}$ *be a segment. The activation* $\sigma$ *is* $L_\sigma$-*Lipschitz and* $\varepsilon_{\mathrm{ln}} > 0$ *(Section A.1). Suppose that both pre-LN row-sets* $\mathcal{T}_{\ell,F}$ *and* $\mathcal{T}_{\ell,G}$ *from Definition D.6 have positive stabilized variance lower bounds:*

$$m_{\ell,F} \triangleq \inf_{\mathbf{u} \in \mathcal{T}_{\ell,F}} \left( \mathrm{Var}(\mathbf{u}) + \varepsilon_{\mathrm{ln}} \right) > 0, \qquad m_{\ell,G} \triangleq \inf_{\mathbf{u} \in \mathcal{T}_{\ell,G}} \left( \mathrm{Var}(\mathbf{u}) + \varepsilon_{\mathrm{ln}} \right) > 0.$$

*Then the* $\ell$*th post-LN encoder layer map* $\mathrm{Layer}^{(\ell)} : \mathbb{R}^{M \times d} \to \mathbb{R}^{M \times d}, \mathbf{Y}^{(\ell-1)} \mapsto \mathbf{Y}^{(\ell)}$, *as defined in* (16), *is Lipschitz on* $\mathcal{S}_{\ell-1}$ *under* $\|\cdot\|_{2,\infty}$ *with constant*

$$L_\ell(\mathcal{S}) \leq L_{\mathrm{LN}}\left( \mathcal{T}_{\ell,G} \right) \cdot \left( 1 + L_{\mathrm{FFN}}^{(\ell)} \right) \cdot L_{\mathrm{LN}}\left( \mathcal{T}_{\ell,F} \right) \cdot \left( 1 + L_{\mathrm{MHSA}}^{(\ell)}(\mathcal{S}_{\ell-1}) \right),$$

*where*

$$L_{\mathrm{FFN}}^{(\ell)} \triangleq \|\mathbf{W}_2^{(\ell)}\|_{\mathrm{op}} \cdot L_\sigma \cdot \|\mathbf{W}_1^{(\ell)}\|_{\mathrm{op}},$$

$$L_{\mathrm{MHSA}}^{(\ell)}(\mathcal{S}_{\ell-1}) \triangleq \|\mathbf{W}_O^{(\ell)}\|_{\mathrm{op}} \cdot \sum_{h=1}^{H} \left( \left( \Gamma_Q^{(\ell,h)} \cdot \Gamma_V^{(\ell,h)} \cdot \|\mathbf{W}_K^{(\ell,h)}\|_{\mathrm{op}} \right) + \left( \Gamma_K^{(\ell,h)} \cdot \Gamma_V^{(\ell,h)} \cdot \|\mathbf{W}_Q^{(\ell,h)}\|_{\mathrm{op}} \right) + \|\mathbf{W}_V^{(\ell,h)}\|_{\mathrm{op}} \right),$$

*and the head-wise row-norm envelopes are*

$$\Gamma_Q^{(\ell,h)} \triangleq \frac{1}{\sqrt{d_h}} \cdot \sup_{\mathbf{Y} \in \mathcal{S}_{\ell-1}} \left\| \mathbf{Y} \cdot \mathbf{W}_Q^{(\ell,h)} \right\|_{2,\infty},$$

$$\Gamma_K^{(\ell,h)} \triangleq \sup_{\mathbf{Y} \in \mathcal{S}_{\ell-1}} \left\| \mathbf{Y} \cdot \mathbf{W}_K^{(\ell,h)} \right\|_{2,\infty},$$

$$\Gamma_V^{(\ell,h)} \triangleq \max\left\{ \sup_{\mathbf{Y} \in \mathcal{S}_{\ell-1}} \left\| \mathbf{Y} \cdot \mathbf{W}_V^{(\ell,h)} \right\|_{2,\infty},\ \sup_{\mathbf{Y}' \in \mathcal{S}_{\ell-1}} \left\| \mathbf{Y}' \cdot \mathbf{W}_V^{(\ell,h)} \right\|_{2,\infty} \right\} = \sup_{\mathbf{Y} \in \mathcal{S}_{\ell-1}} \left\| \mathbf{Y} \cdot \mathbf{W}_V^{(\ell,h)} \right\|_{2,\infty}.$$

*In particular,* $L_\ell(\mathcal{S})$ *is controlled by two LayerNorm constants evaluated on the* pre-LN row-sets *and by row-norm envelopes and operator norms of weight matrices.*

*Proof.* Let $\mathbf{Y}, \mathbf{Y}' \in \mathcal{S}_{\ell-1}$. We define the residual-LN maps

$$\mathbf{F}^{(\ell)}(\mathbf{Y}) \triangleq \mathrm{LayerNorm}\left( \mathbf{Y} + \mathrm{MHSA}^{(\ell)}(\mathbf{Y}) \right),$$

$$\mathbf{G}^{(\ell)}(\mathbf{A}) \triangleq \mathrm{LayerNorm}\left( \mathbf{A} + \mathrm{FFN}^{(\ell)}(\mathbf{A}) \right),$$

so that the layer map is the composition $\mathrm{Layer}^{(\ell)} = \mathbf{G}^{(\ell)} \circ \mathbf{F}^{(\ell)}$.

*Step 1: bounding* $\mathbf{F}^{(\ell)}$ *on* $\mathcal{S}_{\ell-1}$. By Definition D.6, for any $\mathbf{Y} \in \mathcal{S}_{\ell-1}$ each row of $\mathbf{Y} + \mathrm{MHSA}^{(\ell)}(\mathbf{Y})$ lies in $\mathcal{T}_{\ell,F}$. Applying Lemma D.4 row-wise yields

$$\left\|\mathbf{F}^{(\ell)}(\mathbf{Y}) - \mathbf{F}^{(\ell)}(\mathbf{Y}')\right\|_{2,\infty} \le L_{\mathrm{LN}}\left(\mathcal{T}_{\ell,F}\right) \cdot \left\|\left(\mathbf{Y} + \mathrm{MHSA}^{(\ell)}(\mathbf{Y})\right) - \left(\mathbf{Y}' + \mathrm{MHSA}^{(\ell)}(\mathbf{Y}')\right)\right\|_{2,\infty}$$
$$\le L_{\mathrm{LN}}\left(\mathcal{T}_{\ell,F}\right) \cdot \left(\left\|\mathbf{Y} - \mathbf{Y}'\right\|_{2,\infty} + \left\|\mathrm{MHSA}^{(\ell)}(\mathbf{Y}) - \mathrm{MHSA}^{(\ell)}(\mathbf{Y}')\right\|_{2,\infty}\right).$$

*Step 1a: bounding the MHSA difference head-by-head.* For a head $h$, we define projected queries / keys / values

$$\mathbf{Q}^{(h)}(\mathbf{Y}) \triangleq \mathbf{Y} \cdot \mathbf{W}_Q^{(\ell,h)}, \qquad \mathbf{K}^{(h)}(\mathbf{Y}) \triangleq \mathbf{Y} \cdot \mathbf{W}_K^{(\ell,h)}, \qquad \mathbf{V}^{(h)}(\mathbf{Y}) \triangleq \mathbf{Y} \cdot \mathbf{W}_V^{(\ell,h)}.$$

By Lemma D.1,

$$\left\|\mathbf{Q}^{(h)}(\mathbf{Y}) - \mathbf{Q}^{(h)}(\mathbf{Y}')\right\|_{2,\infty} \le \|\mathbf{W}_Q^{(\ell,h)}\|_{\mathrm{op}} \cdot \|\mathbf{Y} - \mathbf{Y}'\|_{2,\infty},$$
$$\left\|\mathbf{K}^{(h)}(\mathbf{Y}) - \mathbf{K}^{(h)}(\mathbf{Y}')\right\|_{2,\infty} \le \|\mathbf{W}_K^{(\ell,h)}\|_{\mathrm{op}} \cdot \|\mathbf{Y} - \mathbf{Y}'\|_{2,\infty},$$
$$\left\|\mathbf{V}^{(h)}(\mathbf{Y}) - \mathbf{V}^{(h)}(\mathbf{Y}')\right\|_{2,\infty} \le \|\mathbf{W}_V^{(\ell,h)}\|_{\mathrm{op}} \cdot \|\mathbf{Y} - \mathbf{Y}'\|_{2,\infty}.$$

Moreover, by definition of $\Gamma_Q^{(\ell,h)}, \Gamma_K^{(\ell,h)}, \Gamma_V^{(\ell,h)}$ we have, uniformly over $\mathbf{Y}, \mathbf{Y}' \in \mathcal{S}_{\ell-1}$,

$$\frac{1}{\sqrt{d_h}} \cdot \left\|\mathbf{Q}^{(h)}(\mathbf{Y})\right\|_{2,\infty} \le \Gamma_Q^{(\ell,h)}, \qquad \left\|\mathbf{K}^{(h)}(\mathbf{Y})\right\|_{2,\infty} \le \Gamma_K^{(\ell,h)}, \qquad \max\left\{\left\|\mathbf{V}^{(h)}(\mathbf{Y})\right\|_{2,\infty}, \left\|\mathbf{V}^{(h)}(\mathbf{Y}')\right\|_{2,\infty}\right\} \le \Gamma_V^{(\ell,h)}.$$

The head map $(\mathbf{Q}, \mathbf{K}, \mathbf{V}) \mapsto \mathrm{Atten}(\mathbf{Q}; \mathbf{K}, \mathbf{V})$ varies in all three arguments, so we decompose its difference across $\mathbf{Q}, \mathbf{K}, \mathbf{V}$ by the triangle inequality. Lemma D.3 (with $N_q = N_k = M$ and $d_k = d_h$) controls the value and key variations; the query variation satisfies the symmetric bound $\left\|\mathrm{Atten}(\mathbf{Q}; \mathbf{K}, \mathbf{V}) - \mathrm{Atten}(\mathbf{Q}'; \mathbf{K}, \mathbf{V})\right\|_{2,\infty} \le \frac{1}{\sqrt{d_h}} \cdot \|\mathbf{K}\|_{2,\infty} \cdot \Gamma_V^{(\ell,h)} \cdot \|\mathbf{Q} - \mathbf{Q}'\|_{2,\infty}$ (the same $\ell_\infty \to \ell_1$ softmax argument applied to the $\mathbf{Q}$-induced logit perturbation, using $\frac{1}{\sqrt{d_h}}\|\mathbf{K}^{(h)}(\mathbf{Y}')\|_{2,\infty} \le \Gamma_K^{(\ell,h)}$). Combining the three contributions gives

$$\left\|\mathrm{Atten}\big(\mathbf{Q}^{(h)}(\mathbf{Y}); \mathbf{K}^{(h)}(\mathbf{Y}), \mathbf{V}^{(h)}(\mathbf{Y})\big) - \mathrm{Atten}\big(\mathbf{Q}^{(h)}(\mathbf{Y}'); \mathbf{K}^{(h)}(\mathbf{Y}'), \mathbf{V}^{(h)}(\mathbf{Y}')\big)\right\|_{2,\infty}$$
$$\le \left(\Gamma_Q^{(\ell,h)} \cdot \Gamma_V^{(\ell,h)} \cdot \|\mathbf{W}_K^{(\ell,h)}\|_{\mathrm{op}} + \Gamma_K^{(\ell,h)} \cdot \Gamma_V^{(\ell,h)} \cdot \|\mathbf{W}_Q^{(\ell,h)}\|_{\mathrm{op}} + \|\mathbf{W}_V^{(\ell,h)}\|_{\mathrm{op}}\right) \cdot \|\mathbf{Y} - \mathbf{Y}'\|_{2,\infty}.$$

*Step 1b: combining heads and output projection.* Let $\mathbf{U}^{(h)}(\mathbf{Y}) \in \mathbb{R}^{M \times d_h}$ denote the output of head $h \in \{1, \dots, H\}$ before the final output projection. We define the concatenation operator

$$\mathrm{Concat}_h \, \mathbf{U}^{(h)}(\mathbf{Y}) \triangleq \big(\mathbf{U}^{(1)}(\mathbf{Y}) \,\big|\, \mathbf{U}^{(2)}(\mathbf{Y}) \,\big|\, \cdots \,\big|\, \mathbf{U}^{(H)}(\mathbf{Y})\big) = \bigoplus_h \mathbf{U}^{(h)}(\mathbf{Y}) \in \mathbb{R}^{M \times (Hd_h)},$$

i.e., concatenation along the column (feature) dimension. Equivalently, for each row $i \in [M]$,

$$\big(\mathrm{Concat}_h \, \mathbf{U}^{(h)}(\mathbf{Y})\big)_{i,:} = \big(\mathbf{U}^{(1)}(\mathbf{Y})_{i,:}, \, \mathbf{U}^{(2)}(\mathbf{Y})_{i,:}, \, \dots, \, \mathbf{U}^{(H)}(\mathbf{Y})_{i,:}\big).$$

**Difference of concatenations.** For $\mathbf{Y}, \mathbf{Y}'$, we define

$$\mathrm{Concat}_h \, \mathbf{U}^{(h)}(\mathbf{Y}) - \mathrm{Concat}_h \, \mathbf{U}^{(h)}(\mathbf{Y}') \triangleq \big(\mathbf{U}^{(1)}(\mathbf{Y}) - \mathbf{U}^{(1)}(\mathbf{Y}') \,\big|\, \cdots \,\big|\, \mathbf{U}^{(H)}(\mathbf{Y}) - \mathbf{U}^{(H)}(\mathbf{Y}')\big) \in \mathbb{R}^{M \times (Hd_h)}.$$

The multi-head concatenation satisfies, for each row $i \in [M]$,

$$\left\|\big(\mathbf{U}_{i,:}^{(1)}, \dots, \mathbf{U}_{i,:}^{(H)}\big)\right\|_2 = \sqrt{\sum_{h=1}^{H} \|\mathbf{U}_{i,:}^{(h)}\|_2^2} \le \sum_{h=1}^{H} \|\mathbf{U}_{i,:}^{(h)}\|_2,$$

hence $\left\|\mathrm{Concat}_h\,\mathbf{U}^{(h)}(\mathbf{Y}) - \mathrm{Concat}_h\,\mathbf{U}^{(h)}(\mathbf{Y}')\right\|_{2,\infty} \le \sum_{h=1}^H \|\mathbf{U}^{(h)}(\mathbf{Y}) - \mathbf{U}^{(h)}(\mathbf{Y}')\|_{2,\infty}$. Finally, applying the output projection and Lemma D.1 gives

$$\left\|\mathrm{MHSA}^{(\ell)}(\mathbf{Y}) - \mathrm{MHSA}^{(\ell)}(\mathbf{Y}')\right\|_{2,\infty} \;\le\; L_{\mathrm{MHSA}}^{(\ell)}(\mathcal{S}_{\ell-1}) \cdot \|\mathbf{Y} - \mathbf{Y}'\|_{2,\infty}.$$

Substituting into the bound for $\mathbf{F}^{(\ell)}$ yields

$$\left\|\mathbf{F}^{(\ell)}(\mathbf{Y}) - \mathbf{F}^{(\ell)}(\mathbf{Y}')\right\|_{2,\infty} \;\le\; L_{\mathrm{LN}}\!\left(\mathcal{T}_{\ell,F}\right) \cdot \left(1 + L_{\mathrm{MHSA}}^{(\ell)}(\mathcal{S}_{\ell-1})\right) \cdot \|\mathbf{Y} - \mathbf{Y}'\|_{2,\infty}.$$

*Step 2: bounding $\mathbf{G}^{(\ell)}$ on $\mathbf{F}^{(\ell)}(\mathcal{S}_{\ell-1})$.* Let $\mathbf{A}, \mathbf{A}' \in \mathbf{F}^{(\ell)}(\mathcal{S}_{\ell-1})$. By Definition D.6, the rows of $\mathbf{A} + \mathrm{FFN}^{(\ell)}(\mathbf{A})$ lie in $\mathcal{T}_{\ell,G}$. Applying Lemma D.4 row-wise gives

$$\left\|\mathbf{G}^{(\ell)}(\mathbf{A}) - \mathbf{G}^{(\ell)}(\mathbf{A}')\right\|_{2,\infty} \le L_{\mathrm{LN}}\!\left(\mathcal{T}_{\ell,G}\right) \cdot \left(\left\|\mathbf{A} - \mathbf{A}'\right\|_{2,\infty} + \left\|\mathrm{FFN}^{(\ell)}(\mathbf{A}) - \mathrm{FFN}^{(\ell)}(\mathbf{A}')\right\|_{2,\infty}\right).$$

For the FFN, applying Lemma D.1 twice and using the Lipschitzness of $\sigma$ yield:

$$\begin{aligned}
\left\|\mathrm{FFN}^{(\ell)}(\mathbf{A}) - \mathrm{FFN}^{(\ell)}(\mathbf{A}')\right\|_{2,\infty} &= \left\|\sigma\!\left(\mathbf{A}\cdot\mathbf{W}_1^{(\ell)} + \mathbf{b}_1^{(\ell)}\right)\cdot\mathbf{W}_2^{(\ell)} - \sigma\!\left(\mathbf{A}'\cdot\mathbf{W}_1^{(\ell)} + \mathbf{b}_1^{(\ell)}\right)\cdot\mathbf{W}_2^{(\ell)}\right\|_{2,\infty} \\
&\le \|\mathbf{W}_2^{(\ell)}\|_{\mathrm{op}} \cdot L_\sigma \cdot \|\mathbf{W}_1^{(\ell)}\|_{\mathrm{op}} \cdot \|\mathbf{A} - \mathbf{A}'\|_{2,\infty} \\
&= L_{\mathrm{FFN}}^{(\ell)} \cdot \|\mathbf{A} - \mathbf{A}'\|_{2,\infty}.
\end{aligned}$$

Thus

$$\left\|\mathbf{G}^{(\ell)}(\mathbf{A}) - \mathbf{G}^{(\ell)}(\mathbf{A}')\right\|_{2,\infty} \;\le\; L_{\mathrm{LN}}\!\left(\mathcal{T}_{\ell,G}\right) \cdot \left(1 + L_{\mathrm{FFN}}^{(\ell)}\right) \cdot \|\mathbf{A} - \mathbf{A}'\|_{2,\infty}.$$

*Step 3: composing.* With $\mathbf{A} = \mathbf{F}^{(\ell)}(\mathbf{Y})$ and $\mathbf{A}' = \mathbf{F}^{(\ell)}(\mathbf{Y}')$, combining Step 1 and Step 2 gives:

$$\left\|\mathrm{Layer}^{(\ell)}(\mathbf{Y}) - \mathrm{Layer}^{(\ell)}(\mathbf{Y}')\right\|_{2,\infty} \;\le\; L_{\mathrm{LN}}\!\left(\mathcal{T}_{\ell,G}\right) \cdot \left(1 + L_{\mathrm{FFN}}^{(\ell)}\right) \cdot L_{\mathrm{LN}}\!\left(\mathcal{T}_{\ell,F}\right) \cdot \left(1 + L_{\mathrm{MHSA}}^{(\ell)}(\mathcal{S}_{\ell-1})\right) \cdot \|\mathbf{Y} - \mathbf{Y}'\|_{2,\infty}.$$

Any per-layer Lipschitz constant $L_\ell(\mathcal{S})$ may be chosen as the right-hand side coefficient, proving the claim. $\qquad\square$

**Theorem D.8** (Forward-pass-computable upper bound for $L_\ell(\mathcal{S})$). *Let $\ell \in \{1, \dots, L_{\mathrm{mix}}\}$, and let $\mathcal{S} \subseteq \mathbb{R}^{M\times d}$ be a segment. Let $L_\ell(\mathcal{S})$ be any Lipschitz constant of the $\ell$th encoder layer on $\mathcal{S}_{\ell-1}$. Then the quantity*

$$\overline{L}_\ell(\mathcal{S}) \triangleq L_{\mathrm{LN}}\!\left(\mathcal{T}_{\ell,G}\right) \cdot \left(1 + L_{\mathrm{FFN}}^{(\ell)}\right) \cdot L_{\mathrm{LN}}\!\left(\mathcal{T}_{\ell,F}\right) \cdot \left(1 + L_{\mathrm{MHSA}}^{(\ell)}(\mathcal{S}_{\ell-1})\right)$$

*satisfies $L_\ell(\mathcal{S}) \le \overline{L}_\ell(\mathcal{S})$ and is* forward-pass computable *in the following sense.*

*Each factor in $\overline{L}_\ell(\mathcal{S})$ admits an explicit* conservative *upper bound computed from the two endpoint activations $\mathbf{Y}_{\ell-1}^{\mathrm{det}}, \mathbf{Y}_{\ell-1}$ of $\mathcal{S}_{\ell-1}$ and the weight operator norms (e.g., by power iteration):*

1. *$\Gamma_Q^{(\ell,h)}, \Gamma_K^{(\ell,h)}, \Gamma_V^{(\ell,h)}$ can be upper bounded by evaluating $\left\|\mathbf{Y}_{\ell-1}^{\mathrm{det}} \cdot \mathbf{W}_{\{Q,K,V\}}^{(\ell,h)}\right\|_{2,\infty}$ and $\left\|\mathbf{Y}_{\ell-1} \cdot \mathbf{W}_{\{Q,K,V\}}^{(\ell,h)}\right\|_{2,\infty}$ and taking the maximum;*
2. *$L_{\mathrm{FFN}}^{(\ell)}$ depends only on $\|\mathbf{W}_1^{(\ell)}\|_{\mathrm{op}}$, $\|\mathbf{W}_2^{(\ell)}\|_{\mathrm{op}}$, and $L_\sigma$;*
3. *$L_{\mathrm{LN}}(\mathcal{T}_{\ell,F})$ and $L_{\mathrm{LN}}(\mathcal{T}_{\ell,G})$ admit endpoint-based upper bounds by (i) bounding the possible deviation of pre-LN rows away from the endpoint rows using sublayer Lipschitz constants and (ii) translating this deviation into conservative bounds on the variance and centered norm terms that control $L_{\mathrm{LN}}(\cdot)$ (as described in the proof).*

*Proof.* **Part I.** The bound $L_\ell(\mathcal{S}) \le \overline{L}_\ell(\mathcal{S})$ is the product form of Lemma D.7: each residual sublayer contributes a $(1+\mathrm{Lip})$ factor and each LayerNorm a $L_{\mathrm{LN}}(\cdot)$ factor.

**Part II (checkability).** On $\mathcal{S}_{\ell-1}$ the map $\mathbf{Y} \mapsto \|\mathbf{Y}\cdot\mathbf{W}\|_{2,\infty}$ is convex, so its supremum is attained at an endpoint; evaluating it at $\mathbf{Y}_{\ell-1}^{\mathrm{det}}$ and $\mathbf{Y}_{\ell-1}$ and taking the maximum bounds the head envelopes $\Gamma_Q^{(\ell,h)}, \Gamma_K^{(\ell,h)}, \Gamma_V^{(\ell,h)}$, hence $L_{\mathrm{MHSA}}^{(\ell)}(\mathcal{S}_{\ell-1})$. The factor $L_{\mathrm{FFN}}^{(\ell)} = \|\mathbf{W}_2^{(\ell)}\|_{\mathrm{op}} \cdot L_\sigma \cdot \|\mathbf{W}_1^{(\ell)}\|_{\mathrm{op}}$ is computed from operator norms (power iteration).

It remains to bound the two LayerNorm constants. By Lemma D.4, $L_{\mathrm{LN}}(\mathcal{T})$ is controlled by $m_{\mathcal{T}} = \inf_{\mathbf{u} \in \mathcal{T}}(\mathrm{Var}(\mathbf{u}) + \varepsilon_{\mathrm{ln}})$ and $C_{\mathcal{T}} = \sup_{\mathbf{u} \in \mathcal{T}} \|\mathbf{u} - \mu(\mathbf{u})\mathbf{1}\|_2$. The pre-LN map $(\mathbf{H}_{\ell,F}(\mathbf{Y}) = \mathbf{Y} + \mathrm{MHSA}^{(\ell)}(\mathbf{Y})$ for $\mathcal{T}_{\ell,F}$, and $\mathbf{H}_{\ell,G}(\mathbf{A}) = \mathbf{A} + \mathrm{FFN}^{(\ell)}(\mathbf{A})$ for $\mathcal{T}_{\ell,G})$ is Lipschitz with constant $L^{\mathrm{pre}} = 1 + L_{\mathrm{sub}}$, $L_{\mathrm{sub}} \in \{L_{\mathrm{MHSA}}^{(\ell)}(\mathcal{S}_{\ell-1}), L_{\mathrm{FFN}}^{(\ell)}\}$, so every pre-LN row lies within $\ell_2$-distance $\Delta \triangleq L^{\mathrm{pre}} \cdot r$ of a row of an endpoint image, where $r$ is the $\|\cdot\|_{2,\infty}$ radius of the segment (resp. of its image under $\mathbf{F}^{(\ell)}$). The mean map is $1/\sqrt{d}$-Lipschitz and the centering map $\mathbf{u} \mapsto \mathbf{u} - \mu(\mathbf{u})\mathbf{1}$ is 2-Lipschitz, so the centered norm of any pre-LN row deviates from that of the nearest endpoint row by at most $2\Delta$. Writing $c^{\mathrm{min}}, c^{\mathrm{max}}$ for the smallest and largest centered norm over the finitely many endpoint rows,

$$m^{\mathrm{check}} \triangleq \tfrac{1}{d}\big(\max\{0,\, c^{\mathrm{min}} - 2\Delta\}\big)^2 + \varepsilon_{\mathrm{ln}} \ \leq\ m_{\mathcal{T}}, \qquad C^{\mathrm{check}} \triangleq c^{\mathrm{max}} + 2\Delta \ \geq\ C_{\mathcal{T}},$$

computed separately for $\mathcal{T}_{\ell,F}$ and $\mathcal{T}_{\ell,G}$. Substituting these into Lemma D.4 bounds $L_{\mathrm{LN}}(\mathcal{T}_{\ell,F})$ and $L_{\mathrm{LN}}(\mathcal{T}_{\ell,G})$ from endpoint activations. $\qquad\square$

## E. A Deterministic Stability Bound for Softmax Attention

Lemma E.1 bounds the end-to-end attention output deviation in the row-wise $(2,\infty)$ norm in terms of key / value perturbations.

**Lemma E.1** (Deterministic perturbation bound for softmax attention). *Let* $\mathbf{Q} \in \mathbb{R}^{N_q \times d_k}$. *Let* $(\mathbf{K}, \mathbf{V})$ *and* $(\mathbf{K}', \mathbf{V}')$ *be two key / value pairs with* $\mathbf{K}, \mathbf{K}' \in \mathbb{R}^{N \times d_k}$ *and* $\mathbf{V}, \mathbf{V}' \in \mathbb{R}^{N \times d_v}$. *We define the row-wise* $(2,\infty)$ *matrix norm*

$$\|\mathbf{X}\|_{2,\infty} \triangleq \max_i \|\mathbf{X}_{i,:}\|_2,$$

*and the perturbation magnitudes*

$$\rho_K \triangleq \|\mathbf{K} - \mathbf{K}'\|_{2,\infty}, \qquad \rho_V \triangleq \|\mathbf{V} - \mathbf{V}'\|_{2,\infty}.$$

*We define also the query-dependent sensitivity factor and a value-row bound*

$$\Gamma_Q^{(N)} \triangleq \frac{1}{\sqrt{d_k}}\|\mathbf{Q}\|_{2,\infty}, \qquad V_{\mathrm{max}} \triangleq \|\mathbf{V}\|_{2,\infty}.$$

*Then*

$$\big\|\mathrm{Atten}(\mathbf{Q};\mathbf{K},\mathbf{V}) - \mathrm{Atten}(\mathbf{Q};\mathbf{K}',\mathbf{V}')\big\|_F \ \leq\ \sqrt{N_q} \cdot \Big(\Gamma_Q^{(N)} \cdot \rho_K \cdot V_{\mathrm{max}} + \rho_V\Big). \tag{37}$$

*Proof.* **Setup for one query.** For $j \in [N_q]$, we write $\mathbf{q} \triangleq \mathbf{Q}_{j,:} \in \mathbb{R}^{d_k}$, with scaled logits

$$\mathbf{s} \triangleq \frac{\mathbf{q} \cdot \mathbf{K}^\top}{\sqrt{d_k}} \in \mathbb{R}^N, \qquad \mathbf{s}' \triangleq \frac{\mathbf{q} \cdot \mathbf{K}'^\top}{\sqrt{d_k}} \in \mathbb{R}^N,$$

and the corresponding softmax weight vectors

$$\boldsymbol{\alpha} \triangleq \mathrm{softmax}(\mathbf{s}) \in \mathbb{R}^N, \qquad \boldsymbol{\alpha}' \triangleq \mathrm{softmax}(\mathbf{s}') \in \mathbb{R}^N.$$

The associated attention output rows are

$$\mathbf{y} \triangleq \boldsymbol{\alpha}^\top \cdot \mathbf{V} \in \mathbb{R}^{d_v}, \qquad \mathbf{y}' \triangleq \boldsymbol{\alpha}'^\top \cdot \mathbf{V}' \in \mathbb{R}^{d_v}.$$

We decompose

$$\begin{aligned} \mathbf{y} - \mathbf{y}' &= \boldsymbol{\alpha}^\top \cdot \mathbf{V} - \boldsymbol{\alpha}'^\top \cdot \mathbf{V}' \\ &= \underbrace{(\boldsymbol{\alpha} - \boldsymbol{\alpha}')^\top \cdot \mathbf{V}}_{\text{weight change}} + \underbrace{\boldsymbol{\alpha}'^\top(\mathbf{V} - \mathbf{V}')}_{\text{value change}}. \end{aligned} \tag{38}$$

**Step 1: bounding the value-change term.** Since $\boldsymbol{\alpha}'$ is a probability vector,

$$
\begin{aligned}
\left\|\boldsymbol{\alpha}'^\top (\mathbf{V} - \mathbf{V}')\right\|_2 &= \left\|\sum_{i=1}^N \alpha_i' \cdot \left(\mathbf{V}_{i,:} - \mathbf{V}_{i,:}'\right)\right\|_2 \\
&\leq \sum_{i=1}^N \alpha_i' \cdot \left\|\mathbf{V}_{i,:} - \mathbf{V}_{i,:}'\right\|_2 \qquad \text{(triangle inequality)} \\
&\leq \max_{i \in [N]} \left\|\mathbf{V}_{i,:} - \mathbf{V}_{i,:}'\right\|_2 \\
&= \rho_V.
\end{aligned}
$$

**Step 2: bounding the weight-change term via an $\ell_1$ control.** Writing $(\boldsymbol{\alpha} - \boldsymbol{\alpha}')^\top \cdot \mathbf{V} = \sum_i (\alpha_i - \alpha_i') \cdot \mathbf{V}_{i,:}$, the triangle inequality gives

$$
\begin{aligned}
\left\|(\boldsymbol{\alpha} - \boldsymbol{\alpha}')^\top \cdot \mathbf{V}\right\|_2 &\leq \sum_{i=1}^N |\alpha_i - \alpha_i'| \cdot \|\mathbf{V}_{i,:}\|_2 \\
&\leq \left(\max_{i \in [N]} \|\mathbf{V}_{i,:}\|_2\right) \cdot \sum_{i=1}^N |\alpha_i - \alpha_i'| \\
&= V_{\max} \cdot \|\boldsymbol{\alpha} - \boldsymbol{\alpha}'\|_1.
\end{aligned}
\tag{39}
$$

**Step 3: Lipschitz control of the softmax weights by the logits.** By Lemma D.2 (softmax is $\ell_\infty \to \ell_1$ Lipschitz),

$$
\|\boldsymbol{\alpha} - \boldsymbol{\alpha}'\|_1 \leq \|\mathbf{s} - \mathbf{s}'\|_\infty.
\tag{40}
$$

It remains to bound $\|\mathbf{s} - \mathbf{s}'\|_\infty$ using the key perturbation.

For each $i \in [N]$, we have

$$
\begin{aligned}
|\mathbf{s}_i - \mathbf{s}_i'| &= \left| \frac{\langle \mathbf{q}, \mathbf{K}_{i,:} \rangle}{\sqrt{d_k}} - \frac{\langle \mathbf{q}, \mathbf{K}_{i,:}' \rangle}{\sqrt{d_k}} \right| \\
&= \frac{1}{\sqrt{d_k}} \cdot \left| \langle \mathbf{q}, \mathbf{K}_{i,:} - \mathbf{K}_{i,:}' \rangle \right| \\
&\leq \frac{1}{\sqrt{d_k}} \cdot \|\mathbf{q}\|_2 \cdot \|\mathbf{K}_{i,:} - \mathbf{K}_{i,:}'\|_2 \qquad \text{(Cauchy–Schwarz)} \\
&\leq \frac{1}{\sqrt{d_k}} \cdot \|\mathbf{Q}\|_{2,\infty} \cdot \|\mathbf{K} - \mathbf{K}'\|_{2,\infty} \\
&= \Gamma_Q^{(N)} \cdot \rho_K.
\end{aligned}
$$

Taking the maximum over $i$ yields

$$
\|\mathbf{s} - \mathbf{s}'\|_\infty \leq \Gamma_Q^{(N)} \cdot \rho_K.
\tag{41}
$$

Combining (39), (40), and (41) gives

$$
\left\|(\boldsymbol{\alpha} - \boldsymbol{\alpha}')^\top \cdot \mathbf{V}\right\|_2 \leq V_{\max} \cdot \Gamma_Q^{(N)} \cdot \rho_K.
\tag{42}
$$

**Step 4: concluding a per-row bound.** Applying the triangle inequality to (38) and using Step 1 and (42),

$$
\|\mathbf{y} - \mathbf{y}'\|_2 \leq \Gamma_Q^{(N)} \cdot \rho_K \cdot V_{\max} + \rho_V.
\tag{43}
$$

The bound is uniform in $j$.

**Step 5: lifting to the Frobenius norm.** Letting $\mathbf{Y} \triangleq \mathrm{Atten}(\mathbf{Q}; \mathbf{K}, \mathbf{V})$ and $\mathbf{Y}' \triangleq \mathrm{Atten}(\mathbf{Q}; \mathbf{K}', \mathbf{V}')$, (43) gives

$$
\begin{aligned}
\|\mathbf{Y} - \mathbf{Y}'\|_F^2 &= \sum_{j=1}^{N_q} \|\mathbf{Y}_{j,:} - \mathbf{Y}'_{j,:}\|_2^2 \\
&\leq \sum_{j=1}^{N_q} \left( \Gamma_Q^{(N)} \cdot \rho_K \cdot V_{\max} + \rho_V \right)^2 \\
&= N_q \cdot \left( \Gamma_Q^{(N)} \cdot \rho_K \cdot V_{\max} + \rho_V \right)^2.
\end{aligned}
$$

Taking square roots yields (37). $\qquad\square$

## F. A Forward-Pass Deterministic Comparator for Stage I

**Assumption F.1.** We assume keys and prototypes are $\ell_2$-normalized row-wise:

$$
\|\mathbf{K}_{i,:}\|_2 = 1 \text{ for all } i, \qquad \|\mathbf{P}_{j,:}\|_2 = 1 \text{ for all } j. \tag{44}
$$

$\triangle$

Let $M \in \mathbb{Z}_{>0}$ be a macro length with $M \ll N_k$, and let $\mathbf{P} \in \mathbb{R}^{M \times d_k}$ denote the prototypes. Stage I forms routing logits $\mathbf{S}$ and soft routing weights $\mathbf{A}$.

**Hard-routing comparator.** From one forward pass, we define the (deterministic) hard assignment

$$
c(i) \triangleq \arg\max_{j \in [M]} \mathbf{S}_{i,j}, \qquad i \in [N_k],
$$

and the associated hard routing matrix

$$
\mathbf{A}_{i,j}^{\mathrm{hard}} \triangleq \mathbf{1}\{j = c(i)\}, \qquad (i,j) \in [N_k] \times [M].
$$

We define cluster sizes

$$
n_j \triangleq \sum_{i=1}^{N_k} \mathbf{A}_{i,j}^{\mathrm{hard}}, \qquad j \in [M],
$$

and for each $j$ with $n_j > 0$ define the cluster means

$$
\bar{\mathbf{K}}_{j,:} \triangleq \frac{1}{n_j} \sum_{i:\, c(i)=j} \mathbf{K}_{i,:}, \qquad\qquad \bar{\mathbf{V}}_{j,:} \triangleq \frac{1}{n_j} \sum_{i:\, c(i)=j} \mathbf{V}_{i,:}, \tag{45}
$$

and set $\bar{\mathbf{K}}_{j,:} \triangleq \mathbf{0}$ and $\bar{\mathbf{V}}_{j,:} \triangleq \mathbf{0}$ when $n_j = 0$.

**Reconstruction operator.** We define the *hard-routing reconstruction operator* $\mathcal{R}_c : \mathbb{R}^{N_k \times d} \to \mathbb{R}^{N_k \times d}$ by

$$
\left( \mathcal{R}_c(\mathbf{X}) \right)_{i,:} \triangleq \begin{cases} \frac{1}{n_{c(i)}} \sum_{i':\, c(i')=c(i)} \mathbf{X}_{i',:}, & n_{c(i)} > 0, \\ \mathbf{0}, & n_{c(i)} = 0, \end{cases} \tag{46}
$$

where $i \in [N_k]$. With this notation the quantized (reconstructed) keys and values are

$$
\mathbf{K}^{\mathrm{q}} \triangleq \mathcal{R}_c(\mathbf{K}), \qquad \mathbf{V}^{\mathrm{q}} \triangleq \mathcal{R}_c(\mathbf{V}), \tag{47}
$$

i.e.,

$$
\mathbf{K}_{i,:}^{\mathrm{q}} \triangleq \bar{\mathbf{K}}_{c(i),:}, \qquad \mathbf{V}_{i,:}^{\mathrm{q}} \triangleq \bar{\mathbf{V}}_{c(i),:}.
$$

**Forward-pass radii.** We define the (deterministic) Stage I radii

$$\rho_K(M) \triangleq \|\mathbf{K} - \mathbf{K}^{\mathsf{q}}\|_{2,\infty}, \qquad \rho_V(M) \triangleq \|\mathbf{V} - \mathbf{V}^{\mathsf{q}}\|_{2,\infty}. \tag{48}$$

Let

$$\mathbf{Y}_{\mathsf{q}} \triangleq \mathrm{Atten}(\mathbf{Q}; \mathbf{K}^{\mathsf{q}}, \mathbf{V}^{\mathsf{q}}), . \tag{49}$$

The soft-routing output $\mathbf{Y}_{\mathrm{soft}} = \mathrm{Atten}(\mathbf{Q}; \mathbf{K}, \mathbf{V})$ is defined in (1).

**Lemma F.2** (Stage I comparator bound). *We define*

$$V_{\max} \triangleq \|\mathbf{V}\|_{2,\infty}, \qquad \Gamma_Q^{(N_k)} \triangleq \frac{\|\mathbf{Q}\|_{2,\infty}}{\sqrt{d_k}}.$$

*Then the end-to-end deviation between the original attention and the hard-routing comparator satisfies*

$$\|\mathbf{Y}_{\mathrm{soft}} - \mathbf{Y}_{\mathsf{q}}\|_F \le \sqrt{N_q} \cdot \left( \Gamma_Q^{(N_k)} \cdot \rho_K(M) \cdot V_{\max} + \rho_V(M) \right). \tag{50}$$

*Proof.* We apply Lemma E.1 to $(\mathbf{K}, \mathbf{V})$ and $(\mathbf{K}', \mathbf{V}') \triangleq (\mathbf{K}^{\mathsf{q}}, \mathbf{V}^{\mathsf{q}})$ with $N = N_k$. By (48), $\|\mathbf{K} - \mathbf{K}^{\mathsf{q}}\|_{2,\infty} = \rho_K(M)$ and $\|\mathbf{V} - \mathbf{V}^{\mathsf{q}}\|_{2,\infty} = \rho_V(M)$, which yields (50). $\square$

**Vanishing error as $M \to \infty$.** By Lemma F.2, $\rho_K(M) \to 0$ and $\rho_V(M) \to 0$ suffice; Theorem F.4 establishes such decay under a covering-number condition.

**Definition F.3** ($\epsilon$-net (covering number) in $\ell_2$). For a set $\mathcal{S} \subset \mathbb{R}^{d_k}$ and $\epsilon > 0$, an $\epsilon$-*net* is a finite set $\mathcal{N} \subset \mathbb{R}^{d_k}$ such that for every $\mathbf{x} \in \mathcal{S}$ there exists $\mathbf{p} \in \mathcal{N}$ with $\|\mathbf{x} - \mathbf{p}\|_2 \le \epsilon$. Let $\mathcal{N}(\mathcal{S}, \epsilon)$ denote the minimal cardinality of an $\epsilon$-net of $\mathcal{S}$. $\triangle$

**Theorem F.4** (A sufficient condition for $\rho_K(M) \to 0$ and a quantitative decay bound). *For a fixed forward pass, let $\mathcal{S}_K \triangleq \{\mathbf{K}_{i,:} : i \in [N_k]\} \subset \mathbb{R}^{d_k}$. We assume that the prototypes $\{\mathbf{P}_{j,:}\}_{j=1}^M$ form an $\epsilon$-net of $\mathcal{S}_K$ in $\ell_2$:*

$$\forall i \in [N_k] \, \exists j \in [M] \text{ s.t. } \|\mathbf{K}_{i,:} - \mathbf{P}_{j,:}\|_2 \le \epsilon. \tag{51}$$

*Let $c(i) \triangleq \arg\max_{j \in [M]} \langle \mathbf{K}_{i,:}, \mathbf{P}_{j,:} \rangle$ be the induced hard routing. Then the Stage I reconstruction error satisfies*

$$\rho_K(M) = \|\mathbf{K} - \mathcal{R}_c(\mathbf{K})\|_{2,\infty} \le 2\epsilon. \tag{52}$$

*In particular, if $M$ increases so that there exists $\epsilon_M \to 0$ with (51) holding for $\epsilon = \epsilon_M$, then $\rho_K(M) \to 0$.*

*Proof.* We bound each row error $\|\mathbf{K}_{i,:} - \mathbf{K}_{i,:}^{\mathsf{q}}\|_2$ and then take the maximum over $i$.

*Step 1 (hard routing picks a prototype that is still close).* For any $i \in [N_k]$, we write $\mathbf{k} \triangleq \mathbf{K}_{i,:}$. By (51), we pick $j^\star \in [M]$ such that $\|\mathbf{k} - \mathbf{P}_{j^\star,:}\|_2 \le \epsilon$. Since $c(i)$ maximizes the inner product with $\mathbf{k}$,

$$\langle \mathbf{k}, \mathbf{P}_{c(i),:} \rangle \ge \langle \mathbf{k}, \mathbf{P}_{j^\star,:} \rangle. \tag{53}$$

*Step 2 (inner product maximality implies a distance bound).* Using $\|\mathbf{a} - \mathbf{b}\|_2^2 = \|\mathbf{a}\|_2^2 + \|\mathbf{b}\|_2^2 - 2\langle \mathbf{a}, \mathbf{b} \rangle$,

$$\begin{aligned} \|\mathbf{k} - \mathbf{P}_{c(i),:}\|_2^2 &= \|\mathbf{k}\|_2^2 + \|\mathbf{P}_{c(i),:}\|_2^2 - 2\langle \mathbf{k}, \mathbf{P}_{c(i),:} \rangle \\ &\overset{(a)}{\le} \|\mathbf{k}\|_2^2 + \|\mathbf{P}_{c(i),:}\|_2^2 - 2\langle \mathbf{k}, \mathbf{P}_{j^\star,:} \rangle \\ &= \|\mathbf{k} - \mathbf{P}_{j^\star,:}\|_2^2 + \left( \|\mathbf{P}_{c(i),:}\|_2^2 - \|\mathbf{P}_{j^\star,:}\|_2^2 \right), \end{aligned}$$

where step $(a)$ is by (53). Under (44), the parenthetical term is zero, so

$$\|\mathbf{k} - \mathbf{P}_{c(i),:}\|_2^2 \le \|\mathbf{k} - \mathbf{P}_{j^\star,:}\|_2^2 \le \epsilon^2,$$

hence

$$\|\mathbf{K}_{i,:} - \mathbf{P}_{c(i),:}\|_2 \le \epsilon. \tag{54}$$

*Step 3 (keys routed to the same prototype are within $2\epsilon$ of each other).* For any $j \in [M]$ and any $i, i'$ with $c(i) = c(i') = j$, the triangle inequality and (54) give

$$\|\mathbf{K}_{i,:} - \mathbf{K}_{i',:}\|_2 \leq \|\mathbf{K}_{i,:} - \mathbf{P}_{j,:}\|_2 + \|\mathbf{K}_{i',:} - \mathbf{P}_{j,:}\|_2 \leq \epsilon + \epsilon = 2\epsilon. \tag{55}$$

*Step 4 (distance to the cluster mean is at most the cluster diameter).* For any $i$, we set $j \triangleq c(i)$. Then $n_j \geq 1$ and $\mathbf{K}_{i,:}^{\mathrm{q}} = \bar{\mathbf{K}}_{j,:}$. Using (45) and the triangle inequality,

$$
\begin{aligned}
\|\mathbf{K}_{i,:} - \bar{\mathbf{K}}_{j,:}\|_2 &= \left\| \frac{1}{n_j} \sum_{i':\, c(i')=j} (\mathbf{K}_{i,:} - \mathbf{K}_{i',:}) \right\|_2 \\
&\leq \frac{1}{n_j} \sum_{i':\, c(i')=j} \|\mathbf{K}_{i,:} - \mathbf{K}_{i',:}\|_2 \\
&\leq \frac{1}{n_j} \sum_{i':\, c(i')=j} 2\epsilon \\
&= 2\epsilon,
\end{aligned}
$$

where the last inequality uses (55). Thus, for every $i \in [N_k]$,

$$\|\mathbf{K}_{i,:} - \mathbf{K}_{i,:}^{\mathrm{q}}\|_2 = \|\mathbf{K}_{i,:} - \bar{\mathbf{K}}_{c(i),:}\|_2 \leq 2\epsilon.$$

*Step 5 (taking the maximum over rows).* Taking the maximum over $i$ gives

$$\rho_K(M) = \|\mathbf{K} - \mathcal{R}_c(\mathbf{K})\|_{2,\infty} = \max_{i \in [N_k]} \|\mathbf{K}_{i,:} - \mathbf{K}_{i,:}^{\mathrm{q}}\|_2 \leq 2\epsilon,$$

which is (52). $\qquad\square$

**Decay rate via covering numbers.** Let $\mathbb{S}^{d_k-1} \triangleq \{\mathbf{x} \in \mathbb{R}^{d_k} : \|\mathbf{x}\|_2 = 1\}$; under (44), $\mathcal{S}_K \subset \mathbb{S}^{d_k-1}$. A standard volumetric argument on the sphere gives

$$\mathcal{N}(\mathbb{S}^{d_k-1}, \epsilon) \leq \left(\frac{3}{\epsilon}\right)^{d_k}, \qquad \epsilon \in (0,1),$$

see, e.g., Vershynin (2018, Ch. 4). Therefore, if

$$M \geq \left(\frac{3}{\epsilon}\right)^{d_k},$$

then there exists a set of $M$ unit-norm prototypes that forms an $\epsilon$-net of $\mathbb{S}^{d_k-1}$, and hence can satisfy (51) for any $\mathcal{S}_K \subset \mathbb{S}^{d_k-1}$. Equivalently, setting $\epsilon_M \triangleq 3 \cdot M^{-1/d_k}$ gives

$$\rho_K(M) \leq 2\epsilon_M = 6 \cdot M^{-1/d_k}$$

by Theorem F.4.

*Constructions of prototypes for the finite set $\mathcal{S}_K$.* For the finite set $\mathcal{S}_K = \{\mathbf{K}_{i,:}\}_{i=1}^{N_k}$, three standard constructions provide $M$ representatives minimizing $\max_i \min_j \|\mathbf{K}_{i,:} - \mathbf{P}_{j,:}\|_2$:

- **$k$-center / farthest-first traversal (greedy $\epsilon$-net).** This construction starts from any key as the first prototype and repeatedly adds the key farthest (in $\ell_2$) from the current prototype set until $M$ prototypes are chosen. It is the classic greedy algorithm for the metric $k$-center problem and yields a constant-factor approximation to the optimal worst-case radius (Gonzalez, 1985). In particular, the resulting prototypes directly control $\max_i \min_j \|\mathbf{K}_{i,:} - \mathbf{P}_{j,:}\|_2$, which is exactly the $\epsilon$ appearing in (51).
- **$k$-means on normalized keys (Lloyd updates).** Lloyd's algorithm (Lloyd, 1982) runs $k$-means with $k = M$ on $\{\mathbf{K}_{i,:}\}_{i=1}^{N_k}$, optionally with $k$-means++ initialization (Arthur & Vassilvitskii, 2007), and the resulting centroids are normalized to unit norm to satisfy (44).

- **Learned prototypes (default in PLASH).** Here $\mathbf{P}$ is a set of trainable parameters updated by backpropagation. End-to-end training drives the prototypes toward regions of high key density (as in learned vector quantization (van den Oord et al., 2017)); the forward-pass quantity $\rho_K(M)$ in (48) measures the current coverage of $\mathcal{S}_K$.

*Implication for the Stage I comparator error.* Substituting $\rho_K(M) \leq 6 \cdot M^{-1/d_k}$ into Lemma F.2 yields

$$\|\mathbf{Y}_{\text{soft}} - \mathbf{Y}_{\text{q}}\|_F \leq \sqrt{N_q} \cdot \left( \Gamma_Q^{(N_k)} \cdot \left( 6 \cdot M^{-1/d_k} \right) \cdot V_{\max} + \rho_V(M) \right),$$

where $\rho_V(M)$ admits an analogous bound under the same type of covering condition in $\mathbb{R}^{d_v}$.

## G. Proof of Theorem 3.1: CountSketch as the Only Source of Randomness at Stage II

**Stage II inputs.** Stage II takes the normalized features $\widetilde{\mathbf{G}} \in \mathbb{R}^{M \times d'}$ from (7); each row $\widetilde{\mathbf{G}}_{j,:}$ is embedded into $\mathbb{R}^{D_k}$ via deterministic zero-padding when $D_k \geq d'$.

**Notation.** We consider arbitrary $D \in \mathbb{Z}_{>0}$ and write $[D] \triangleq \{0, 1, \ldots, D-1\}$. For $\mathbf{a}, \mathbf{b} \in \mathbb{R}^D$, the *circular convolution* $\mathbf{a} * \mathbf{b} \in \mathbb{R}^D$ is defined componentwise by

$$\left( \mathbf{a} * \mathbf{b} \right)_t \triangleq \sum_{s \in [D]} \mathbf{a}_s \cdot \mathbf{b}_{(t-s) \bmod D}, \qquad t \in [D].$$

For an integer $k \geq 1$ and $\mathbf{x} \in \mathbb{R}^D$, the $k$-fold circular convolution power is

$$\mathbf{x}^{(*k)} \triangleq \underbrace{\mathbf{x} * \mathbf{x} * \cdots * \mathbf{x}}_{k \text{ times}} \in \mathbb{R}^D.$$

**Implemented Stage II features and embeddings (random).** The implemented Stage II enrichment uses TensorSketch. For each macro index $j \in [M]$ and degree $k \in \mathcal{K}$, the sketched feature is

$$\mathbf{z}_{j,k}^{\text{ts}} \triangleq \text{TS}_k(\widetilde{\mathbf{G}}_{j,:}; D_k) \in \mathbb{R}^{D_k}. \tag{56}$$

The degree-mixture feature and Stage II embedding are

$$\mathbf{v}_j^{\text{ts}} \triangleq \bigoplus_{k \in \mathcal{K}} \beta_k \cdot \mathbf{z}_{j,k}^{\text{ts}} \in \mathbb{R}^{D_{\text{tot}}}, \quad (\mathbf{Y}_{\text{enh}})_{j,:} = \mathbf{W}_{\text{out}} \cdot \mathbf{v}_j^{\text{ts}} \in \mathbb{R}^d, \tag{57}$$

where $D_{\text{tot}}, \mathbf{Y}_{\text{enh}}$, and $\mathbf{W}_{\text{out}}$ are defined in (9) and (10), respectively. In this construction, all randomness enters exclusively through the TensorSketch map $\text{TS}_k$ in (56).

**Deterministic Stage II reference (no randomness).** We introduce a deterministic reference that keeps the interface $(\mathcal{K}, \{\beta_k\}_{k \in \mathcal{K}}, \{D_k\}_{k \in \mathcal{K}}, \mathbf{W}_{\text{out}})$ unchanged and replaces TensorSketch by exact $k$-fold circular convolution on $[D_k]$. When $\widetilde{\mathbf{G}}_{j,:} \in \mathbb{R}^{d'}$ with $d' \leq D_k$ we zero-pad to length $D_k$; otherwise the convolution acts directly on length $D_k$:

$$\mathbf{z}_{j,k}^{\text{det}} \triangleq \underbrace{\widetilde{\mathbf{G}}_{j,:} * \widetilde{\mathbf{G}}_{j,:} * \cdots * \widetilde{\mathbf{G}}_{j,:}}_{k \text{ times}} \in \mathbb{R}^{D_k}. \tag{58}$$

Equivalently, $\mathbf{z}_{j,k}^{\text{det}} = \text{IFFT}\left( \text{FFT}(\widetilde{\mathbf{G}}_{j,:})^{\odot k} \right)$ on length $D_k$. The associated degree-mixture feature is

$$\mathbf{v}_j^{\text{det}} \triangleq \bigoplus_{k \in \mathcal{K}} (\beta_k \cdot \mathbf{z}_{j,k}^{\text{det}}) \in \mathbb{R}^{D_{\text{tot}}}, \tag{59}$$

and the deterministic Stage II embedding is

$$(\mathbf{Y}_{\text{enh}}^{\text{det}})_{j,:} \triangleq \mathbf{W}_{\text{out}} \cdot \mathbf{v}_j^{\text{det}} \in \mathbb{R}^d. \tag{60}$$

Applying the same deterministic mixer / Stage III composition $\mathcal{A}_{\mathbf{Q}}$ (defined by (11) and (12)) to $\mathbf{Y}_{\text{enh}}^{\text{det}}$ yields the reference output $\mathbf{Y}^{\text{det}} \triangleq \mathcal{A}_{\mathbf{Q}}(\mathbf{Y}_{\text{enh}}^{\text{det}})$.

## G.1. Stage II Error Certificate: TensorSketch vs. Deterministic Convolution

**Lemma G.1** (Norm growth under repeated circular convolution). *Let $D \in \mathbb{Z}_{>0}$ and let $*$ denote circular convolution on $\mathbb{R}^D$. For any $\mathbf{x} \in \mathbb{R}^D$ and any integer $k \geq 1$,*

$$\|\mathbf{x}^{(*k)}\|_2 \; \leq \; D^{\frac{k-1}{2}} \cdot \|\mathbf{x}\|_2^k. \tag{61}$$

*Proof.* **Step 1: A Young-type inequality.** By the discrete Young convolution inequality on $\mathbb{R}^D$, for any $\mathbf{a}, \mathbf{b} \in \mathbb{R}^D$,

$$\|\mathbf{a} * \mathbf{b}\|_2 \; \leq \; \|\mathbf{a}\|_1 \cdot \|\mathbf{b}\|_2. \tag{62}$$

**Step 2: Bounding $\ell_1$ by $\ell_2$ on $\mathbb{R}^D$.** By Cauchy–Schwarz,

$$\|\mathbf{a}\|_1 \; = \; \sum_{t \in [D]} |\mathbf{a}_t| \; \leq \; \sqrt{D} \cdot \left( \sum_{t \in [D]} |\mathbf{a}_t|^2 \right)^{1/2} \; = \; \sqrt{D} \cdot \|\mathbf{a}\|_2. \tag{63}$$

**Step 3: Iterating the one-step bound.** We define $\mathbf{x}^{(*1)} \triangleq \mathbf{x}$ and $\mathbf{x}^{(*(m+1))} \triangleq \mathbf{x}^{(*m)} * \mathbf{x}$. Substituting $\mathbf{a} = \mathbf{x}^{(*m)}$ and $\mathbf{b} = \mathbf{x}$ into (62) and (63):

$$\|\mathbf{x}^{(*(m+1))}\|_2 = \|\mathbf{x}^{(*m)} * \mathbf{x}\|_2 \leq \|\mathbf{x}^{(*m)}\|_1 \|\mathbf{x}\|_2 \leq \sqrt{D} \cdot \|\mathbf{x}^{(*m)}\|_2 \cdot \|\mathbf{x}\|_2.$$

Applying this inequality for $m = 1, 2, \ldots, k-1$:

$$\|\mathbf{x}^{(*k)}\|_2 \leq (\sqrt{D})^{k-1} \|\mathbf{x}\|_2^k = D^{\frac{k-1}{2}} \|\mathbf{x}\|_2^k,$$

which is (61). $\qquad \square$

**Corollary G.2** (Uniform deterministic bound for $\|\mathbf{z}_{j,k}^{\mathrm{det}}\|_2$). *By the Stage II norm control (7), $\|\widetilde{\mathbf{G}}_{j,:}\|_2 \leq 1/\tau_{\mathrm{g}}$ for all $j \in [M]$. Then for all $j \in [M]$ and $k \in \mathcal{K}$,*

$$\|\mathbf{z}_{j,k}^{\mathrm{det}}\|_2 \; \leq \; D_k^{\frac{k-1}{2}} \cdot \tau_{\mathrm{g}}^{-k}. \tag{64}$$

*Proof.* For any $j \in [M]$ and $k \in \mathcal{K}$, by (58), $\mathbf{z}_{j,k}^{\mathrm{det}} = \widetilde{\mathbf{G}}_{j,:}^{(*k)}$ in $\mathbb{R}^{D_k}$. Applying Lemma G.1 with $D = D_k$ and $\mathbf{x} = \widetilde{\mathbf{G}}_{j,:}$:

$$\|\mathbf{z}_{j,k}^{\mathrm{det}}\|_2 = \|\widetilde{\mathbf{G}}_{j,:}^{(*k)}\|_2 \leq D_k^{\frac{k-1}{2}} \|\widetilde{\mathbf{G}}_{j,:}\|_2^k.$$

By (7), $\|\widetilde{\mathbf{G}}_{j,:}\|_2 \leq 1/\tau_{\mathrm{g}}$, yielding (64). $\qquad \square$

**Lemma G.3** (Uniform high-probability norm bound for TensorSketch features). *Let $\eta, \delta \in (0,1)$. We adopt the Stage II norm control in (7) and Assumption C.2. The per-degree budgets $\{\delta_k\}_{k \in \mathcal{K}} \subset (0,1)$ are chosen such that $\sum_{k \in \mathcal{K}} \delta_k \leq \delta$. If*

$$D_k \; \geq \; \frac{2 C_k M}{\eta^2 \cdot \delta_k}, \qquad \forall k \in \mathcal{K}, \tag{65}$$

*then with probability at least $1 - \delta$, simultaneously for all $j \in [M]$ and $k \in \mathcal{K}$,*

$$\|\mathbf{z}_{j,k}^{\mathrm{ts}}\|_2 \; \leq \; \sqrt{1+\eta} \cdot \|\widetilde{\mathbf{G}}_{j,:}\|_2^k \; \leq \; \sqrt{1+\eta} \cdot \tau_{\mathrm{g}}^{-k}. \tag{66}$$

*Proof.* For any pair $(j, k)$, we define $\mathbf{x} \triangleq \widetilde{\mathbf{G}}_{j,:}$ and $D \triangleq D_k$. Let

$$Z \; \triangleq \; \|\mathrm{TS}_k(\mathbf{x}; D)\|_2^2.$$

Under Assumption C.2, applying Proposition C.4 with $\mathbf{y} = \mathbf{x}$ (the self inner-product) gives the moment bounds

$$\mathbb{E}(Z) = \|\mathbf{x}\|_2^{2k}, \qquad \mathrm{Var}(Z) \leq \frac{2 C_k}{D} \cdot \|\mathbf{x}\|_2^{4k}.$$

By Chebyshev's inequality (Tchébychef, 1867),

$$
\begin{aligned}
\Pr\big(Z > (1+\eta)\|\mathbf{x}\|_2^{2k}\big) = \Pr\big(Z - \mathbb{E}(Z) > \eta\|\mathbf{x}\|_2^{2k}\big) \\
\leq \frac{\mathrm{Var}(Z)}{\eta^2\|\mathbf{x}\|_2^{4k}} \\
\leq \frac{2C_k}{D\eta^2}.
\end{aligned}
$$

The per-pair failure budget is $\delta'_{j,k} \triangleq \delta_k/M$, allocating $\delta_k$ over the $M$ row indices. Under (65), we have $D_k \geq 2C_k/(\eta^2\delta'_{j,k})$, hence

$$
\Pr\big(Z > (1+\eta)\|\mathbf{x}\|_2^{2k}\big) \leq \delta'_{j,k}.
$$

Therefore, with probability at least $1 - \delta'_{j,k}$,

$$
\|\mathrm{TS}_k(\mathbf{x}; D_k)\|_2 \leq \sqrt{1+\eta} \cdot \|\mathbf{x}\|_2^k.
$$

A union bound over all $M \cdot |\mathcal{K}|$ events (one per pair $(j,k)$) yields total failure probability at most $\sum_{j\in[M],k\in\mathcal{K}} \delta'_{j,k} = \sum_{k\in\mathcal{K}} \delta_k \leq \delta$ by the assumption on the budget allocation, proving the first inequality in (66). Finally, (7) implies $\|\mathbf{x}\|_2 \leq 1/\tau_{\mathrm{g}}$, giving the second inequality. $\qquad\square$

## G.2. A Complete Stage II Feature Discrepancy Bound (No Difference Computation)

**Theorem G.4** (Uniform high-probability bound for $\|\mathbf{z}^{\mathrm{ts}}_{j,k} - \mathbf{z}^{\mathrm{det}}_{j,k}\|_2$). *Let $\eta, \delta \in (0,1)$, $M \in \mathbb{Z}_{>0}$, finite $\mathcal{K} \subset \mathbb{Z}_{\geq 1}$, and $\tau_{\mathrm{g}} \in \mathbb{R}_{>0}$. The normalized macro features $\widetilde{\mathbf{G}}$ are those of (7), and Assumption C.2 holds. For each $k \in \mathcal{K}$, let $\mathbf{z}^{\mathrm{ts}}_{j,k} \in \mathbb{R}^{D_k}$ denote the degree-$k$ TensorSketch feature (Definition C.1) and let $\mathbf{z}^{\mathrm{det}}_{j,k} \in \mathbb{R}^{D_k}$ denote the degree-$k$ deterministic convolutional feature (58). The per-degree failure budgets $\{\delta_k\}_{k\in\mathcal{K}} \subset (0,1)$ satisfy $\sum_{k\in\mathcal{K}} \delta_k \leq \delta$, and the sketch sizes $\{D_k\}_{k\in\mathcal{K}} \subset \mathbb{Z}_{>0}$ satisfy (65). Then, with probability at least $1 - \delta$, simultaneously for all $j \in [M]$ and $k \in \mathcal{K}$,*

$$
\big\|\mathbf{z}^{\mathrm{ts}}_{j,k} - \mathbf{z}^{\mathrm{det}}_{j,k}\big\|_2 \leq \left(\sqrt{1+\eta} + D_k^{\frac{k-1}{2}}\right) \cdot \tau_{\mathrm{g}}^{-k}. \tag{67}
$$

*In particular, for any target tolerance $\epsilon_{\mathrm{feat}} > 0$, the deterministic choice*

$$
\tau_{\mathrm{g}} \geq \max_{k\in\mathcal{K}}\left(\left(\frac{\sqrt{1+\eta} + (D_k)^{(k-1)/2}}{\epsilon_{\mathrm{feat}}}\right)^{1/k}\right) \tag{68}
$$

*implies $\big\|\mathbf{z}^{\mathrm{ts}}_{j,k} - \mathbf{z}^{\mathrm{det}}_{j,k}\big\|_2 \leq \epsilon_{\mathrm{feat}}$ uniformly over all $(j,k)$ on the same probability-$(1-\delta)$ event.*

*Proof.* We condition on the uniform TensorSketch norm event $\mathcal{E}_{\mathrm{ts}}$ from Lemma G.3, then apply the triangle inequality.

**Step 1: defining the high-probability TensorSketch norm event.** Let

$$
\mathcal{E}_{\mathrm{ts}} \triangleq \left\{\forall j \in [M],\ \forall k \in \mathcal{K}:\ \big\|\mathbf{z}^{\mathrm{ts}}_{j,k}\big\|_2 \leq \sqrt{1+\eta} \cdot \tau_{\mathrm{g}}^{-k}\right\}.
$$

From Lemma G.3 with $D_k \geq 2C_k M/(\eta^2\delta_k)$, $\Pr(\mathcal{E}_{\mathrm{ts}}) \geq 1 - \delta$.

**Step 2: using the deterministic bound for the reference features.** From Corollary G.2, for all $j \in [M]$ and $k \in \mathcal{K}$,

$$
\big\|\mathbf{z}^{\mathrm{det}}_{j,k}\big\|_2 \leq D_k^{\frac{k-1}{2}} \cdot \tau_{\mathrm{g}}^{-k}.
$$

**Step 3: combining via the triangle inequality.** On the event $\mathcal{E}_{\mathrm{ts}}$, for any $(j,k)$,

$$
\begin{aligned}
\big\|\mathbf{z}^{\mathrm{ts}}_{j,k} - \mathbf{z}^{\mathrm{det}}_{j,k}\big\|_2 &\leq \big\|\mathbf{z}^{\mathrm{ts}}_{j,k}\big\|_2 + \big\|\mathbf{z}^{\mathrm{det}}_{j,k}\big\|_2 \\
&\leq \sqrt{1+\eta} \cdot \tau_{\mathrm{g}}^{-k} + D_k^{\frac{k-1}{2}} \cdot \tau_{\mathrm{g}}^{-k} \\
&= \left(\sqrt{1+\eta} + D_k^{\frac{k-1}{2}}\right) \cdot \tau_{\mathrm{g}}^{-k},
\end{aligned}
$$

which is (67), holding simultaneously over all $(j, k)$ with probability $\geq 1 - \delta$.

**Step 4: deriving the sufficient choice of $\tau_{\mathrm{g}}$.** The bound (67) is at most $\epsilon_{\mathrm{feat}}$ for all $k \in \mathcal{K}$ if and only if, for each $k$,

$$\tau_{\mathrm{g}} \geq \left( \frac{\sqrt{1 + \eta} + (D_k)^{(k-1)/2}}{\epsilon_{\mathrm{feat}}} \right)^{1/k}.$$

Taking the maximum over $k \in \mathcal{K}$ yields (68). $\qquad \square$

**Corollary G.5** (Stage II macro-embedding certificate). *Under the conditions of Theorem G.4, with probability at least $1 - \delta$,*

$$\left\| \mathbf{Y}_{\mathrm{enh}} - \mathbf{Y}_{\mathrm{enh}}^{\mathrm{det}} \right\|_{2,\infty} \leq \| \mathbf{W}_{\mathrm{out}} \|_{\mathrm{op}} \cdot \left( \sum_{k \in \mathcal{K}} \beta_k^2 \cdot \left( \left( \sqrt{1 + \eta} + D_k^{\frac{k-1}{2}} \right) \cdot \tau_{\mathrm{g}}^{-k} \right)^2 \right)^{1/2}. \tag{69}$$

*Proof.* **Step 1: decomposing the embedding difference via degree-mixture vectors.** By construction,

$$(\mathbf{Y}_{\mathrm{enh}})_{j,:} = \mathbf{W}_{\mathrm{out}} \cdot \mathbf{v}_j^{\mathrm{ts}}, \qquad (\mathbf{Y}_{\mathrm{enh}}^{\mathrm{det}})_{j,:} = \mathbf{W}_{\mathrm{out}} \cdot \mathbf{v}_j^{\mathrm{det}},$$

where

$$\mathbf{v}_j^{\mathrm{ts}} = \bigoplus_{k \in \mathcal{K}} (\beta_k \cdot \mathbf{z}_{j,k}^{\mathrm{ts}}), \qquad \mathbf{v}_j^{\mathrm{det}} = \bigoplus_{k \in \mathcal{K}} (\beta_k \cdot \mathbf{z}_{j,k}^{\mathrm{det}}).$$

For each row $j \in [M]$:

$$(\mathbf{Y}_{\mathrm{enh}})_{j,:} - (\mathbf{Y}_{\mathrm{enh}}^{\mathrm{det}})_{j,:} = \mathbf{W}_{\mathrm{out}} \left( \mathbf{v}_j^{\mathrm{ts}} - \mathbf{v}_j^{\mathrm{det}} \right).$$

**Step 2: applying the operator-norm bound.** The operator norm of $\mathbf{W}_{\mathrm{out}}$ gives

$$\left\| (\mathbf{Y}_{\mathrm{enh}})_{j,:} - (\mathbf{Y}_{\mathrm{enh}}^{\mathrm{det}})_{j,:} \right\|_2 \leq \| \mathbf{W}_{\mathrm{out}} \|_{\mathrm{op}} \cdot \left\| \mathbf{v}_j^{\mathrm{ts}} - \mathbf{v}_j^{\mathrm{det}} \right\|_2.$$

**Step 3: Pythagoras on disjoint coordinate blocks.** Since $\mathbf{v}_j^{\mathrm{ts}} - \mathbf{v}_j^{\mathrm{det}} = \bigoplus_{k \in \mathcal{K}} \beta_k (\mathbf{z}_{j,k}^{\mathrm{ts}} - \mathbf{z}_{j,k}^{\mathrm{det}})$,

$$\left\| \mathbf{v}_j^{\mathrm{ts}} - \mathbf{v}_j^{\mathrm{det}} \right\|_2^2 = \sum_{k \in \mathcal{K}} \beta_k^2 \cdot \left\| \mathbf{z}_{j,k}^{\mathrm{ts}} - \mathbf{z}_{j,k}^{\mathrm{det}} \right\|_2^2.$$

**Step 4: substituting the uniform feature discrepancy bound.** On the probability-$(1 - \delta)$ event of Theorem G.4, for all $j$ and $k$,

$$\left\| \mathbf{z}_{j,k}^{\mathrm{ts}} - \mathbf{z}_{j,k}^{\mathrm{det}} \right\|_2 \leq \left( \sqrt{1 + \eta} + D_k^{\frac{k-1}{2}} \right) \cdot \tau_{\mathrm{g}}^{-k}.$$

Substituting into Step 3, we take square roots and then take the maximum over $j \in [M]$, which yields (69). $\qquad \square$

# H. Proof of Theorem 3.2: An End-to-End Approximation Error Analysis and a Forward-Pass Checkable Sufficient Condition

Stage II is the only source of algorithmic randomness in PLASH; Stages I and III are deterministic given inputs and parameters. The proof of Theorem 3.2 controls the Stage II approximation error and propagates it to the final output via deterministic stability of the post-processing pipeline.

**Notation and norms.** For a matrix $\mathbf{A} \in \mathbb{R}^{n \times m}$ we write $\| \mathbf{A} \|_{2,\infty} \triangleq \max_{i \in [n]} \| \mathbf{A}_{i,:} \|_2$ and $\| \cdot \|_F$ for the Frobenius norm. We use $\| \cdot \|_{\mathrm{op}}$ for the spectral operator norm (Definition A.3).

### H.1. A Deterministic View of the Mixer and Stage III

**Deterministic post-processing map.** For a fixed forward pass and queries $\mathbf{Q}$, we define the deterministic map

$$\mathcal{T}_{\mathbf{Q}} : \mathbb{R}^{M \times d} \to \mathbb{R}^{N_q \times d_v}$$

as the composition of (i) the macro mixer of Definition A.7, (ii) linear projections to macro keys and values, and (iii) the exact softmax readout from queries to the compressed keys / values (cf. Section 2.2.3 and (11)–(12)):

$$\mathcal{T}_{\mathbf{Q}}(\mathbf{U}) \triangleq \mathrm{Atten}\left(\mathbf{Q};\ \left(\mathrm{Mixer}_{L_{\mathrm{mix}}}(\mathbf{U})\right) \cdot \mathbf{W}_K,\ \left(\mathrm{Mixer}_{L_{\mathrm{mix}}}(\mathbf{U})\right) \cdot \mathbf{W}_V\right).$$

Accordingly,

$$\mathbf{Y}_{\mathrm{PLASH}} \triangleq \mathcal{T}_{\mathbf{Q}}(\mathbf{Y}_{\mathrm{enh}}), \qquad \mathbf{Y}_{\mathrm{PLASH}}^{\mathrm{det}} \triangleq \mathcal{T}_{\mathbf{Q}}(\mathbf{Y}_{\mathrm{enh}}^{\mathrm{det}}),$$

where $\mathbf{Y}_{\mathrm{enh}}$ is the implemented Stage II embedding (57) and $\mathbf{Y}_{\mathrm{enh}}^{\mathrm{det}}$ is the deterministic Stage II comparator (60).

**Lemma H.1** (Deterministic stability of the Mixer and Stage III post-processing pipeline). *For a fixed forward pass and queries $\mathbf{Q}$, let the segment*

$$\mathcal{S} \triangleq \left\{\mathbf{Y}_{\mathrm{enh}}^{\mathrm{det}} + t \cdot \left(\mathbf{Y}_{\mathrm{enh}} - \mathbf{Y}_{\mathrm{enh}}^{\mathrm{det}}\right) :\ t \in [0,1]\right\} \subseteq \mathbb{R}^{M \times d} \tag{70}$$

*connect $\mathbf{Y}_{\mathrm{enh}}^{\mathrm{det}}$ and $\mathbf{Y}_{\mathrm{enh}}$. Theorem D.5 (with the forward-pass-computable upper bound from Theorem D.8) gives, on $\mathcal{S}$, a deterministic constant $L_{\mathrm{mix}} > 0$ such that for all $\mathbf{U}, \mathbf{U}' \in \mathcal{S}$,*

$$\left\|\mathrm{Mixer}_{L_{\mathrm{mix}}}(\mathbf{U}) - \mathrm{Mixer}_{L_{\mathrm{mix}}}(\mathbf{U}')\right\|_{2,\infty} \leq L_{\mathrm{mix}} \cdot \|\mathbf{U} - \mathbf{U}'\|_{2,\infty}. \tag{71}$$

*We define*

$$\begin{aligned}
L_K &\triangleq \|\mathbf{W}_K\|_{\mathrm{op}}, \\
L_V &\triangleq \|\mathbf{W}_V\|_{\mathrm{op}}, \\
\Gamma_Q^{(M)} &\triangleq \frac{1}{\sqrt{d_k}} \cdot \|\mathbf{Q}\|_{2,\infty},
\end{aligned} \tag{72}$$

*and let*

$$\Gamma_V(\mathcal{S}) \triangleq \sup_{\mathbf{U} \in \mathcal{S}} \left\|\mathbf{V}_g(\mathbf{U})\right\|_{2,\infty}, \tag{73}$$

*where $\mathbf{V}_g(\mathbf{U})$ denotes the Stage III macro-value map*

$$\mathbf{V}_g(\mathbf{U}) \triangleq \left(\mathrm{Mixer}_{L_{\mathrm{mix}}}(\mathbf{U})\right) \cdot \mathbf{W}_V.$$

*Then for all $\mathbf{U}, \mathbf{U}' \in \mathcal{S}$,*

$$\left\|\mathcal{T}_{\mathbf{Q}}(\mathbf{U}) - \mathcal{T}_{\mathbf{Q}}(\mathbf{U}')\right\|_F \leq \sqrt{N_q} \cdot L_{\mathrm{post}}(\mathcal{S}) \cdot \|\mathbf{U} - \mathbf{U}'\|_{2,\infty}, \tag{74}$$

*where*

$$L_{\mathrm{post}}(\mathcal{S}) \triangleq L_{\mathrm{mix}} \cdot \left(\Gamma_Q^{(M)} \cdot L_K \cdot \Gamma_V(\mathcal{S}) + L_V\right). \tag{75}$$

*In particular, taking $(\mathbf{U}, \mathbf{U}') = \left(\mathbf{Y}_{\mathrm{enh}}, \mathbf{Y}_{\mathrm{enh}}^{\mathrm{det}}\right)$ yields*

$$\left\|\mathbf{Y}_{\mathrm{PLASH}} - \mathbf{Y}_{\mathrm{PLASH}}^{\mathrm{det}}\right\|_F \leq \sqrt{N_q} \cdot L_{\mathrm{post}}(\mathcal{S}) \cdot \left\|\mathbf{Y}_{\mathrm{enh}} - \mathbf{Y}_{\mathrm{enh}}^{\mathrm{det}}\right\|_{2,\infty}. \tag{76}$$

*Proof.* For $\mathbf{U}, \mathbf{U}' \in \mathcal{S}$, we abbreviate

$$\mathbf{Z} \triangleq \mathrm{Mixer}_{L_{\mathrm{mix}}}(\mathbf{U}), \qquad \mathbf{Z}' \triangleq \mathrm{Mixer}_{L_{\mathrm{mix}}}(\mathbf{U}').$$

**Step 1: stability of the mixer.** By the local Lipschitz property (71),

$$\|\mathbf{Z} - \mathbf{Z}'\|_{2,\infty} \leq L_{\mathrm{mix}} \cdot \|\mathbf{U} - \mathbf{U}'\|_{2,\infty}. \tag{77}$$

**Step 2: stability of linear projections.** We define the induced compressed keys and values

$$\mathbf{K}_g(\mathbf{U}) \triangleq \mathbf{Z} \cdot \mathbf{W}_K, \qquad \mathbf{V}_g(\mathbf{U}) \triangleq \mathbf{Z} \cdot \mathbf{W}_V, \tag{78}$$

and analogously $\mathbf{K}_g(\mathbf{U}'), \mathbf{V}_g(\mathbf{U}')$ using $\mathbf{Z}'$. By Lemma D.1 (right-multiplication stability in $\|\cdot\|_{2,\infty}$),

$$\left\|\mathbf{K}_g(\mathbf{U}) - \mathbf{K}_g(\mathbf{U}')\right\|_{2,\infty} \leq L_K \cdot \|\mathbf{Z} - \mathbf{Z}'\|_{2,\infty}, \tag{79}$$

$$\left\|\mathbf{V}_g(\mathbf{U}) - \mathbf{V}_g(\mathbf{U}')\right\|_{2,\infty} \leq L_V \cdot \|\mathbf{Z} - \mathbf{Z}'\|_{2,\infty}. \tag{80}$$

**Step 3: stability of attention with respect to key / value perturbations.** Applying Lemma E.1 with $N = M$ and the identification

$$(\mathbf{K}, \mathbf{V}) \triangleq \left(\mathbf{K}_g(\mathbf{U}), \mathbf{V}_g(\mathbf{U})\right), \qquad (\mathbf{K}', \mathbf{V}') \triangleq \left(\mathbf{K}_g(\mathbf{U}'), \mathbf{V}_g(\mathbf{U}')\right).$$

This gives

$$\left\|\mathcal{T}_{\mathbf{Q}}(\mathbf{U}) - \mathcal{T}_{\mathbf{Q}}(\mathbf{U}')\right\|_F \leq \sqrt{N_q} \cdot \left(\Gamma_Q^{(M)} \cdot \left\|\mathbf{K}_g(\mathbf{U}) - \mathbf{K}_g(\mathbf{U}')\right\|_{2,\infty} \cdot \left\|\mathbf{V}_g(\mathbf{U})\right\|_{2,\infty} + \left\|\mathbf{V}_g(\mathbf{U}) - \mathbf{V}_g(\mathbf{U}')\right\|_{2,\infty}\right). \tag{81}$$

**Step 4: uniform control of values on the segment.** By definition (73), for all $\mathbf{U} \in \mathcal{S}$,

$$\left\|\mathbf{V}_g(\mathbf{U})\right\|_{2,\infty} \leq \Gamma_V(\mathcal{S}). \tag{82}$$

**Step 5: combining the bounds.** Substituting (82), (79), and (80) into (81) yields

$$\left\|\mathcal{T}_{\mathbf{Q}}(\mathbf{U}) - \mathcal{T}_{\mathbf{Q}}(\mathbf{U}')\right\|_F \leq \sqrt{N_q} \cdot \left(\Gamma_Q^{(M)} \cdot L_K \cdot \Gamma_V(\mathcal{S}) + L_V\right) \cdot \|\mathbf{Z} - \mathbf{Z}'\|_{2,\infty}.$$

Substituting (77) gives

$$\left\|\mathcal{T}_{\mathbf{Q}}(\mathbf{U}) - \mathcal{T}_{\mathbf{Q}}(\mathbf{U}')\right\|_F \leq \sqrt{N_q} \cdot L_{\mathrm{mix}} \cdot \left(\Gamma_Q^{(M)} \cdot L_K \cdot \Gamma_V(\mathcal{S}) + L_V\right) \cdot \|\mathbf{U} - \mathbf{U}'\|_{2,\infty},$$

which is (74) with $L_{\mathrm{post}}(\mathcal{S})$ defined in (75). The specialization (76) follows by taking $(\mathbf{U}, \mathbf{U}') = \left(\mathbf{Y}_{\mathrm{enh}}, \mathbf{Y}_{\mathrm{enh}}^{\mathrm{det}}\right)$. $\qquad \square$

## H.2. From Stage II Discrepancy to an End-to-End Bound

**Theorem H.2** ($\mathcal{K}$-PLASH: controlled approximation to standard attention)**.** *Let $\eta, \delta \in (0, 1)$, $M, N_q \in \mathbb{Z}_{>0}$, and $\tau_{\mathrm{g}} \in \mathbb{R}_{>0}$. The normalized macro features $\widetilde{\mathbf{G}}$ are those of (7), and Assumption C.2 holds. Let $\mathcal{K} \subset \mathbb{Z}_{\geq 1}$ be finite, with mixture weights $\{\beta_k\}_{k \in \mathcal{K}} \subset \mathbb{R}_{\geq 0}$ and sketch lengths $\{D_k\}_{k \in \mathcal{K}} \subset \mathbb{Z}_{>0}$.*

*By construction,*

$$\mathbf{Y}_{\mathrm{PLASH}} = \mathcal{T}_{\mathbf{Q}}\left(\mathbf{Y}_{\mathrm{enh}}\right), \qquad \mathbf{Y}_{\mathrm{PLASH}}^{\mathrm{det}} = \mathcal{T}_{\mathbf{Q}}\left(\mathbf{Y}_{\mathrm{enh}}^{\mathrm{det}}\right),$$

*with the deterministic constants of Lemma H.1 finite on the segment $\mathcal{S}$.*

*The per-degree failure budgets $\{\delta_k\}_{k \in \mathcal{K}} \subset (0, 1)$ satisfy $\sum_{k \in \mathcal{K}} \delta_k \leq \delta$. If, for every $k \in \mathcal{K}$,*

$$D_k \geq \frac{2C_k M}{\eta^2 \cdot \delta_k}, \tag{83}$$

*then with probability at least* $1 - \delta$,

$$\big\|\mathbf{Y}_{\text{PLASH}} - \mathbf{Y}_{\text{PLASH}}^{\text{det}}\big\|_F \;\leq\; \sqrt{N_q} \cdot L_{\text{post}}(\mathcal{S}) \cdot \|\mathbf{W}_{\text{out}}\|_{\text{op}} \cdot \left(\sum_{k \in \mathcal{K}} \beta_k^2\Big(\big(\sqrt{1+\eta} + D_k^{\frac{k-1}{2}}\big)^2 \cdot \tau_{\text{g}}^{-2k}\Big)\right)^{1/2}. \quad (84)$$

*Moreover, with* $\mathbf{Y}_{\text{soft}}$ *defined in* (1) *and* $\mathbf{Y}_{\text{q}}$ *defined in* (49)*, we have the deterministic decomposition*

$$\big\|\mathbf{Y}_{\text{soft}} - \mathbf{Y}_{\text{PLASH}}\big\|_F \;\leq\; \big\|\mathbf{Y}_{\text{soft}} - \mathbf{Y}_{\text{q}}\big\|_F \;+\; \big\|\mathbf{Y}_{\text{q}} - \mathbf{Y}_{\text{PLASH}}^{\text{det}}\big\|_F \;+\; \big\|\mathbf{Y}_{\text{PLASH}}^{\text{det}} - \mathbf{Y}_{\text{PLASH}}\big\|_F, \quad (85)$$

*where the first term is deterministically bounded by* (50) *and the third term is bounded with probability at least* $1 - \delta$ *by* (84).

*Proof.* **Step 1: Stage II feature discrepancy under the high-probability event** $(1 - \delta)$**.** We apply Corollary G.5 degree-wise and take a union bound over $k \in \mathcal{K}$ using budgets $\{\delta_k\}_{k \in \mathcal{K}}$. Under (83), with probability at least $1 - \delta$ we have simultaneously for all $k \in \mathcal{K}$ the Stage II feature discrepancy needed by the corollary. Therefore, on this event,

$$\big\|\mathbf{Y}_{\text{enh}} - \mathbf{Y}_{\text{enh}}^{\text{det}}\big\|_{2,\infty} \;\leq\; \|\mathbf{W}_{\text{out}}\|_{\text{op}} \cdot \left(\sum_{k \in \mathcal{K}} \beta_k^2 \cdot \big(\sqrt{1+\eta} + D_k^{\frac{k-1}{2}}\big)^2 \cdot \tau_{\text{g}}^{-2k}\right)^{1/2}. \quad (86)$$

**Step 2: propagating Stage II discrepancy through deterministic post-processing.** On the same event, we apply Lemma H.1 with $(\mathbf{U}, \mathbf{U}') = (\mathbf{Y}_{\text{enh}}, \mathbf{Y}_{\text{enh}}^{\text{det}})$:

$$\begin{aligned}
\big\|\mathbf{Y}_{\text{PLASH}} - \mathbf{Y}_{\text{PLASH}}^{\text{det}}\big\|_F &= \big\|\mathcal{T}_{\mathbf{Q}}(\mathbf{Y}_{\text{enh}}) - \mathcal{T}_{\mathbf{Q}}(\mathbf{Y}_{\text{enh}}^{\text{det}})\big\|_F \\
&\leq \sqrt{N_q} \cdot L_{\text{post}}(\mathcal{S}) \cdot \big\|\mathbf{Y}_{\text{enh}} - \mathbf{Y}_{\text{enh}}^{\text{det}}\big\|_{2,\infty}.
\end{aligned}$$

Substituting (86) yields (84).

**Step 3: deterministic decomposition relative to standard attention.** By the identity

$$\mathbf{Y}_{\text{soft}} - \mathbf{Y}_{\text{PLASH}} = (\mathbf{Y}_{\text{soft}} - \mathbf{Y}_{\text{q}}) + (\mathbf{Y}_{\text{q}} - \mathbf{Y}_{\text{PLASH}}^{\text{det}}) + (\mathbf{Y}_{\text{PLASH}}^{\text{det}} - \mathbf{Y}_{\text{PLASH}})$$

and the triangle inequality, we obtain (85). The stated bounds on the first and third terms are exactly the referenced results. $\qquad\square$

### H.3. Well-Definedness of the Post-Processing Constant

**Well-definedness of the deterministic constants in Lemma H.1 on** $\mathcal{S}$**.** Lemma H.1 requires (i) a valid local Lipschitz bound $L_{\text{mix}}$ on the segment $\mathcal{S}$ and (ii) finiteness of $\Gamma_V(\mathcal{S})$. Both hold under standard Transformer practice:

1. **Stabilized normalization.** Any LayerNorm / Norm in the mixer uses a stabilizer (e.g., $\varepsilon_{\text{ln}} > 0$), so denominators are bounded away from 0 on $\mathcal{S}$.
2. **Finite weights.** Linear maps have finite operator norms, hence $L_K$ and $L_V$ are finite.
3. **Compactness.** The set $\mathcal{S}$ is a closed line segment in $\mathbb{R}^{M \times d}$ and is therefore compact.

Under these conditions, $\text{Mixer}_{L_{\text{mix}}}$ is continuous on $\mathcal{S}$ with a finite local Lipschitz modulus (Theorem D.5 together with Theorem D.8), and $\mathbf{V}_g(\mathbf{U})$ from (78) is continuous in $\mathbf{U}$. By the extreme value theorem, $\Gamma_V(\mathcal{S}) < \infty$. Therefore $L_{\text{post}}(\mathcal{S})$ in (75) is finite and well-defined.

**Theorem H.3** (Forward-pass checkable $(\epsilon_{\text{out}}, \delta)$ guarantee for $\mathcal{K}$-PLASH). *Let* $\epsilon_{\text{out}} \in \mathbb{R}_{>0}$ *and* $\delta \in (0,1)$*. Let* $\mathbf{Y}_{\text{soft}} = \text{Atten}(\mathbf{Q}; \mathbf{K}, \mathbf{V})$ *and let* $\mathbf{Y}_{\text{PLASH}}$ *be the PLASH output on the same* $(\mathbf{Q}, \mathbf{K}, \mathbf{V})$ *with finite degree set* $\mathcal{K} \subset \mathbb{Z}_{\geq 1}$*. We write*

$$k_{\min} \triangleq \min \mathcal{K} \in \mathbb{Z}_{\geq 1}.$$

*Let* $\eta \in (0,1)$*, and let* $\{\delta_k\}_{k \in \mathcal{K}} \subset (0,1)$ *be per-degree budgets satisfying* $\sum_{k \in \mathcal{K}} \delta_k \leq \delta$*. If, for every* $k \in \mathcal{K}$*,*

$$D_k \;\geq\; \frac{2C_k M}{\eta^2 \cdot \delta_k}, \quad (87)$$

*then with probability at least $1 - \delta$, the Stage II event (69) holds.*

*The following terms are forward-pass computable.*

*(I) Deterministic Stage I term.* *We compute the hard-routing comparator $(\mathbf{K}^{\mathrm{q}}, \mathbf{V}^{\mathrm{q}})$ and radii $\rho_K(M), \rho_V(M)$ as in (48), and set*

$$\epsilon_{\mathrm{I}} \triangleq \sqrt{N_q} \cdot \left( \Gamma_Q^{(N_k)} \cdot \rho_K(M) \cdot V_{\max} + \rho_V(M) \right), \tag{88}$$

*where $\Gamma_Q^{(N_k)}$ and $V_{\max}$ are as in Lemma F.2. Thus $\|\mathbf{Y}_{\mathrm{soft}} - \mathbf{Y}_{\mathrm{q}}\|_F \le \epsilon_{\mathrm{I}}$ deterministically.*

*(det) Deterministic bias term.* *We compute $\mathbf{Y}_{\mathrm{PLASH}}^{\mathrm{det}} = \mathcal{T}_{\mathbf{Q}}\big(\mathbf{Y}_{\mathrm{enh}}^{\mathrm{det}}\big)$ and define*

$$\epsilon_{\mathrm{det}} \triangleq \big\|\mathbf{Y}_{\mathrm{q}} - \mathbf{Y}_{\mathrm{PLASH}}^{\mathrm{det}}\big\|_F. \tag{89}$$

*(II) Random Stage II term propagated through deterministic post-processing.* *We compute $L_{\mathrm{post}}(\mathcal{S})$ via (75) using any valid local Lipschitz upper bound for $L_{\mathrm{mix}}$ on the segment $\mathcal{S}$, and define*

$$\epsilon_{\mathrm{II}} \triangleq \epsilon_{\mathrm{out}} - \epsilon_{\mathrm{I}} - \epsilon_{\mathrm{det}}. \tag{90}$$

*Sufficient condition and conclusion.* *We assume $\epsilon_{\mathrm{II}} > 0$ and $\tau_{\mathrm{g}} \ge 1$ (so that $\tau_{\mathrm{g}}^{-2k} \le \tau_{\mathrm{g}}^{-2k_{\min}}$ for all $k \ge k_{\min}$, used below). If $\tau_{\mathrm{g}}$ satisfies*

$$\tau_{\mathrm{g}} \ge \epsilon_{\mathrm{II}}^{-1/k_{\min}} \cdot \left( \sqrt{N_q} \cdot L_{\mathrm{post}}(\mathcal{S}) \cdot \|\mathbf{W}_{\mathrm{out}}\|_{\mathrm{op}} \cdot \left( \sum_{k \in \mathcal{K}} |\beta_k|^2 \right)^{1/2} \cdot \max_{k \in \mathcal{K}} \left( \sqrt{1 + \eta} + D_k^{\frac{k-1}{2}} \right) \right)^{1/k_{\min}}, \tag{91}$$

*where $k_{\min} \triangleq \min\{k \mid k \in \mathcal{K}\}$, then, under (87), we have with probability at least $1 - \delta$,*

$$\big\|\mathbf{Y}_{\mathrm{soft}} - \mathbf{Y}_{\mathrm{PLASH}}\big\|_F \le \epsilon_{\mathrm{out}}. \tag{92}$$

*Proof.* By (85) and the triangle inequality,

$$\big\|\mathbf{Y}_{\mathrm{soft}} - \mathbf{Y}_{\mathrm{PLASH}}\big\|_F \le \underbrace{\big\|\mathbf{Y}_{\mathrm{soft}} - \mathbf{Y}_{\mathrm{q}}\big\|_F}_{(\mathrm{I})} + \underbrace{\big\|\mathbf{Y}_{\mathrm{q}} - \mathbf{Y}_{\mathrm{PLASH}}^{\mathrm{det}}\big\|_F}_{(\mathrm{det})} + \underbrace{\big\|\mathbf{Y}_{\mathrm{PLASH}}^{\mathrm{det}} - \mathbf{Y}_{\mathrm{PLASH}}\big\|_F}_{(\mathrm{II})}. \tag{93}$$

**Step 1: bounding term (I).** By Lemma F.2 and (88),

$$\big\|\mathbf{Y}_{\mathrm{soft}} - \mathbf{Y}_{\mathrm{q}}\big\|_F \le \epsilon_{\mathrm{I}} \qquad \text{deterministically.}$$

**Step 2: term (det) is exactly $\epsilon_{\mathrm{det}}$.** By definition (89),

$$\big\|\mathbf{Y}_{\mathrm{q}} - \mathbf{Y}_{\mathrm{PLASH}}^{\mathrm{det}}\big\|_F = \epsilon_{\mathrm{det}} \qquad \text{deterministically.}$$

**Step 3: applying the Stage II discrepancy bound and the post-processing Lipschitz property to term (II).** We first apply Lemma H.1 with $(\mathbf{U}, \mathbf{U}') = (\mathbf{Y}_{\mathrm{enh}}, \mathbf{Y}_{\mathrm{enh}}^{\mathrm{det}})$:

$$\big\|\mathbf{Y}_{\mathrm{PLASH}}^{\mathrm{det}} - \mathbf{Y}_{\mathrm{PLASH}}\big\|_F \le \sqrt{N_q} \cdot L_{\mathrm{post}}(\mathcal{S}) \cdot \big\|\mathbf{Y}_{\mathrm{enh}}^{\mathrm{det}} - \mathbf{Y}_{\mathrm{enh}}\big\|_{2,\infty}. \tag{94}$$

Next, on the probability-$(1 - \delta)$ event implied by (87), the Stage II bound (69) holds. Therefore, on that same event,

$$\big\|\mathbf{Y}_{\mathrm{enh}}^{\mathrm{det}} - \mathbf{Y}_{\mathrm{enh}}\big\|_{2,\infty} \le \|\mathbf{W}_{\mathrm{out}}\|_{\mathrm{op}} \cdot \left( \sum_{k \in \mathcal{K}} |\beta_k|^2 \cdot \big(\sqrt{1 + \eta} + D_k^{\frac{k-1}{2}}\big)^2 \cdot \tau_{\mathrm{g}}^{-2k} \right)^{1/2}$$

$$\le \|\mathbf{W}_{\mathrm{out}}\|_{\mathrm{op}} \cdot \left( \sum_{k \in \mathcal{K}} |\beta_k|^2 \right)^{1/2} \cdot \max_{k \in \mathcal{K}} \big(\sqrt{1 + \eta} + D_k^{\frac{k-1}{2}}\big) \cdot \tau_{\mathrm{g}}^{-k_{\min}}. \tag{95}$$

Substituting (95) into (94), on the same probability-$(1 - \delta)$ event,

$$\left\|\mathbf{Y}_{\text{PLASH}}^{\text{det}} - \mathbf{Y}_{\text{PLASH}}\right\|_F \leq \sqrt{N_q} \cdot L_{\text{post}}(\mathcal{S}) \cdot \|\mathbf{W}_{\text{out}}\|_{\text{op}} \cdot \left(\sum_{k \in \mathcal{K}} |\beta_k|^2\right)^{1/2} \cdot \max_{k \in \mathcal{K}}\left(\sqrt{1+\eta} + D_k^{\frac{k-1}{2}}\right) \cdot \tau_{\text{g}}^{-k_{\min}}. \quad (96)$$

**Step 4: choosing $\tau_{\text{g}}$.** Rearranging (96), condition (91) gives $\|\mathbf{Y}_{\text{PLASH}}^{\text{det}} - \mathbf{Y}_{\text{PLASH}}\|_F \leq \epsilon_{\text{II}}$ on the probability-$(1 - \delta)$ event. Summing the three bounds via (93),

$$\left\|\mathbf{Y}_{\text{soft}} - \mathbf{Y}_{\text{PLASH}}\right\|_F \leq \epsilon_{\text{I}} + \epsilon_{\text{det}} + \epsilon_{\text{II}} = \epsilon_{\text{out}},$$

which proves (92). $\qquad\qquad\qquad\qquad\qquad\qquad\qquad\qquad\qquad\qquad\qquad\qquad\qquad\qquad\qquad\qquad\qquad\qquad\quad\square$

**Checkability and computational overhead for general $\mathcal{K}$.** All terms in Theorem H.3 are *forward-pass computable* and do not require forming the $N_q \times N_k$ logit matrix. The sketch sizing rule (e.g., (87) with per-degree parameters) is a one-time design choice (fixed before training / inference).

At inference time:

1. **Stage I term $\epsilon_{\text{I}}$.** The bound uses only Stage I tensors (routing weights and compressed keys / values) and is computed from quantities already produced in Stage I (cf. the Stage I bound, e.g., (48)).

2. **Deterministic bias $\epsilon_{\text{det}}$.** By definition (89), $\epsilon_{\text{det}}$ compares the PLASH output to a *deterministic reference* that replaces the randomized Stage II sketches by their deterministic counterparts at each degree $k \in \mathcal{K}$. We do *not* compute $\mathbf{Y}_{\text{PLASH}}^{\text{det}}$ explicitly. Instead, (89) expresses $\epsilon_{\text{det}}$ through intermediate deterministic quantities on the compressed path (Stage I outputs and deterministic Stage II features / embeddings). Thus evaluating $\epsilon_{\text{det}}$ requires only running the same length-$M$ Stage II pipeline once with the deterministic per-degree maps $\{\mathbf{z}_{j,k}^{\text{det}}\}_{j \in [M], k \in \mathcal{K}}$ (and then applying the same readout / mixer / projections, as required by (89)). This introduces no $N_q \times N_k$ computation.

3. **Stage II stochastic term $\epsilon_{\text{II}}$ and the sufficient condition.** For general $\mathcal{K}$, the stochastic term is controlled by the discrepancy $\|\mathbf{Y}_{\text{enh}} - \mathbf{Y}_{\text{enh}}^{\text{det}}\|_{2,\infty}$ where both tensors lie in $\mathbb{R}^{M \times d}$. The sufficient condition (91) and guarantee (92) use only such length-$M$ quantities and deterministic post-processing factors (cf. Appendix H), and therefore never introduce an $N_q \times N_k$ computation.

Overall, certification adds only *compressed-side* overhead: it scales with the bottleneck sizes $(M, \{D_k\}_{k \in \mathcal{K}})$ (e.g., one additional deterministic Stage II pass on length $M$) and preserves PLASH's linear-in-length structure in $(N_q, N_k)$.

## I. Complexity Analysis of the PLASH Module

The proof of Theorem 3.3 accounts stage-wise for the dominant linear-algebra and FFT operations in Algorithm 1. Each bound is stated in a form that makes the dependence on $(N_q, N_k, M)$ explicit.

**Theorem I.1** (Baseline complexity of Stages I–III (no accuracy constraints))**.** *We consider Algorithm 1 with $\mathbf{Q} \in \mathbb{R}^{N_q \times d_k}$, $\mathbf{K} \in \mathbb{R}^{N_k \times d_k}$, $\mathbf{V} \in \mathbb{R}^{N_k \times d_v}$, and macro length $M \ll N_k$. Let $\mathcal{K} \subset \mathbb{Z}_{\geq 1}$ be the degree set and let $\{D_k\}_{k \in \mathcal{K}}$ be the TensorSketch dimensions, with $D_{\text{tot}} \triangleq \sum_{k \in \mathcal{K}} D_k$. Stage II computes TensorSketch features via CountSketch followed by FFT-based circular convolutions, and the mixer is a Transformer encoder of depth $L_{\text{mix}}$.*

*Then the runtime decomposes as*

$$T_{\text{PLASH}} = T_{\text{I}} + T_{\text{II}} + T_{\text{III}},$$

*with stage-wise bounds*

$$T_{\text{I}} = O\left(N_k \cdot M \cdot d_k\right) + O\left(N_k \cdot M\right) + O\left(N_k \cdot M \cdot d_k\right) + O\left(N_k \cdot M \cdot d_v\right),$$

$$T_{\text{II}} = O\left(M \cdot \text{cost}(\psi)\right) + O\left(M \cdot d \cdot D_{\text{tot}}\right) + O\left(M \cdot \sum_{k \in \mathcal{K}} k \cdot \left(d' + D_k \log D_k\right)\right) + O\left(L_{\text{mix}} \cdot \left(M^2 \cdot d + M \cdot d^2\right)\right),$$

$$T_{\text{III}} = O\left(N_q \cdot M \cdot d_k\right) + O\left(N_q \cdot M\right) + O\left(N_q \cdot M \cdot d_v\right).$$

*In particular, if $(d, d', d_k, d_v, \mathcal{K}, \{D_k\}_{k \in \mathcal{K}}, L_{\mathrm{mix}})$ are treated as fixed independently of $(N_q, N_k)$, then*

$$T_{\mathrm{PLASH}} \;=\; O\Big((N_q + N_k) \cdot M\Big).$$

*If the routing weights $\mathbf{A} \in \mathbb{R}^{N_k \times M}$ and the readout attention weights $\mathrm{softmax}\Big(\mathbf{Q} \cdot \mathbf{K}_g^\top / \sqrt{d_k}\Big) \in \mathbb{R}^{N_q \times M}$ are materialized, then the peak memory for these two matrices is $O(N_k \cdot M + N_q \cdot M)$. Both Stage I and Stage III admit streaming implementations that avoid storing these matrices.*

*Proof.* We use the standard cost rules: a dense product $(a \times b)(b \times c)$ costs $O(abc)$; a row-wise softmax on an $a \times c$ matrix costs $O(ac)$; an FFT or IFFT of length $D$ costs $O(D \log D)$ (Cooley & Tukey, 1965). Constants and lower-order terms are suppressed.

**Stage I.** The logits $\mathbf{S} = \mathbf{K} \cdot \mathbf{P}^\top$ cost $O(N_k M d_k)$; the row-wise softmax $\mathbf{A}$ costs $O(N_k M)$; the aggregations $\widetilde{\mathbf{K}} = \mathbf{A}^\top \cdot \mathbf{K}$ and $\widetilde{\mathbf{V}} = \mathbf{A}^\top \cdot \mathbf{V}$ cost $O(N_k M d_k)$ and $O(N_k M d_v)$. Summing gives $T_{\mathrm{I}}$.

**Stage II.** The row-wise pre-map costs $O(M \cdot \mathrm{cost}(\psi))$. For each row and degree $k \in \mathcal{K}$, the $k$ CountSketch passes cost $O(kd')$ and the $k$-fold FFT convolution $\mathrm{IFFT}\big(\bigodot_t \mathrm{FFT}(\mathbf{c}_{k,t})\big)$ costs $O(k D_k \log D_k)$; summing over $j \in [M]$ and $k \in \mathcal{K}$ gives $O\big(M \sum_{k \in \mathcal{K}} k(d' + D_k \log D_k)\big)$. The linear readout $\mathbf{W}_{\mathrm{out}} \cdot \mathbf{v}_j$ costs $O(M d D_{\mathrm{tot}})$. One encoder layer on $\mathbb{R}^{M \times d}$ costs $O(M^2 d + M d^2)$ (MHSA projections $O(M d^2)$, per-head logits and value aggregation $O(M^2 d)$; FFN $O(M d \cdot d_{\mathrm{ff}}) = O(M d^2)$ under $d_{\mathrm{ff}} = \Theta(d)$ (Vaswani et al., 2017); LayerNorm and residuals $O(M d)$ are lower-order), so the depth-$L_{\mathrm{mix}}$ mixer costs $O\big(L_{\mathrm{mix}}(M^2 d + M d^2)\big)$. Summing gives $T_{\mathrm{II}}$.

**Stage III.** The logits $\mathbf{Q} \cdot \mathbf{K}_g^\top$ cost $O(N_q M d_k)$, the row-wise softmax $O(N_q M)$, and the weighted sum by $\mathbf{V}_g$ costs $O(N_q M d_v)$, giving $T_{\mathrm{III}}$.

With $(d, d', d_k, d_v, \mathcal{K}, \{D_k\}_{k \in \mathcal{K}}, L_{\mathrm{mix}})$ fixed independently of $(N_q, N_k)$, every Stage II term depends only on $M$, so $T_{\mathrm{PLASH}} = O\big((N_q + N_k)M\big)$.

**Memory.** Materializing $\mathbf{A} \in \mathbb{R}^{N_k \times M}$ and the readout weights in $\mathbb{R}^{N_q \times M}$ costs $O\big((N_k + N_q)M\big)$, but both are avoided by streaming: accumulate $\widetilde{\mathbf{K}}, \widetilde{\mathbf{V}}$ row-by-row in Stage I, and compute Stage III attention one query row at a time. The remaining working storage ($O(M d)$, $O(M D_{\mathrm{tot}})$, and FFT arrays $O(\max_{k \in \mathcal{K}} D_k)$) does not scale with $N_q N_k$. $\qquad\square$

## J. Block-Causal PLASH for Autoregressive Language Modeling

The main PLASH block uses a global prototype routing matrix $\mathbf{A} \in \mathbb{R}^{M \times N_k}$, which couples every key / value position to every prototype. For autoregressive language modeling, we replace global routing with a *block-causal* variant that respects the lower-triangular masking required by next-token prediction while preserving the linear-in-$N_k$ complexity of PLASH.

### J.1. Construction

Let $N, B \in \mathbb{Z}_{>0}$ with $B \le N$ denote the sequence length and block size; throughout this appendix we work in the autoregressive language-modeling setting where queries and keys share the same sequence length, i.e., $N_q = N_k = N$ (the general $N_q \ne N_k$ case follows by replacing $N$ with $N_q$ or $N_k$ as appropriate). We partition the sequence into contiguous blocks of size $B$, indexed by $b \in \{0, 1, \dots, \lceil N/B \rceil - 1\}$. For block $b$, the Stage I prototype computation uses only the union of preceding blocks $0, \dots, b-1$ and the prefix of the current block up to the query position. Concretely, let $\mathbf{A}^{(b)} \in \mathbb{R}^{M \times bB}$ denote the prefix routing matrix obtained from a causal prefix sum over $\mathbf{A}_{:,1:bB}$; the block-$b$ prototype keys and values are

$$\widetilde{\mathbf{K}}^{(b)} \;=\; \mathbf{A}^{(b)} \cdot \mathbf{K}_{1:bB,:}, \qquad\qquad \widetilde{\mathbf{V}}^{(b)} \;=\; \mathbf{A}^{(b)} \cdot \mathbf{V}_{1:bB,:}.$$

Stage II enriches $\widetilde{\mathbf{K}}^{(b)}, \widetilde{\mathbf{V}}^{(b)}$ via TensorSketch as in the non-causal block. In Stage III, queries inside block $b$ compute scaled dot-product softmax attention against the concatenation of (i) the enriched prototypes from the strictly preceding blocks and (ii) the standard lower-triangular masked attention over the current block.

The Stage I causal prefix sum costs $O(NBd)$ and the prototype attention costs $O(NMd)$, giving end-to-end $O(NBd + NMd)$ versus $O(N^2 d)$ for standard softmax attention, with $B$ a constant independent of $N$.

*Table 6.* Best validation loss on TinyStories with Block-Causal PLASH. Lower is better.

| Scale | Standard | ABC | **PLASH** (ours) |
|---|---|---|---|
| 124M (Small) | 1.194 | 1.647 | **1.500** |
| 774M (Large) | 0.968 | 1.234 | **1.267** |
| 1.5B (XL) | 0.919 | 1.257 | **1.044** |

*Table 7.* Attention-layer forward-pass time (ms) on A800-80GB at $d=512$, batch 16.

| $N$ | Standard (ms) | **PLASH** (ms) | Speed-up |
|---|---|---|---|
| 512 | 4.6 | **3.6** | 1.3× |
| 1,024 | 13.5 | **6.5** | 2.1× |
| 2,048 | 43.4 | **11.4** | 3.8× |
| 4,096 | 159.1 | **22.0** | 7.2× |
| 8,192 | 615.2 | **43.0** | **14.3×** |

## J.2. Causal Correctness Verification

For a trained 124M model and 40 independent test inputs, we (i) recorded the forward pass on the full sequence, (ii) replaced all tokens beyond a random position $t$ with random noise, and (iii) re-ran the forward pass and compared outputs at positions $\leq t$. Across all 40 trials, the maximum absolute output difference at protected positions was $\leq 3.24 \times 10^{-5}$, matching bf16 non-determinism.

## J.3. Language Modeling Quality (GPT-2 Scales on TinyStories)

We train GPT-2 (Radford et al., 2019) (Small / Large / XL: 124M, 774M, 1.5B parameters) on TinyStories (Eldan & Li, 2023) with $M=16$ and $B=64$, sharing identical hyper-parameters across the three architectures. Standard attention is accelerated by FlashAttention-2 (Dao, 2024); we compare against ABC (Peng et al., 2022) as a representative learned-latent baseline.

The approximation gap to Standard narrows with model capacity, from $+0.31$ at 124M to $+0.13$ at 1.5B.

## J.4. Latency Scaling (2-Layer GPT, $d=512$, Batch 1, bf16)

Standard scales quadratically ($4.6 \to 615$ ms, $134\times$ for $16\times$ longer $N$); PLASH scales linearly ($3.6 \to 43$ ms, $12\times$), faster than Standard for $N \geq 512$.

## J.5. Discussion

**Hyper-parameter portability.** $M=16$ suffices for every GPT-2 scale (124M–1.5B on TinyStories); on Qwen3-4B (PG-19), $M=16$ is optimal at $N=16$K and $M=32$ at $N=32$K. Across architectures, $M$ is set from sequence length and head dimension and not tuned per task. The deviation certificate of Theorem 3.2 depends only on per-head dimension $d_k$ and prototype count $M$, hence applies identically across architectures.

# K. Empirical Prevalence of the Certificate on Synthetic Inputs

The certificate of Theorem H.3 is forward-pass checkable and induces an empirical certification rate, analogous to certified accuracy in randomized smoothing (Cohen et al., 2019). Concretely, given a collection of $T$ evaluation inputs $\{(\mathbf{Q}^{(t)}, \mathbf{K}^{(t)}, \mathbf{V}^{(t)})\}_{t=1}^{T}$, we define

$$\Delta^{(t)} \triangleq \epsilon_{\text{out}} - \epsilon_{\text{I}}^{(t)} - \epsilon_{\text{det}}^{(t)},$$

$$C^{(t)} \triangleq \sqrt{N_q} \cdot L_{\text{post}}^{(t)}(\mathcal{S}) \cdot \|\mathbf{W}_{\text{out}}\|_{\text{op}} \cdot \left( \sum_{k \in \mathcal{K}} |\beta_k|^2 \right)^{1/2} \cdot \max_{k \in \mathcal{K}} \left( \sqrt{1+\eta} + D_k^{\frac{k-1}{2}} \right),$$

where $\epsilon_{\mathrm{I}}^{(t)}$ and $\epsilon_{\mathrm{det}}^{(t)}$ are computed by (88) and (89), and $L_{\mathrm{post}}^{(t)}(\mathcal{S})$ is computed via (75). Then the empirical certification rate is

$$\mathrm{CertRate}(\epsilon_{\mathrm{out}}) \triangleq \frac{1}{T} \sum_{t=1}^{T} \mathbf{1}\left\{ \{\Delta^{(t)} > 0\} \wedge \left\{ \tau_{\mathrm{g}} \geq \left( \frac{C^{(t)}}{\Delta^{(t)}} \right)^{1/k_{\min}} \right\} \right\},$$

with $k_{\min} \triangleq \min \mathcal{K}$ (and $k_{\min} = 1$ for the $\mathcal{K} = \{1\}$ specialization). **The certified set contains open neighborhoods (non-vacuity).** Whenever the certificate holds with strict slack at one input, it also holds on a neighborhood of nearby inputs.

*Step 1: continuity of the forward-pass quantities.* The map $(\mathbf{Q}, \mathbf{K}, \mathbf{V}) \mapsto \left( \epsilon_{\mathrm{I}}, \epsilon_{\mathrm{det}}, L_{\mathrm{post}}(\mathcal{S}) \right)$ is continuous as a composition of continuous primitives (matrix products, softmax, stabilized normalization, linear maps), wherever the constants of Lemma H.1 are finite.

*Step 2: strict inequalities persist under small perturbations.* We suppose the certificate holds at an input $(\mathbf{Q}, \mathbf{K}, \mathbf{V})$ with strict slack:

$$\epsilon_{\mathrm{out}} - \epsilon_{\mathrm{I}} - \epsilon_{\mathrm{det}} > 0 \quad \text{and} \quad \tau_{\mathrm{g}} > \left( \frac{C}{\epsilon_{\mathrm{out}} - \epsilon_{\mathrm{I}} - \epsilon_{\mathrm{det}}} \right)^{1/k_{\min}},$$

where $C$ denotes the corresponding coefficient built from $L_{\mathrm{post}}(\mathcal{S})$ and other forward-pass quantities (as in the definition of $C^{(t)}$). By continuity, there exists a neighborhood of $(\mathbf{Q}, \mathbf{K}, \mathbf{V})$ in which both strict inequalities remain true. Hence the certified set has nonempty interior, justifying the empirical rate $\mathrm{CertRate}(\epsilon_{\mathrm{out}})$ as a measure of how frequently the model operates in this stable regime. Sweeping $(M, \{D_k\}_{k \in \mathcal{K}}, \tau_{\mathrm{g}})$ over standard evaluation sets yields $\mathrm{CertRate}(\epsilon_{\mathrm{out}})$ from forward-pass quantities alone.

**Certificate for $\mathcal{K} = \{1\}$.** For $\mathcal{K} = \{1\}$, Stage II reduces to a linear randomized feature map; the only approximation sources are Stage I compression ($\epsilon_{\mathrm{I}}$) and Stage II randomization (controlled via (87)).

**Simulation protocol.** Figure 4 reports $\mathrm{CertRate}(\epsilon_{\mathrm{out}})$ over a grid of $(\tau_{\mathrm{g}}, \epsilon_{\mathrm{out}})$, with every pixel an explicit Monte-Carlo estimate.

**Common-range, multi-$M$ visualization.** Four panels vary $M \in \{16, 32, 48, 64\}$ with all other architectural and sampling hyperparameters fixed, on a $15 \times 15$ log-spaced grid with $\tau_{\mathrm{g}} \in [6.31 \times 10^2, \, 1.585 \times 10^3]$ and $\epsilon_{\mathrm{out}} \in [5.20 \times 10^2, \, 8.53 \times 10^2]$. The shared axis range attributes shifts of the transition band across panels to $M$ alone.

**Sampling setup.** At each grid point $(\tau_{\mathrm{g}}, \epsilon_{\mathrm{out}})$ and each $M$, the experiment draws $T = 200$ i.i.d. attention inputs $(\mathbf{Q}, \mathbf{K}, \mathbf{V})$ with $\mathbf{Q} \in \mathbb{R}^{N_q \times h \times d}$ and $\mathbf{K}, \mathbf{V} \in \mathbb{R}^{N_k \times h \times d}$ ($N_q = 32$, $N_k = 64$, $h = 4$, $d = 32$); heads are sampled independently. A per-trial scale $\sigma \sim \mathrm{Unif}[0.004, 0.08]$ varies the input magnitude, and entries of $\mathbf{Q}, \mathbf{K}, \mathbf{V}$ are drawn i.i.d. from $\mathcal{N}(0, \sigma^2)$.

**PLASH instantiation ($\mathcal{K} = \{1\}$).** For each draw we run the $\mathcal{K} = \{1\}$ PLASH block in the standard $(\mathbf{Q}, \mathbf{K}, \mathbf{V})$ interface with sketch dimension $D = 256$: Stage I computes routing scores $\mathbf{S} = \mathbf{K} \cdot \mathbf{P}^\top$ to $M$ prototypes $\mathbf{P} \in \mathbb{R}^{M \times d}$ and forms compressed summaries $\widetilde{\mathbf{K}} = \mathbf{A}^\top \cdot \mathbf{K}$ and $\widetilde{\mathbf{V}} = \mathbf{A}^\top \cdot \mathbf{V}$ with $\mathbf{A} = \mathrm{softmax}(\mathbf{S}/\tau)$ (row-wise; $\tau = 1$). Stage II applies the pre-map $\psi$ to the concatenated compressed features and enforces stabilized norm control

$$\widetilde{\mathbf{G}}_{j,:} = \frac{\mathbf{G}_{j,:}}{\max\{\|\mathbf{G}_{j,:}\|_2, \varepsilon_g\} \cdot \tau_{\mathrm{g}}},$$

followed by the only randomized module, CountSketch (TensorSketch with $\mathcal{K} = \{1\}$) of dimension $D$ and a linear readout. The mixer applies a deterministic mixer over the length-$M$ compressed sequence. Stage III then performs *exact* scaled dot-product attention from $\mathbf{Q}$ to the resulting compressed keys / values, producing $\mathbf{Y}_{\mathrm{PLASH}}$. As a baseline we compute the standard attention output $\mathbf{Y}_{\mathrm{soft}} = \mathrm{Atten}(\mathbf{Q}; \mathbf{K}, \mathbf{V})$.

**Certification criterion.** On each trial we compute the forward-pass quantities in Theorem H.3: the Stage I term $\epsilon_{\mathrm{I}}$ (via the hard-routing reconstruction and radii $\rho_K(M) = \|\mathbf{K} - \mathbf{K}^{\mathrm{q}}\|_{2,\infty}$, $\rho_V(M) = \|\mathbf{V} - \mathbf{V}^{\mathrm{q}}\|_{2,\infty}$), the deterministic bias $\epsilon_{\mathrm{det}}$ (using the deterministic $k{=}1$ comparator obtained by padding / truncation to length $D$), and a checkable upper bound on $L_{\mathrm{post}}(\mathcal{S})$ as in (75). Given $\epsilon_{\mathrm{out}}$, the certificate holds if the slack $\Delta = \epsilon_{\mathrm{out}} - \epsilon_{\mathrm{I}} - \epsilon_{\mathrm{det}}$ is positive and the sufficient condition (91) is satisfied. We then report $\mathrm{CertRate}(\epsilon_{\mathrm{out}}) = T^{-1} \cdot \sum_{t=1}^{T} \mathbf{1}\{\text{certificate holds on trial } t\}$.

Within the shared window the heatmaps contain substantial mass with $0 < \mathrm{CertRate}(\epsilon_{\mathrm{out}}) < 1$ (quantization step $1/T = 0.005$): the condition is selective yet frequently satisfied.

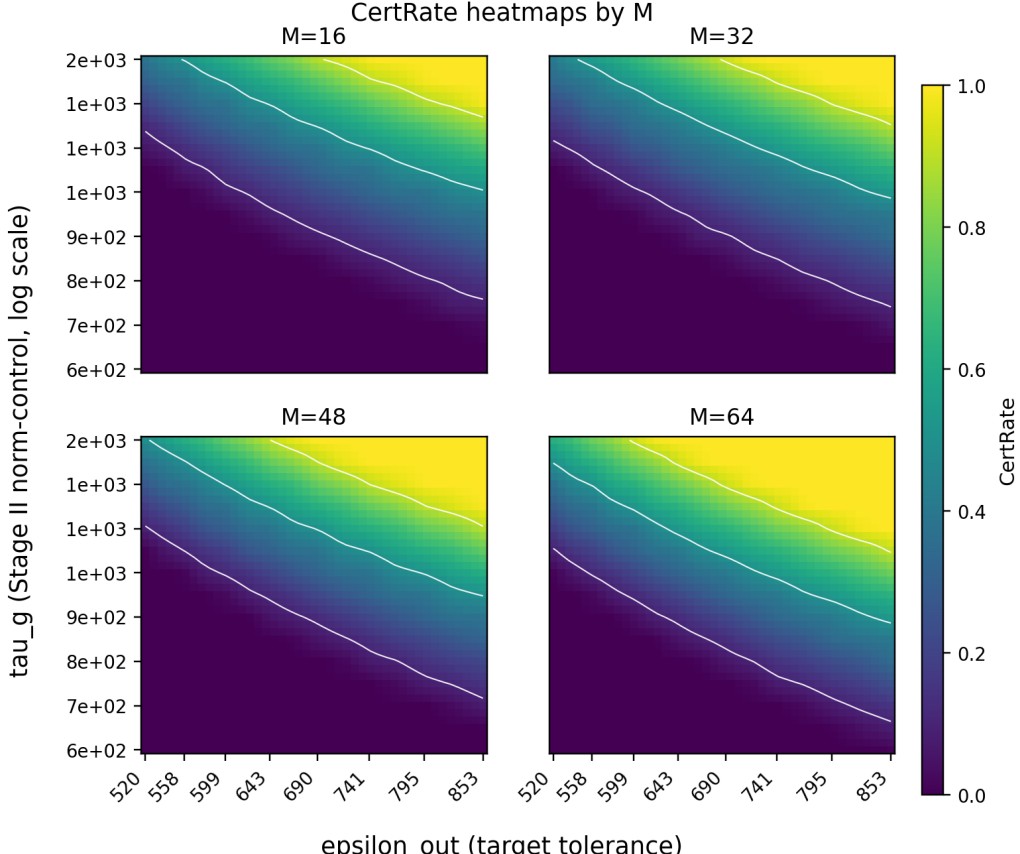

*Figure 4.* **Empirical certification rate for $\mathcal{K} = \{1\}$ PLASH, shown at a common $(\tau_\mathrm{g}, \epsilon_\mathrm{out})$ range for different $M$.** Each panel corresponds to a different number of groups $M \in \{16, 32, 48, 64\}$, while keeping $(N_q, N_k, h, d, D) = (32, 64, 4, 32, 256)$ fixed. Axes are shared across panels: a $15 \times 15$ log-spaced grid with $\tau_\mathrm{g} \in [6.31 \times 10^2, 1.585 \times 10^3]$ and $\epsilon_\mathrm{out} \in [5.20 \times 10^2, 8.53 \times 10^2]$. Each cell reports $\mathrm{CertRate}(\epsilon_\mathrm{out})$ computed from $T = 200$ i.i.d. draws of $(\mathbf{Q}, \mathbf{K}, \mathbf{V})$ with per-trial scale $\sigma \sim \mathrm{Unif}[0.004, 0.08]$. Color encodes the fraction of trials for which the forward-pass condition in Theorem H.3 certifies $\|\mathbf{Y}_\mathrm{soft} - \mathbf{Y}_\mathrm{PLASH}\|_F \leq \epsilon_\mathrm{out}$.

**Effect of $M$: certification becomes easier.** Holding $(\tau_\mathrm{g}, \epsilon_\mathrm{out})$ fixed, the certified region expands with $M$. Quantitatively, over the common $(\tau_\mathrm{g}, \epsilon_\mathrm{out})$ grid the mean certification rate increases from 0.246 ($M = 16$) to 0.368 ($M = 64$), and the fraction of grid points with $\mathrm{CertRate} \geq 0.9$ increases from 0.053 to 0.160. The $\mathrm{CertRate} = 0.9$ transition also shifts toward smaller tolerances / temperatures: the median $\epsilon_\mathrm{out}$ needed to reach $\mathrm{CertRate} \geq 0.9$ decreases from $\approx 767$ at $M = 16$ to $\approx 715$ at $M = 64$, and the median $\tau_\mathrm{g}$ needed decreases from $\approx 1.48 \times 10^3$ to $\approx 1.39 \times 10^3$ (within this window). These shifts match the theorem structure: increasing $M$ reduces the Stage I approximation radii (and thus $\epsilon_\mathrm{I}$), increasing the slack available for certification.

## L. Certificate Validation on Real Test Inputs

The certification heatmaps in the main text (Figure 2, generated on synthetic random inputs at $\mathcal{K} = \{1\}$) establish that the bound of Theorem 3.2 is non-vacuous on representative inputs. This section evaluates the certificate on *real* forecasting inputs at the trained $(M, D_k, \mathcal{K})$ configurations of the main experiments.

For each trained model (one per dataset), the certificate is evaluated on the full real test split with degree set $\mathcal{K} = \{1, 2\}$. Let $\theta(\mathbf{x})$ denote the certificate's forward-pass-computable deviation upper bound for input $\mathbf{x}$, and let $\|\Delta(\mathbf{x})\|$ denote the realized layer-output deviation. We report (i) coverage $= \mathrm{Pr}_\mathbf{x}(\|\Delta(\mathbf{x})\| \leq \theta(\mathbf{x}))$ and (ii) the rank correlation between $\theta$ and $\|\Delta\|$.

Across 35,058 real test samples, the certificate holds with zero violations. The Spearman correlation between the certificate value $\theta$ and the realized deviation is positive on ETTm1 and ECL ($p < 0.05$), so $\theta$ ranks inputs by approximation difficulty and identifies, at inference time without ground-truth labels, the inputs with the largest realized deviation. The Weather correlation is statistically null and ETTh1 is negative: on those datasets, the realized deviation lies far below $\theta$, so $\theta$ does not

*Table 8.* Certificate validation on real test inputs. Coverage is the fraction of samples for which the predicted deviation bound $\theta(\mathbf{x})$ holds. Spearman $\rho$ measures how well $\theta$ ranks inputs by realized deviation; $p$ is the two-sided null-hypothesis significance.

| Dataset | Samples | Coverage | Spearman $\rho$ | $p$-value |
|---------|---------|----------|-----------------|-----------|
| ETTm1 | 11,425 | **1.000** | 0.221 | $< 10^{-126}$ |
| ECL | 6,918 | **1.000** | 0.316 | $< 10^{-24}$ |
| Weather | 13,930 | **1.000** | 0.003 | 0.73 |
| ETTh1 | 2,785 | **1.000** | $-0.484$ | $< 10^{-10}$ |
| **Total** | **35,058** | **1.000** | — | — |

*Table 9.* Component ablations on ETTm1 ($d$=128, 4 heads, 2 layers, $M$=64). The TensorSketch $\to$ Multi-Layer Perceptron (MLP) swap loses the certificate because an MLP lacks the analytic polynomial structure that Theorem 3.1 bounds.

| Variant | MSE | Effect |
|---------|-----|--------|
| **PLASH (full)** | **0.760** | Reference |
| Remove mixer (Stage IIb) | 0.792 | +4.2% MSE |
| Prototypes $\to$ average pooling | 0.791 | +4.1% MSE |
| TensorSketch $\to$ MLP | 0.893 | +17.5% MSE, *loses certificate* |

*Table 10.* Long-context forecasting MSE at $L$=336. PLASH outperforms standard attention on all four datasets.

| Dataset | Standard ($L$=336) | **PLASH** ($L$=336) |
|---------|--------------------|--------------------|
| WTH | 0.323 | **0.305** |
| ETTh1 | 0.632 | **0.621** |
| ETTh2 | 0.853 | **0.688** |
| ETTm1 | 0.279 | **0.272** |

rank within-bulk variation.

# M. Component Ablations

The following ablations isolate the contribution of each architectural component on ETTm1 with $d$=128, 4 heads, 2 layers, and $M$=64.

Replacing TensorSketch with an equal-capacity MLP increases MSE by $17.5\%$ and *eliminates the deviation certificate*: the MLP carries no analytic polynomial-approximation error bound, whereas the TensorSketch error is governed by Theorem 3.1.

# N. Sensitivity to Routing Temperature $\tau$ and Sketch Temperature $\tau_g$

PLASH has two temperature parameters: $\tau$ controlling Stage I routing granularity and $\tau_g$ controlling Stage II sketch precision.

**Routing temperature $\tau$ (ETTm1).** Varying $\tau \in \{0.2, 0.3, 0.5\}$ yields RMSE 1799/1806/1807, a relative spread below $0.5\%$.

**Sketch temperature $\tau_g$ (ETTm1).** $\tau_g$=1.0 minimizes RMSE (3.681); $\tau_g$=2.0 maximizes the rank correlation between $\theta$ and realized deviation (Spearman $\rho$=0.290). The two parameters control different quantities: $\tau$ controls Stage I routing accuracy, and $\tau_g$ controls the accuracy / certificate-tightness trade-off in Stage II. No statistical interaction between $\tau$ and $\tau_g$ is observed.

# O. Long-Context Forecasting at $L$=336

The four forecasting datasets are re-run at $L$=336 (versus $L$=96 in the main table) with the per-dataset optimal $(M, D_k)$ configurations; MSE is the best of three runs.

The largest MSE reduction ($-0.165$) is on ETTh2; the linear-complexity scaling of PLASH thus admits longer history at a fixed compute budget.

