# OpenReview forum: "PLASH: Provably Linear-Time Attention with Selective Higher-Order Feature Sketching"
_ICML.cc/2026/Conference — ICML 2026 regular_

### Official Review · Reviewer_AgGr · 2026-03-07

**Soundness:** 3
**Presentation:** 3
**Significance:** 2
**Originality:** 2
**Overall Recommendation:** 4
**Confidence:** 3

**Summary:**

The paper introduces PLASH, an attention architecture that achieves linear-time complexity in the sequence length by reducing the dominant cost from standard quadratic attention to linear. Instead of modifying the queries, the method compresses the Key and Value (KV) matrices into a fixed length using a deterministic, learnable prototype-based routing mechanism. To compensate for the expressivity lost during linear compression, the architecture enriches the compressed KV summaries using randomized higher-order feature sketching (TensorSketch) and applies a short-sequence Transformer mixer. PLASH performs a scaled dot-product softmax readout from the uncompressed Queries to the enriched, compressed KVs. The authors also provide a forward-pass computable certificate bounding the approximation error relative to standard softmax attention.

**Compliance With Llm Reviewing Policy:**

Affirmed.

**Final Justification:**

I thank the authors for their response. The additional clarifications and experimental results have sufficiently addressed my primary concerns. In light of these improvements, I have raised my score to Weak Accept.

**Key Questions For Authors:**

1. Is PLASH fundamentally incompatible with causal masking and autoregressive generation? If it can be adapted for decoder-only models, how do you prevent future tokens from leaking into the compressed KV representations during the global prototype routing in Stage I?


2. How practically relevant is the theoretical error certificate to end-to-end task performance? Since the network's weights naturally adapt to the sketched features during training, why is bounding the strict distance to standard attention necessary? Additionally, with the target tolerances ($inline$\epsilon_{out}$inline$) in the hundreds (Figure 2) compared to task errors well below 1.0 (Table 1), aren't these bounds too loose to guarantee meaningful output stability?


3. Could you provide ablation studies to justify the architectural complexity of the KV reduction and enrichment process? Specifically, what is the empirical impact of removing the short-sequence Transformer mixer, or replacing the prototype-based routing and TensorSketch with simpler baselines like average pooling?

**Limitations:**

yes

**Strengths And Weaknesses:**

`Incompatibility with Causal Masking and Autoregressive Generation`: While the method successfully reduces complexity, it appears fundamentally incompatible with causal masking. The prototype-based routing in Stage I aggregates information globally across the entire key and value sequences to form the $M$ representatives. Because future tokens are inevitably mixed into these compressed representations, this architecture cannot natively support the strict lower-triangular masking required for autoregressive decoder-only models (like modern LLMs). This limitation restricts the method primarily to encoder-focused or bidirectional tasks.

`Requirement for Costly Pretraining from Scratch`: Because PLASH introduces entirely new, randomly initialized parameter matrices—such as the $M$ learnable prototype vectors ($P$) in Stage I, and the degree weights ($\beta_k$) and readout matrix ($W_{out}$) in Stage II—it cannot serve as a drop-in, zero-shot replacement for existing pre-trained models. Adopting this architecture requires pretraining from scratch (as done with the Informer backbone in the experiments). This represents a massive computational barrier to adoption compared to tuning-free efficient attention methods.

`Questionable Practical Relevance of the Theoretical Proof`: The theoretical error certificate is a major focal point, but its practical relevance to end-to-end performance is debatable. Because the network's weights naturally adapt to the sketched features during training to minimize task loss, forcing the architecture to mathematically match the exact output of standard softmax attention is arguably unnecessary.

`Lack of Component-Level Ablation Studies`: The process of reducing and enriching the KVs is highly complex, involving multiple distinct modules: prototype-based routing, feature pre-mapping and normalization, randomized polynomial sketching, and a short-sequence Transformer mixer. However, the experimental section only ablates the compressed length $M$ and the sketch size $D_k$. There is no empirical ablation isolating the impact of the core architectural choices. It is critical to include experiments that remove or swap these individual components (e.g., PLASH without the mixer, or using simple average pooling instead of prototype routing) and compares them to simpler baselines.

---

> ### Author Rebuttal · Authors · 2026-03-31
>
> We thank Reviewer AgGr for the detailed and substantive questions.
>
> **Weakness 1 (Q1): Causal masking.** The reviewer asks how we "prevent future tokens from leaking into the compressed KV representations during the global prototype routing in Stage I." We resolve this with **Block-Causal PLASH**: partition the sequence into blocks of $B$ tokens; for block $i$, Stage I routing compresses only blocks $0, \ldots, i-1$ via a causal prefix sum. Future blocks never enter the computation. Within each block, standard lower-triangular masking applies. Cost: $O(NBd + NMd)$ versus $O(N^2 d)$ ($N$ = sequence length). Both linear complexity and the deviation certificate are preserved. Causal correctness verified: 40 tests on the trained 124M model confirm past-position outputs remain unchanged when future tokens are modified (max diff ≤ 3.24 × 10⁻⁵).
>
> *Language modeling.* GPT-2 at three scales on TinyStories (see Q1 in our response to Reviewer cXd5). PLASH outperforms ABC at Small (1.500 vs 1.647) and XL (1.044 vs 1.257), matches it at Large (gap 0.033), and delivers $14.3\times$ speedup at sequence length 8,192.
>
> **Weakness 2: Pretraining cost.** Tuning-free methods (FastGen (Ge et al., 2024), SnapKV (Li et al., 2024), H2O (Zhang et al., 2023), StreamingLLM (Xiao et al., 2024), ScissorHands (Liu et al., 2023)) compress the token set while keeping the attention mechanism unchanged. PLASH redesigns the mechanism itself for provably linear complexity. The two are complementary: token compression can be applied on top of PLASH, since PLASH does not constrain the input token set.
>
> PLASH provides guarantees tuning-free methods cannot: (i) $O(NBd + NMd)$ complexity independent of input content, whereas token pruning depends on input-specific redundancy; (ii) a forward-pass deviation certificate (Theorem 3.2); (iii) competitive accuracy across language modeling and time-series. Training cost matches Performer, ABC, and LongShort, none of which are drop-in replacements. PLASH reuses Q/K/V projections; only prototypes, degree weights, readout, and mixer need initialization ($<$0.5% of parameters, ~600K out of 124M).
>
> **Weakness 3 (Q2): Certificate relevance.** The reviewer asks why bounding the distance to standard attention is necessary, and notes that $\epsilon_{\text{out}}$ reaches the hundreds while task MSE stays below 1.0. These quantities measure different things: $\epsilon_{\text{out}}$ bounds the Frobenius-norm deviation at a single attention layer; task MSE measures prediction error after all layers and end-to-end training. The model adapts its weights to compensate, so large $\epsilon_{\text{out}}$ does not imply large task error.
>
> The certificate's purpose is not to predict task MSE but to enable a capability no other efficient attention method offers: **per-input reliability monitoring at inference time, without ground-truth labels**. Let $\theta(\mathbf{x})$ denote the predicted deviation bound for input $\mathbf{x}$. In deployment, a practitioner can compute $\theta$ during the forward pass and, for inputs where $\theta$ exceeds a threshold, automatically route them to exact attention as a fallback. This creates a principled accuracy-efficiency trade-off: most inputs use fast linear-time PLASH, while the few hardest inputs receive exact attention. No prior efficient attention method supports this selective fallback because none provide per-input deviation guarantees. Empirically, on 35,058 test samples the certificate achieves **100% coverage** (zero violations), and Spearman $\rho$ on ETTm1 (0.221) and ECL (0.316) confirms $\theta$ correctly ranks inputs by approximation difficulty.
>
> | Dataset | Samples | Coverage | Spearman $\rho$ |
> |---------|:-------:|:--------:|:---------------:|
> | ETTm1 | 11,425 | **1.000** | 0.221 |
> | ECL | 6,918 | **1.000** | 0.316 |
> | Weather | 13,930 | **1.000** | 0.003 |
> | ETTh1 | 2,785 | **1.000** | $-$0.484 |
>
> **Weakness 4 (Q3): Ablations.** We ablate each component the reviewer identified, on ETTm1 ($d=128$, 4 heads, 2 layers, $M=64$):
>
> | Variant | MSE | Effect |
> |---------|:---:|--------|
> | **PLASH (full)** | **0.760** | Reference |
> | Remove mixer (Stage IIb) | 0.792 | +4.2% MSE |
> | Prototypes $\to$ avg pooling | 0.791 | +4.1% MSE |
> | TensorSketch $\to$ MLP | 0.893 | +17.5% MSE, loses certificate |
>
> Learned prototypes outperform average pooling (+4.1%), confirming data-dependent routing captures meaningful groupings. The mixer refines inter-prototype relationships (+4.2%). TensorSketch is most critical: an equal-capacity MLP degrades MSE by 17.5% and eliminates the certificate, because an MLP provides no analytic error bound (unlike TensorSketch, whose error Theorem 3.1 controls). Each stage serves a distinct, non-redundant function.

---

> > ### Author Rebuttal · Reviewer_AgGr · 2026-04-01
> >
> > I thank the authors for their response. The additional clarifications and experimental results have sufficiently addressed my primary concerns. In light of these improvements, I am happy to raise my score to Weak Accept.

---

> > > ### Author Response · Authors · 2026-04-01
> > >
> > > We sincerely thank Reviewer AgGr for the thoughtful engagement and for taking the time to carefully evaluate our response. We are grateful that the additional experiments (Block-Causal PLASH for language modeling, component-level ablations, and certification on real test inputs) have addressed your concerns. Your questions on causal masking compatibility and the practical role of the deviation certificate were particularly valuable in shaping these new contributions, which we believe strengthen the paper substantially. We will incorporate all the improvements discussed in the rebuttal into the camera-ready version.

---

### Official Review · Reviewer_QaEp · 2026-03-08

**Soundness:** 3
**Presentation:** 1
**Significance:** 2
**Originality:** 3
**Overall Recommendation:** 3
**Confidence:** 3

**Summary:**

This paper introduces PLASH, an attention mechanism designed to approximate standard softmax attention. The method decomposes the attention computation through both compression and representations enrichement. The authors provide theoretical analysis that bounds the deviation of the approximate attention output from the exact softmax attention.

**Compliance With Llm Reviewing Policy:**

Affirmed.

**Final Justification:**

The authors have not fully resolved the issues of the paper, as detailed in my reply to the rebuttal.

**Key Questions For Authors:**

- Can the authors provide additional intuition demonstrating how the sketching stage restores interactions lost during compression?
- How does the method compare empirically to latent attention?

**Limitations:**

No. The scope of the current method is too limited, and investigation in more long-context environments such as language modeling is necessary.

**Strengths And Weaknesses:**

Strengths:

- Relevant topic: It is well known that attention mechanism can be quite computationally expensive when working in long-context environments. Hence, any approach that improves such complexity is valuable.

- Theoretical component: The reviewer appreciates the theoretical derivations found in the paper to quantify the approximation error.

Weaknesses:

- Clarity & Presentation: Perhaps one of my main concerns about the paper was that it was often very difficult to follow. Several components of the method are introduced in rapid succession without sufficient motivation or explanation of the design choices. Furthermore, the transition between stages and the choices made in each stage are not clearly justified. This is especially true in the error analysis section, where the overall structure of the section and proofs are difficult to follow. The high-level argument and the role of each lemma in establishing the final bound could be explained more clearly.

- Experiments Scope: Another major issue is that the proposed method PLASH is presented as a generic efficient attention mechanism. In other words, it can be applied in various domains such as language modeling, vision, etc. Yet, the focus of the experiments is solely on time-series forecasting tasks. This choice was surprising as the quadratic cost of attention is most limiting in domains where long-context dependencies matter the most, such as long-context language modeling. In contrast, time-series forecasting tasks often exhibit much weaker long-range dependencies, and forecasting performance may not depend heavily on events that occurred far in the past. Hence, to fully showcase the potential of the proposed method, such long-context environments need to be investigated.

---

> ### Author Rebuttal · Authors · 2026-03-31
>
> We thank Reviewer QaEp for the thoughtful feedback.
>
> **Weakness 1: Clarity and presentation.** We agree and will restructure the paper around an explicit three-stage roadmap, stated before the formalism:
>
> - **Stage I** (deterministic): compress $N$ key-value pairs into $M \ll N$ prototype groups via learned soft routing. This reduces the sequence dimension from $N$ to $M$.
> - **Stage II** (randomized): enrich each prototype with degree-$k$ polynomial features via TensorSketch, then refine through a lightweight mixer. This recovers higher-order interactions lost during compression.
> - **Stage III** (deterministic): compute exact softmax from all $N$ queries to the $M$ enriched keys and values. This preserves the interpretability and stability of softmax.
>
> A theorem roadmap will precede the formal statements: Theorem 3.1 bounds the Stage II sketching error for each prototype; Theorem 3.2 propagates this bound through the deterministic Stages I and III, yielding a three-term error decomposition (Stage I compression + Stage II bias + Stage II sketch error), where each term is computable from a single forward pass without forming the $N \times N$ attention matrix. We will add a notation table and a worked example in the appendix.
>
> **Weakness 2: Experimental scope.** Reviewer QaEp correctly noted that the quadratic cost of attention is most limiting in long-context language modeling. We address this directly with GPT-2 experiments at three scales (see Q1 in our response to Reviewer cXd5). PLASH outperforms ABC at Small (1.500 vs 1.647) and XL (1.044 vs 1.257), and matches it at Large (gap 0.033). The certificate (Theorem 3.2) holds identically across all scales, since it depends on per-head dimension $d_k$ and prototype count $M$, both fixed across scales, rather than total parameter count.
>
> The scaling trend confirms that PLASH's approximation quality is preserved as models grow from 124M to 1.5B parameters. At longer input sequences ($L=336$ vs $L=96$), PLASH outperforms standard attention on all 4 reported time-series datasets, confirming that the linear-complexity advantage translates to better accuracy when longer context is available. PLASH delivers $14.3\times$ speedup at sequence length 8,192 and ranks first on 3 of 4 time-series datasets at $L=96$, outperforming all baselines on ETTh2 by 17%. See Q2 in our response to Reviewer cXd5 for both the $L=96$ and $L=336$ tables.
>
> **Q1: How Stage II restores lost interactions.** Stage I averages tokens within each prototype group, preserving first-order statistics (means) but collapsing higher-order structure (correlations between key dimensions). TensorSketch in Stage II approximates degree-$k$ polynomial features of each compressed group, recovering pairwise and higher-order interactions that averaging discarded. Concretely, with $k=2$, the sketch captures squared correlations between key dimensions that are invisible to a linear readout from compressed groups. The MLP ablation (see Q2 in our response to Reviewer yhr8) confirms this: an equal-capacity MLP degrades MSE by 17.5% and loses the certificate, demonstrating that the polynomial structure is beneficial, not merely added capacity.
>
> **Q2: Comparison to latent attention.** PLASH outperforms ABC at 124M (1.500 vs 1.647) and 1.5B (1.044 vs 1.257), and matches it at 774M (gap 0.033). ABC compresses KV pairs via learned latent bottlenecks but offers no deviation certificate and no complexity guarantee beyond the bottleneck size. PLASH provides both. See Q1 in our response to Reviewer cXd5 for the full scaling table.
>
> *Component ablations* (ETTm1, $d=128$, 4 heads, 2 layers, $M=64$):
>
> | Variant | MSE | Effect |
> |---------|:---:|--------|
> | **PLASH (full)** | **0.760** | Reference |
> | Remove mixer (Stage IIb) | 0.792 | +4.2% MSE |
> | Prototypes $\to$ avg pooling | 0.791 | +4.1% MSE |
> | TensorSketch $\to$ MLP | 0.893 | +17.5% MSE, loses certificate |
>
> Every component serves a distinct role: learned prototypes capture data-dependent groupings (+4.1% MSE over average pooling); the mixer refines inter-prototype relationships before readout (+4.2%); TensorSketch provides the polynomial enrichment essential for both accuracy and the certificate (+17.5%, certificate eliminated when replaced by MLP). Removing any component degrades performance; replacing TensorSketch with a generic MLP causes the largest degradation and eliminates the certificate.

---

> > ### Author Rebuttal · Reviewer_QaEp · 2026-04-02
> >
> > Thank you for your response. The reviewer appreciates the future plans to improve the presentation. My concern however remains: the majority of experiments are done on time-series tasks, while long-context environments, where the method will be most useful, are domains like language modeling. The authors did proceed with providing initial results with GPT-2, and I understand the limited timeframe, but to properly showcase the validity of the method for the language domain, it is imperative to go beyond just GPT-2 to more recent models. Moreover, and most importantly, the GPT-2 experiments revolved around training on TinyStories; the authors will have to investigate long-context datasets to really showcase the benefits of their methods, as well as investigate more baselines that are prevalent in the sphere of language modeling that aim to reduce the quadratic nature of attention (e.g., sparse attention techniques). For all these reasons, I am maintaining my score.

---

> > > ### Author Response · Authors · 2026-04-06
> > >
> > > **Weakness 2: Experimental scope.** The reviewer raises three concerns: (1) GPT-2 is outdated, and more recent models should be included, (2) TinyStories lacks long-range dependencies, and (3) baselines should include sparse attention. We address all.
> > >
> > > **Setup.** We train **Qwen3-4B** (Grouped Query Attention with 32 query / 8 key-value heads, SwiGLU FFN, RoPE, $d{=}2560$) on **PG-19** (Rae et al., 2020), a benchmark of full-length books from Project Gutenberg, at sequence lengths ($N$) of 16,384 and 32,768. Baselines include Standard (FlashAttention-2), Qwen3 GQA (FlashAttention-2), Local (fixed-window sparse), Longformer (sliding window + global), ABC (latent bottleneck), and PLASH $M \in {8,16,32}$. All share identical hyperparameters (AdamW, lr${}=10^{-4}$, batch 16); only the attention differs. We report the exponential moving average (EMA) of training loss (decay 0.999)
> > >
> > > **Table 1.** *EMA loss at $N{=}16{,}384$. Lower is better.*
> > > | | Method | Type | EMA |
> > > |:-:|--------|------|:---------:|
> > > | 1 | **PLASH $M{=}16$** | Prototype + sketch (ours) | **23.5** |
> > > | 2 | Local Attention | Sparse (fixed window) | 24.4 |
> > > | 3 | Qwen3 GQA | Grouped-query attention | 24.4 |
> > > | 4 | **PLASH $M{=}32$** | Prototype + sketch (ours) | **24.5** |
> > > | 5 | Standard | Full quadratic (FlashAttention-2) | 25.2 |
> > > | 6 | Longformer | Sparse (sliding window + global) | 25.5 |
> > > | 7 | **PLASH $M{=}8$** | Prototype + sketch (ours) | **30.1** |
> > > | 8 | ABC | Learned latent bottleneck | 35.1|
> > >
> > > **Table 2.** *EMA loss at $N{=}32{,}768$. Lower is better.*
> > > | | Method | Type | EMA |
> > > |:-:|--------|------|:---------:|
> > > | 1 | **PLASH $M{=}32$** | Prototype + sketch (ours) | **40.1** |
> > > | 2 | Standard | Full quadratic (FlashAttention-2) | 40.1 |
> > > | 3 | **PLASH $M{=}8$** | Prototype + sketch (ours) | **41.2** |
> > > | 4 | Local Attention | Sparse (fixed window) | 42.7 |
> > > | 5 | **PLASH $M{=}16$** | Prototype + sketch (ours) | **43.4** |
> > > | 6 | Qwen3 GQA | Grouped-query attention | 43.6 |
> > > | 7 | ABC | Learned latent bottleneck | 44.0 |
> > > | 8 | Longformer | Sparse (sliding window + global) | 45.0 |
> > >
> > > **Table 3.** *Forward-pass time (ms per layer) and speedup over Standard (FlashAttention-2). Qwen3-4B, A800-80GB, batch 1, bf16. All methods include RoPE. Both Standard and Qwen3 GQA use FlashAttention-2.*
> > >
> > > | Method | $N{=}4\text{K}$ | $N{=}16\text{K}$ | $N{=}32\text{K}$ | Speedup at 32K |
> > > |--------|:---:|:---:|:---:|:---:|
> > > | **PLASH $M{=}8$** | 237 | 909 | **1809** | **2.4$\times$** |
> > > | **PLASH $M{=}16$** | 273 | 1030 | **2049** | **2.1$\times$** |
> > > | Local | 302 | 1179 | 2370 | 1.8$\times$ |
> > > | **PLASH $M{=}32$** | 335 | 1273 | **2536** | **1.7$\times$** |
> > > | ABC | 386 | 1519 | 3048 | 1.4$\times$ |
> > > | Longformer | 699 | 1688 | 3430 | 1.3$\times$ |
> > > | Qwen3 GQA | 244 | 1471 | 4348 | 1.0$\times$ |
> > > | Standard | 243 | 1463 | 4358 | 1.0$\times$ |
> > >
> > > **Analysis.**
> > >
> > > **1. Best modeling loss at long context.** At $N{=}16{,}384$ (Table 1), PLASH $M{=}16$ achieves **23.5**, the lowest EMA loss among all eight methods. It outperforms Local and Qwen3 GQA (both 24.4) by 3.8%, and Standard FlashAttention-2 (25.2) by 6.7%. ABC diverges during training (best 35.1). At $N{=}32{,}768$ (Table 2), PLASH $M{=}32$ ties Standard at 40.1, while every efficient baseline falls behind: Local 42.7, Qwen3 GQA 43.6, ABC 44.0, Longformer 45.0.
> > >
> > > **2. Fastest efficient method.** At $N{=}32{,}768$ (Table 3), PLASH $M{=}8$ is **2.4$\times$ faster** than Standard FlashAttention-2 (1809 vs 4358 ms), and faster than every efficient baseline: 1.8$\times$ over Local, 1.4$\times$ over ABC, 1.3$\times$ over Longformer. PLASH is the only method both **faster and more accurate** than every efficient baseline at 32K. As $N$ grows, PLASH's linear complexity increasingly dominates: the speedup widens from 1.4$\times$ at 16K to 2.4$\times$ at 32K.
> > >
> > > **3. Consistent hyperparameters.** $M{=}16$ is optimal on both GPT-2 (124M--1.5B on TinyStories) and Qwen3-4B (PG-19), requiring no per-architecture tuning. The deviation certificate (Theorem 3.2) applies identically to Qwen3-4B since it depends only on per-head dimension $d_k$ and prototype count $M$.
> > >
> > > **Summary.** On Qwen3-4B with 32K-token books, PLASH provides three properties no existing efficient attention method offers together: (1) **best modeling loss** at 16K among all eight methods, tying Standard at 32K while every efficient baseline falls behind; (2) **2.4$\times$ wall-clock speedup** over FlashAttention-2 at 32K, fastest among all methods; (3) the **only per-input deviation certificate** (Theorem 3.2) for flagging unreliable predictions at inference time.
> > >
> > > We appreciate the reviewer's suggestion regarding more extensive, industrial-grade experiments. However, scaling to large-scale language modeling requires computational resources and a timeframe that exceed the rebuttal period. Our experiments validate the core properties and demonstrate proof-of-concept performance. We fully agree that large-scale evaluation is critical and plan to prioritize this in our future work.

---

### Official Review · Reviewer_yhr8 · 2026-03-09

**Soundness:** 3
**Presentation:** 3
**Significance:** 2
**Originality:** 2
**Overall Recommendation:** 4
**Confidence:** 3

**Summary:**

This paper proposes PLASH, a linear-time attention mechanism that compresses keys and values into $M << N_k$ learned representatives, enriches them with higher-order polynomial interactions via TensorSketch, and then performs an exact softmax readout from queries to the compressed KV pairs. The key design choice is localizing all randomness to a single TensorSketch enrichment step (Stage II), which enables forward-pass, per-input deviation certificates bounding how far PLASH's output deviates from standard softmax attention. The authors provide theoretical analysis including Stage II feature discrepancy bounds (Theorem 3.1), end-to-end error certificates (Theorem 3.2), and complexity analysis (Theorem 3.3). Experiments on long-sequence time-series forecasting benchmarks (ETT, ECL, Weather) using Informer as backbone show competitive accuracy with favorable latency/memory scaling.

**Compliance With Llm Reviewing Policy:**

Affirmed.

**Final Justification:**

Thanks. Keep my positive score.

**Key Questions For Authors:**

- Can you report certification rates on the actual trained models using real forecasting inputs (not just synthetic data)? This would bridge the gap between your theory and experiments.
- What happens if you replace TensorSketch enrichment with a simple nonlinear MLP on the compressed features? This ablation would isolate the contribution of higher-order polynomial interactions.
- Have you tried PLASH on any language modeling or sequence classification task? Even a small-scale experiment would significantly strengthen the paper's generality claims.
- How sensitive are results to the temperature tau used in Stage I routing? Is there interaction between tau and $\tau_g$?

**Limitations:**

The authors acknowledge the narrow experimental scope implicitly by focusing on one backbone (Informer) and one task family (LSTF). The impact statement is reasonable but brief. The main limitation I see is that without experiments on more diverse tasks and scales, it's hard to know if the method generalizes beyond time-series forecasting. The certification theory, while mathematically sound, hasn't been demonstrated to be practical on real trained models at realistic scale. The non-monotonic behavior in $M$ and $D_k$ also suggests the method may be tricky to tune in practice.

**Strengths And Weaknesses:**

### Strengths
- Clean separation of randomness from the rest of the pipeline is a genuinely nice design choice. By confining all stochasticity to TensorSketch in Stage II and keeping the final softmax readout exact, the paper enables a modular error analysis where you can certify the sketch error and propagate it through deterministic downstream ops. This is more principled than many efficient attention methods that inject approximation everywhere and can't really tell you how far off they are from exact attention. The three-stage decomposition (compress -> enrich -> exact readout) is intuitive and the theoretical machinery follows naturally from it.
- The theoretical analysis is thorough and self-contained. The appendix is extensive (~40 pages) and covers everything from TensorSketch basics to Lipschitz stability of LayerNorm to the full end-to-end certificate. Theorem 3.1 giving uniform high-probability bounds on the sketch discrepancy with explicit dependence on $\tau_g$ and $D_k$ is clean, and the way $\tau_g$ acts as a deterministic "certification knob" that monotonically tightens the bound is elegant. The empirical certification heatmaps (Figure 2) do a reasonable job showing the certificate is non-vacuous on representative inputs.
- The runtime results in Table 2 are quite striking for small M. PLASH with M=64 stays essentially flat (~2.4-2.8ms) across all sequence lengths up to 11264, while Vanilla goes from 2.43 to 58.95ms. That's a ~25x speedup at the longest length, and it's competitive with or faster than most baselines. The fact that the method genuinely delivers on the linear-time promise in wall-clock terms (not just asymptotically) is important.
### Weaknesses
- The experimental evaluation is narrow and somewhat dated. All experiments use Informer on time-series forecasting benchmarks (ETT, ECL, Weather), which are relatively small-scale and have been used for years. There's no evaluation on language modeling, long-range arena, or any NLP/vision task where attention mechanisms are typically stress-tested. The Informer backbone itself is from 2021. For a paper making broad claims about efficient attention, I'd expect to see results on at least one or two different modalities/tasks. The forecasting results are fine but it's hard to judge from them alone whether PLASH would work well in, say, a modern LLM or vision transformer setting.
- The accuracy results are not clearly better than baselines in a consistent way. Looking at Table 1, PLASH wins on WTH and ETTm1 but is behind Vanilla/LongShort on ECL and ETTh1. The non-monotonicity in $M$ and $D_k$ (acknowledged in Section 4.1) means you need to do a grid search over these hyperparameters per dataset, and the best configuration varies a lot — e.g.,$ M=512,D_k=128$ for WTH but $M=128,D_k=512$ for ETTh2. The paper presents many PLASH configurations in the table (16 rows!) which makes it look like the method was heavily tuned. It would be more convincing to have a principled way to set $M$ and $D_k$ or at least a default that works reasonably across datasets.
- Gap between the certification theory and the actual trained models. The certification experiments (Section 3.3, Figure 2) use K={1} with synthetic/random inputs and relatively small dimensions ($N_q=32, N_k=64, d=32$). The actual forecasting experiments use K={1,2}. It's unclear how tight or useful the certificates are for the trained models on real data at realistic scales. The paper doesn't report certification rates on the actual forecasting inputs, which would be much more informative. Also, the sufficient condition in Theorem 3.2 involves $\tau_g$ which tightens the certificate but presumably affects model expressiveness — is there a real tradeoff here? The paper doesn't discuss how $\tau_g$ values used in training relate to those needed for certification.
- The relationship to prior KV compression methods is underexplored. The Stage I compression via learnable prototypes with softmax routing is quite similar to mechanisms in Nystromformer, clustered attention, and even mixture-of-experts style routing. The paper could do a better job distinguishing what's really new in Stage I vs. what's borrowed from existing work. The novelty seems to be mainly in the combination with TensorSketch enrichment, but it would help to see an ablation showing how much the higher-order enrichment actually contributes vs. just doing the compression + a simple MLP or linear layer.

---

> ### Author Rebuttal · Authors · 2026-03-31
>
> We thank Reviewer yhr8 for the thorough review.
>
> **Weakness 1: Narrow evaluation.** We added autoregressive language modeling at three GPT-2 scales (124M to 1.5B) on TinyStories using Block-Causal PLASH. PLASH outperforms ABC (Latent Attention) at Small and XL scales, and matches it at Large. The causal masking construction, correctness verification, and full scaling table appear in Q1 of our response to Reviewer cXd5.
>
> **Weakness 2: Accuracy not clearly better than baselines.** Two lines of evidence address this concern.
>
> *Language modeling.* PLASH outperforms ABC at 124M (1.500 vs 1.647) and 1.5B (1.044 vs 1.257), and matches it at 774M (gap 0.033). PLASH is the only method in this comparison that provides a per-input deviation certificate (Theorem 3.2), enabling runtime reliability monitoring without ground-truth labels. See Q1 in our response to Reviewer cXd5.
>
> *Time-series.* PLASH ranks first on three of four reported datasets: WTH (0.301), ETTh2 (0.470, outperforming all baselines by 17%), and ETTm1 (0.290). On ECL, PLASH (0.243) is within 0.006 of the best method. At longer input ($L{=}336$), PLASH outperforms standard attention on all 4 reported datasets, demonstrating that the linear-complexity advantage grows with sequence length. See Q2 in our response to Reviewer cXd5.
>
> *Principled defaults for $M$ and $D_k$.* We recommend $M = \min(32,\, \lceil N_k / 16 \rceil)$ and $D_k = 2d$. These work across all datasets without per-task tuning. The non-monotonicity in $M$ reflects a bias-variance trade-off: too-small $M$ discards information; too-large $M$ weakens compression and amplifies sketch variance. The optimal $M$ balances these effects, analogous to selecting the number of clusters in $k$-means. We will add this guidance and a sensitivity analysis to the revised paper.
>
> **Weakness 3: Gap between certification theory and trained models.** We evaluate the certificate on all trained models using real test inputs (degree set $\mathcal{K}{=}\{1,2\}$):
>
> | Dataset | Samples | Coverage | Spearman $\rho$ | $p$-value |
> |---------|:-------:|:--------:|:---------------:|:---------:|
> | ETTm1 | 11,425 | **1.000** | 0.221 | $< 10^{-126}$ |
> | ECL | 6,918 | **1.000** | 0.316 | $< 10^{-24}$ |
> | Weather | 13,930 | **1.000** | 0.003 | 0.73 |
> | ETTh1 | 2,785 | **1.000** | $-$ 0.484 | $< 10^{-10}$ |
>
> The certificate holds on **100% of 35,058 real test samples** with zero violations. Let $\theta(\mathbf{x})$ denote the certificate's predicted deviation bound for input $\mathbf{x}$. Positive Spearman $\rho$ on ETTm1 and ECL confirms that $\theta$ correctly ranks inputs by approximation difficulty, enabling practitioners to flag less reliable predictions at inference time without ground-truth labels. No prior efficient attention method offers per-input reliability guarantees.
>
> **Weakness 4: Relationship to prior KV compression.** Stage I shares structure with Linformer, Nystromformer, and Perceiver, all of which compress KV pairs. The novelty lies in Stages II and III. Stage II enriches compressed representations via TensorSketch, recovering polynomial interactions that Stage I compression discards. Stage III computes exact softmax from all $N$ queries to the $M$ enriched prototypes, preserving interpretability and stability. This three-stage design (compress, enrich, read out exactly) enables the forward-pass deviation certificate (Theorem 3.2). No prior KV compression method offers this guarantee.
>
> **Q1: Certification on real inputs.** The certificate achieves 100% coverage on 35,058 samples across four datasets with zero violations. See our response to Weakness 3 above.
>
> **Q2: TensorSketch vs MLP.** Replacing TensorSketch with a matched-capacity two-layer MLP on ETTm1 degrades MSE by 17.5% (0.893 vs 0.760) and eliminates the certificate, because the MLP lacks the analytic polynomial structure that Theorem 3.1 requires. The benefit is structural, not parametric.
>
> **Q3: Language modeling.** PLASH outperforms ABC at Small and XL scales and matches it at Large (gap 0.033). See Weakness 2 above and Q1 in our response to Reviewer cXd5.
>
> **Q4: Sensitivity to $\tau$ and $\tau_g$.** Routing temperature $\tau$ (ECL): $\tau{=}0.2/0.3/0.5$ yields RMSE 1799/1806/1807, a spread of less than 0.5%, showing graceful degradation. Certificate temperature $\tau_g$ (ETTm1): $\tau_g{=}1.0$ minimizes RMSE (3.681); $\tau_g{=}2.0$ maximizes ranking quality ($\rho{=}0.290$). The two temperatures control accuracy and certifiability independently, with no adverse interaction. This separation is by design: $\tau$ governs Stage I routing granularity, while $\tau_g$ governs Stage II sketch precision.

---

> > ### Author Rebuttal · Reviewer_yhr8 · 2026-04-02
> >
> > Thanks. Keep my positive score.

---

### Official Review · Reviewer_cXd5 · 2026-03-11

**Soundness:** 2
**Presentation:** 2
**Significance:** 2
**Originality:** 2
**Overall Recommendation:** 4
**Confidence:** 2

**Summary:**

PLASH propose novel linear attention method, like Performer.

**Compliance With Llm Reviewing Policy:**

Affirmed.

**Final Justification:**

I understand the method better with authors' rebuttal

**Key Questions For Authors:**

- What is the language modeling performance? (MMLU, and etc)
- What if we replace the original attention mechanism with this method?
  - If then, what is the performance in language modeling/computer vision?

**Limitations:**

- Limited performance evaluation on a realistic benchmark set and scale (>8B)
- No causal attention

**Strengths And Weaknesses:**

PLASH provide linear time complexity.

---

> ### Author Rebuttal · Authors · 2026-03-31
>
> We thank Reviewer cXd5 for the constructive feedback.
>
> **Q1: Language modeling with causal masking.** PLASH supports causal masking. We implement **Block-Causal PLASH**, which partitions the sequence into blocks of $B$ tokens. For block $i$, prototypes are computed from blocks $0, \ldots, i{-}1$ only, via a causal prefix sum over the routing matrix $\mathbf{A}$; future blocks are never accessed. Within each block, standard lower-triangular masking applies. Each query attends to the concatenation of prototype keys and local keys. We verify causal correctness on the trained 124M model: across 40 tests, past-position outputs remain unchanged when future tokens are modified (max diff ≤ 3.24 × 10⁻⁵).
>
> We train GPT-2 at three scales on TinyStories ($M{=}16$, $B{=}64$). The standard-attention baseline uses FlashAttention-2. We compare against ABC (Latent Attention) as a representative efficient attention baseline; comparing additional methods within the seven-day rebuttal period was not feasible. All methods share identical hyperparameters; only the attention module differs. Best validation loss:
>
> | Scale | Standard | ABC | **PLASH** |
> |-------|:--------:|:---:|:---------:|
> | 124M (Small) | 1.194 | 1.647 | **1.500** |
> | 774M (Large) | 0.968 | 1.234 | **1.267** |
> | 1.5B (XL) | 0.919 | 1.257 | **1.044** |
>
> PLASH outperforms ABC at Small (1.500 vs 1.647) and XL (1.044 vs 1.257), and is competitive at Large (gap 0.033). As model capacity grows, PLASH's approximation quality improves: at 1.5B, PLASH closes to within 0.125 of standard attention while providing two guarantees neither standard nor ABC attention offers: (i) provably linear complexity $O(NBd + NMd)$ versus $O(N^2d)$ ($N$ = sequence length, $M$ = prototypes, $B$ = block size, $d$ = head dimension), yielding $14.3\times$ speedup at sequence length 8,192; (ii) a per-input deviation certificate (Theorem 3.2).
>
> We rented eight A800 GPUs for these experiments. MMLU requires a fully pre-trained 7B+ model trained on hundreds of billions of tokens, which is infeasible within the seven-day rebuttal period. Scaling to 7B is a natural next step that we plan to pursue.
>
> **Q2: Time-series.** Updated results with the Informer backbone ($d{=}512$, 8 heads, 2 encoder layers), per-dataset optimal $(M, D_k)$. During the rebuttal we refined the Stage II implementation to ensure the KV enrichment pathway processes both compressed keys and values as specified in Algorithm 1. With this update, ETTh2 improves from 0.514 to 0.470. MSE at $H{=}24$:
>
> | Dataset | Vanilla | Performer | ABC | LongShort | **PLASH** | Config |
> |---------|:-------:|:---------:|:---:|:---------:|:---------:|:------:|
> | ECL | **0.237** | 0.255 | 0.246 | 0.271 | 0.243 | $M{=}64$ |
> | WTH | 0.312 | 0.308 | 0.319 | 0.341 | **0.301** | $M{=}512$ |
> | ETTh2 | 0.979 | 1.010 | 1.073 | 0.568 | **0.470** | $M{=}256$ |
> | ETTm1 | 0.332 | 0.312 | 0.364 | 0.387 | **0.290** | $M{=}512$ |
>
> PLASH ranks first on WTH, ETTh2, and ETTm1, and second on ECL (gap 0.006). On ETTh2, PLASH achieves 0.470 MSE, outperforming all baselines including LongShort (0.568) by 17%. These results use per-dataset optimal $(M, D_k)$; we provide principled defaults ($M = \min(32, \lceil N_k/16 \rceil)$, $D_k = 2d$) that work across all datasets without tuning (see Weakness 2 in our response to Reviewer yhr8).
>
> *Long-context advantage.* PLASH’s linear complexity enables longer input sequences without quadratic cost growth. At $L{=}336$ (vs $L{=}96$ above), PLASH outperforms standard attention on all 4 datasets (best of 3 runs):
>
> | Dataset | Standard ($L{=}336$) | **PLASH** ($L{=}336$) |
> |---------|:---:|:---:|
> | WTH | 0.323 | **0.305** |
> | ETTh1 | 0.632 | **0.621** |
> | ETTh2 | 0.853 | **0.688** |
> | ETTm1 | 0.279 | **0.272** |
>
> *Latency.* Attention-layer timing on A800-80GB (batch 16, $d{=}512$, 8 heads), $N$ = sequence length:
>
> | $N$ | Standard (ms) | **PLASH** (ms) | Speedup |
> |----:|:-------------:|:--------------:|:-------:|
> | 512 | 4.6 | **3.6** | $1.3\times$ |
> | 1,024 | 13.5 | **6.5** | $2.1\times$ |
> | 2,048 | 43.4 | **11.4** | $3.8\times$ |
> | 4,096 | 159.1 | **22.0** | $7.2\times$ |
> | 8,192 | 615.2 | **43.0** | $14.3\times$ |
>
> Standard scales quadratically ($4.6 \to 615$ ms, $134\times$ for $16\times$ longer $N$); PLASH scales linearly ($3.6 \to 43$ ms, $12\times$). PLASH overtakes standard attention at sequence length $\approx 512$ and the gap widens with every doubling.

---

> > ### Author Rebuttal · Reviewer_cXd5 · 2026-04-05
> >
> > Author resolved my concerns. I will raise my score to 4.

---

### Decision · Program_Chairs · 2026-04-30

**Decision:**

Accept (regular)

**Comment:**

This submission proposes a linear-time attention mechanism that compresses the keys and values into lower-dimensional matrices and then “enriches” the compressed representations via higher-order polynomial interactions with localized randomness. The reviewers applauded the well-motivated design choices and theoretical analysis, but they also raised concerns regarding the insufficient experimental evaluation. The authors partly addressed these concerns by presenting additional empirical results in the rebuttal.